

# Rate coefficients for the reactions of OH radical with C3-C11
# alkanes determined by the relative rate technique
Yanyan Xin [1,2], Xiaoxiu Lun [1], Shuyang Xie [2], Junfeng Liu [2], Chengtang Liu [2]*, Yujing
Mu [2]
[1] Beijing Forestry University, Beijing, 100083, China
[2] Research Center for Eco-Environmental Sciences, Chinese Academy of Sciences,
Beijing, 100085, China.
*Correspondence to: Chengtang Liu (ctliu@rcees.ac.cn).
**Abstract**: Rate coefficients for the reactions of OH radicals with C3-C11 alkanes were
determined using the multivariate relative rate technique in various bath gases ($N_2$, Air,
$O_2$). A total of 25 relative rate coefficients at room temperature and 24 Arrhenius
expressions in the temperature range of 273-323 K were obtained. Notably, a new room
temperature relative rate constant for 3-methylheptane that had not been previously
reported were determined, and the obtained $K_{OH}$ values (in units of $10^{-12}$ $cm^3 \cdot molecule^{-1} \cdot s^{-1}$) in different bath gases were $N_2$, 7.90±0.25; Air, 7.93±0.33; and $O_2$, 7.36±0.11.
Interestingly, whilst results for n-alkanes agreed well with available structure activity
relationship (SAR) calculations, the three cyclo-alkanes and two trimethylpentane were
found to be less reactive than predicted by SAR. Conversely, the SAR estimate for 2,3-
dimethylbutane were approximately 22% lower than the experimental value,
highlighting that the limited understanding of the oxidation chemistry of these
compounds. Arrhenius expressions (in units of $cm^3 \cdot molecule^{-1} \cdot s^{-1}$) for the reactions of
various cyclo- and branched alkanes with OH were determined for the first time:
methylcyclopentane, $(1.62±0.14)×10^{-11} exp [-(256±25)/T]$ ; 2-
methylhexane, $(1.22±0.04)×10^{-11} exp [-(206±9)/T]$ ; 3-
methylhexane, $(2.27±0.31)×10^{-11} exp [-(559±42)/T]$ ; 2-methylheptane,
$(1.62±0.37)×10^{-11} exp [-(265±70)/T]$ , and 3-methylheptane,
$(3.54±0.45)×10^{-11} exp [-(374±49)/T]$ . In addition, the rate coefficients for the 24





previous studied OH + alkanes reactions in different bath gases were consistent with
existing literature values, demonstrating the reliability and efficiency of this method for
simultaneous investigation of gas-phase reaction kinetics.
**Keywords:** Relative rate coefficients; Atmospheric simulation chamber; Alkanes; OH
radical; Arrhenius expressions

## 1. Introduction

Volatile organic compounds (VOCs), a category of compounds found ubiquitously
in the atmosphere, primarily consist of alkanes, alkenes, aromatics and oxygenated
volatile organic compounds (OVOCs) (Lewis et al., 2000; Goldstein and Galbally, 2007;
Anderson et al., 2004). Research has shown that alkanes, including straight-chain,
branched-chain, and cyclic alkanes within the C3-C11 range, often constitute a
significant portion. For example, recent studies conducted by Liang et al. and Dunmore
et al. in major cities in China and the U.K. have indicated that C2-C12 alkanes make
up 66.5% and 50% of the local hydrocarbon content, respectively (Liang et al., 2023;
Dunmore et al., 2015). The primary mechanism for alkanes removal involves hydrogen
abstraction reactions with OH· and $NO_3$·, and the dehydrogenation of alkanes leads to
the formation of alkyl radicals (R·), which subsequently react with $O_2$ to generate
alkylperoxy radicals ($RO_2$·). It should be pointed that the rate constants for the reaction
of alkanes with OH· ($K_{OH}$) fall in the range of 0.9 to $11 \times 10^{-12}$ $cm^3 \cdot mol^{-1} \cdot s^{-1}$, which is
approximately five orders of magnitude faster than the reaction with $NO_3$·. The reaction
with OH radicals stands as the principal pathway for the atmospheric oxidation of
alkanes during the daytime. Thus, accurately determining rate constants with OH
radicals is fundamental in evaluating their environmental impact (Finlayson-Pitts and
Pitts, 1997; Atkinson, 2000).
Numerous laboratories have conducted research on the kinetics of the reaction
between alkanes and OH radicals using the absolute rate constant method and the
relative rate constant method. The absolute rate constant method involves calculating
the reaction kinetics parameter $K_{OH}$ for organic compounds with OH radicals during



the experimental process by directly measuring changes in OH radical concentration or
the concentration of the target compound. Greiner measured the first kinetic data for
the reaction of OH radicals with three alkanes in the Ar system at 300 K using the flash
photolysis-resonance fluorescence technique (Greiner, 1967). Over the next decade,
Gorse et al., Overend et al. and Darnall et al. obtained kinetic data for the reaction of
OH radicals with selected alkanes in the carbon monoxide, He and $N_2$ system,
respectively (Gorse and Volman, 1974; Overend et al., 1975; Darnall et al., 1978). Due
to the challenge of directly detecting OH radicals with very short lifetimes, the absolute
rate method is used less frequently. Alternatively, the relative rate method does not
require precise VOC concentration levels or direct detection of OH radicals, and this
approach is more widely used to determine $K_{OH}$ values for organic compounds. From
1980s to 2010s, dozens of papers for the rate coefficients of alkanes with OH measured
by relative rate mehod have been published. For example, Shaw et al. and Phan and Li
obtained rate constants of a series of alkanes in the $N_2$/He system (Phan and Li, 2017;
Shaw et al., 2018; Shaw et al., 2020). Anderson et al. obtained the $K_{OH}$ of C2-C8 several
n-alkanes and cyclic alkanes by the relative technique in the air system at 296 ± 4 K
(Anderson et al., 2004). However, the majority of experiments were conducted solely
on C2-C6 alkanes, more complex and multifunctional alkanes are often poorly
constrained or unmeasured.
Temperature has an important influence on the reaction rate constants of alkanes and
OH radicals. The reaction rate constants of several n-alkanes with OH radicals
measured by Greiner increased by about 70% in the range of 300-500 K (Greiner,
1970a). Perry et al's research found that the rate constants of n-butane multiplied by
72% as the temperature rose from 297 K to 420 K (Perry et al., 1976). And the rate
coefficients of 10 n-alkanes and cycloalkanes obtained by Donahue et al. also increased
in varying degrees at 300-390 K (Donahue et al., 1998). However, most reported
experimental studies on the reactivity of OH radicals with a series of alkanes focus on
temperatures ≥290 K (Greiner, 1970a; Perry et al., 1976; Finlaysonpitts et al., 1993;
Donahue et al., 1998; Atkinson, 2003; Badra and Farooq, 2015), with relatively few
studies at low temperatures (Demore and Bayes, 1999; Li et al., 2006; Wilson et al.,





2006; Sprengnether et al., 2009; Crawford et al., 2011). In addition, a further alkane
had only two, or fewer, individual OH rate coefficient measurements available in the
mentioned temperature range, e.g., 3-methylheptane, and it is unclear whether the rate
constants for the reactions of OH radicals with alkanes differ in a mixed system
containing oxygen compared to an inert gas system. Therefore, further investigations
are required to explore the variations in the rate constants for different types of alkanes
at various temperatures in different bath gases.
In this study, the rate constants for the reactions of 25 different C3-C11 alkanes with
OH radicals were determined using the multivariate relative rate method under different
bath gases ($N_2$, Air, $O_2$), including linear alkanes, cycloalkanes, and methyl-alkanes. To
validate the data and investigate the effect of $O_2$ on the rate constants for the reaction
between alkanes and OH radicals, multiple comparisons were made with previous
literature and structure–activity relationship (SAR) estimated values. Additionally, the
rate constants of certain straight-chain, branched-chain, and methyl-cycloalkanes were
measured at 273-323 K.

## 2. Methods

### 2.1 Experiment

2.1.1 Atmospheric simulation chamber

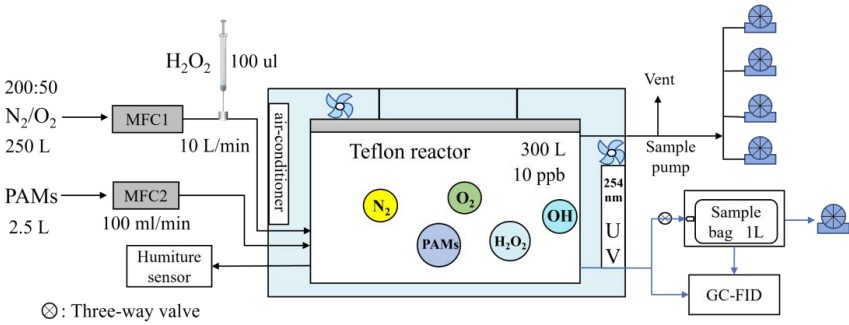


Figure 1. A schematic of the experimental device

As shown in Fig. 1, the chamber experiments were performed at atmospheric



pressure in a climate-controlled box with a temperature range of 263-333 K (accuracy
of ± 0.1 K). A 300 L Teflon airbag was suspended in the climate-controlled box to serve
as the reaction system. The box was equipped with two Teflon-coated fans for rapid
chemical mixing and a 254 nm ultraviolet lamp for photolysis of hydrogen peroxide
($H_2O_2$) to produce OH radicals. The inner walls of climate-controlled box were
constructed with reflective steel plates to enhance ultraviolet light utilization. Bath gas
($N_2$ or $O_2$) and NMHCs were introduced into the Teflon bag through mass flow
controllers with flow rate of 25 L min$^{-1}$ and 100 mL min$^{-1}$, respectively, while excess
$H_2O_2$ was injected through a three-way valve using a micro syringe. Initial conditions
of the different species introduced into the reactor for each experiment are outlined in
Table S1 in the Supplementary Material. By varying the presence of $H_2O_2$, turning
on/off the light, a series of observations were generated, such as $N_2$ + NMHCs + dark
reaction, $N_2$ + NMHCs + hv (254 nm), and $N_2$ + NMHCs + $H_2O_2$ + dark reaction.
2.1.2 Gas sampling and analysis
NMHCs Analyzer (GC-FID) with a time resolution of 1 hour independently
developed by the Research Center for Eco-Environmental Sciences (RCEES) was used
to analyze 25 C3-C11 alkanes. The sample gas was enriched by a 60-80 mesh
Carbopack B adsorption tube under the condition of 183.15 K, and then the adsorption
tube was rapidly heated to 453.15 K. The 25 alkanes were detected by FID at 523.15 K
after programmed heating at253.15 K, 303.15 K and 433.15 K in 30 min (Liu et al.,

2016).

Figure S1(a) reveals that the mixed gas diluted with $N_2$ underwent a 14-hour reaction
in a Teflon reactor without light. The $K_d$ values ranged from 0.00013 to 0.00048 ppbv/h,
implying negligible influence from factors such as alkane loss from reactor walls, self-
consumption, or airbag leakage. Figure S1(b) illustrates that the peak height variation
for 25 alkanes + 50 μl of $H_2O_2$ within 15 hours was less than 3%, indicating the
insignificance of dark reactions between $H_2O_2$ and alkanes. When the same
concentration mixed gas was irradiated for 7 hours without $H_2O_2$, alkane concentration





changes were depicted in Fig. S2. The results indicated that minimal impact from alkane
photolysis on OH radical reaction rate constant determination.

To obtain the reaction rate constants of alkanes with OH radicals in 1-2 hour, the

alkanes mixture exiting the reactor was collected in more than ten polyvinyl fluoride
(PVF) sampling bag (1.0 L) using a transparent vacuum sampling device for GC-FID.
Prior to use, the empty sampling bag was flushed with high-purity nitrogen 3 times and
placed within the vacuum sampler - a system utilizing an oil-free diaphragm air pump
to create a vacuum. The initial concentrations of alkanes sample were collected before
the lamp on, and the following sampling process occurred every 10 minutes. Collected
samples were subsequently analyzed using a self-develop automated injection system
for PVF bag.
2.1.3 Relative rate technique

The rate coefficients were measured by the relative rate method (Atkinson, 1986).

The basic principle is that the rate constant for the reaction of the reactant used as a
reference with OH radicals is known, rate constant for the reaction of OH radicals with
the target compound can be determined by monitoring the simultaneous decay of the
target and reference compounds in the presence of OH radicals due to the competitive
response mechanism. To ensure that the reactants only react with OH radicals, the OH
radicals need to be in excess in the experiment. The research method of this work is
improved and expanded based on the multivariate relative rate method published by
Shaw et al. (Shaw et al., 2018), taking the mixed system as the research object,
broadening the range of compounds that can be examined.

Taking R (reference compounds) and X (target compounds) as examples, the

reaction of OH radicals can be described as follows:

$R+OH \rightarrow Products\ (k_R)$                    (R1)

$X+OH \rightarrow Products\ (k_X)$                    (R2)

$-\dfrac{d[R]}{dt}=k_R[OH][R]$                    (R3)

$-\dfrac{d[X]}{dt}=k_X[OH][X]$                    (R4)





$$\ln\left(\frac{[R]_0}{[R]_t}\right) = k_R \cdot \int [OH]dt \qquad (R5)$$

$$\ln\left(\frac{[X]_0}{[X]_t}\right) = k_X \cdot \int [OH]dt \qquad (R6)$$

$$\ln\left(\frac{[X]_0}{[X]_t}\right) = \frac{k_X}{k_R} \cdot \ln\left(\frac{[R]_0}{[R]_t}\right) \qquad (R7)$$

Where $[R]_0$ and $[X]_0$ are the concentrations of reference compounds and target
compounds before turning on the light; $[R]_t$ and $[X]_t$ are the corresponding
concentrations after turning on the light for time t. $k_R$ and $k_X$ refer to the second-order
rate constants for the reaction of the reference compounds and target compounds with
OH radicals.
2.1.4 Choice of reference k values
It is critical to choose appropriate reference compounds in a kinetics study using
the relative rate technique. Some reported values of the rate constants for reactions of
C3-C11 alkanes with OH radicals have been measured by different methods in different
laboratories, and these measurement results may be quite different. When these rate
constants are measured by the relative rate technique, choosing different reference
values will lead to a change of the final experimental target rate constants. In this work,
selecting 3 different commonly used reference compounds (n-Hexane, Cyclohexane, n-
Octane) to determine the rate constants for each reaction at room temperature to check
the consistency of kinetic results. The selection of k values for reference compounds
and literature comparison comes from several data sets in the NIST chemical kinetics
database (https://kinetics.nist.gov/kinetics/). Among them, at $298 \pm 1$ K, the k values
(in units of $cm^3 \cdot molecule^{-1} \cdot s^{-1}$) of the three reference compounds selected respectively
are $k_{OH+n-Hexane}=5.20 \times 10^{-12}$, which is derived from Atkinson et al (Atkinson and Arey,
2003), updated data evaluation value; $k_{OH+Cyclohexane}=7.14 \times 10^{-12}$, $k_{OH+n-Octane}=8.48 \times 10^{-12}$
$^{-12}$, and the selection of k values of cyclohexane and n-octane is most consistent with the
rate constant of cyclohexane and octane obtained by using n-hexane as reference.





However, the value of the reference compound at different temperatures (273-323 K)
is different than the room temperature. A detailed explanation is reflected in Sec. 3.3.
2.1.5 Materials
The air bath gas was obtained by a mix of nitrogen (200 L) and oxygen (50L).
$H_2O_2$ (30%) was provided by Sinopharm Chemical Reagent Co., Ltd. The standard gas
(PAMs) is a mixed standard sample of 57 kinds of NMHCs produced by Linde Spectra
Environmental Gases (Alpha, NJ). Sampling bag (PVF, 1 L) was provided by Dalian
Delin Gas Packing Co., Ltd. The pump is the NMP830 KNDC model produced by KNF,
Germany, with a maximum air sampling rate of 23 L/min. The climate-controlled box
(ZRG-1000D-C0203) is provided by Shanghai Proline Electronic Technology Co., Ltd.
**2.2 Estimation of the rate constant at 298 K (SAR)**
In the past few decades, researchers have been devoted to finding a reasonable
theoretical estimation method for the kinetic rate constants (Cohen, 1991). Structure-
Activity Relationship (SAR) established and developed by Kwok and Atkinson et al.
(Kwok and Atkinson, 1995), is the most widely used estimation method of rate
constants. Based on the relationship between the structure and the reaction activity of
the compounds, this method assumes that the hydrogen extraction reaction mainly
occurs in the saturated compounds and the addition reaction mainly occurs in the
unsaturated compounds, which is used to estimate the gaseous rate constants for the
reactions of most VOCs with OH radicals. An advantage of the rate constant estimation
is that it gives a measure of the rates of attack at different sites in the molecule, which
is then useful in predicting the overall temperature dependence. The rate constant
estimated by SAR method is in good agreement with the experimental data. The general
error is 2σ. In this relationship, the calculation of the rate constant of the hydrogen atom
on the C-H bond is based on the evaluation of the rate constant of the $-CH_3$, $-CH_2$-,
>CH- group. The relationship between the group structure and the rate constant is as
follows:



$$K(CH_3\text{-}X)=K^0_{prim}F(X)$$
$$K(X\text{-}CH_2\text{-}Y)=K^0_{sec}F(X)F(Y)$$
$$K(X\text{-}CH(Y)Z)=K^0_{tert}F(X)F(Y)F(Z)$$
$$K_{tot} = \sum [K(CH_3\text{-}X)+K(X\text{-}CH_2\text{-}Y)+K(X\text{-}CH(Y)Z)]$$
Where, $K_{tot}$ represents the rate constant of each target compound. $K^0_{prim}$, $K^0_{sec}$,
$K^0_{tert}$ represent the rate constants of each $-CH_3$, $-CH_2-$ and $>CH-$. For standard
substituent groups such as $-CH_3$, $F(-CH_3)=1.00$, X, Y and Z represent substituent
groups, $F(X)$, $F(Y)$ and $F(Z)$ refer to the activity coefficient of substituents (X, Y, Z)
at different positions on carbon groups. At room temperature, $F(-CH_2-)=1.23$,
$F(>CH-)=1.23$. Additionally, Wilson et al. (Wilson et al., 2006) conducted extensive
experiments to obtained the new fundamental rate constants for different positional
groups based on the method of Atkinson and Kwok et al.

## 3. Result and Discussion

### 3.1 Results from relative rate experiments with different bath gases at 298 K

The rate constants for the reactions involving OH with C3-C11 alkanes in the
mixed system with three different bath gas environments ($N_2$, Air, $O_2$) were determined
at 298±1 K. The concentration curves of target alkanes and the reference compound (n-
Hexane) were plotted in Fig. 2. As shown in Fig. 2, the decay of both target and
reference compounds correlated well with eq. (7), and high correlation coefficients ($R^2$)
were observed for most alkanes, exceeding 0.99. Table 1 listed the obtained $K_{OH}$ for
C3-C11 alkanes under three bath gases using the related reference compounds. The
error strip (σ) in Table 1 accounted for data fitting dispersion, reference rate constant
uncertainty, and experimental parameter uncertainties (pressure, temperature, flow rate,
reactant concentration). The results indicated strong agreement (within <15%) between
rate constants for 25 C3-C11 straight-chain, branched-chain, and cycloalkanes, using





different reference compounds. For example, the $K_{OH}$ obtained for propane with n-hexane, cyclohexane and n-octane as the reference compound were $(1.45\pm0.01)\times10^{-12}$, $(1.34\pm0.03)\times10^{-12}$ and $(1.47\pm0.17)\times10^{-12}$, respectively (within 10%). This suggests that reference compound variation minimally affects results, indicating reliable experimental methods and data. Notably, the rate constant for 3-Methylheptane's reaction with OH radicals at room temperature was determined for the first time. As shown in Fig. 3, for the different bath gases, the obtained $K_{OH}$ for C3-C11 alkanes showed high agreement. Additionally, it can be clearly seen in the figure that the reactivity of linear alkanes ($RCH_2R$) with OH radicals increasing as the number of carbon atoms in the hydrocarbon molecules increases, indicating that the increase of R-terminal alkyl chain length will provide additional hydrogen extraction sites.

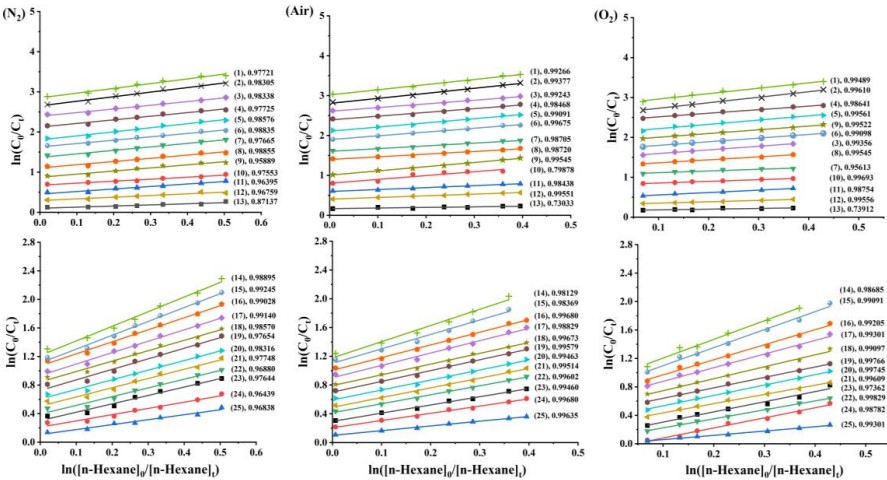

Figure 2. Typical kinetic data as acquired with the multivariate relative rate technique at 298 K and a fixed reaction time of 70 min for the reaction of C3-C11 alkanes with the OH radical using n-hexane as reference compound in different bath gases ($N_2$, Air, $O_2$). The numbers in parentheses correspond to each substance, followed by the correlation coefficient $R^2$. The following data have been displaced for reasons of clarity: ($N_2$): (1) Methylcyclopentane, (2) Cyclohexane, (3) Cyclopentane, (4) 2-Methylpentane, (5) 2,3-Dimethylbutane, (6) 2,4-Dimethylpentane, (7) Isopentane, (8) 1-pentane, (9) 3-Methylpentane, (10) Isobutane, (11) n-Butane, (12) 2,2-Dimethylbutane, (13) Propane (14) n-Undecane, (15) n-Decane, (16) Nonane, (17)





Methylcyclohexane, (18) n-Octane, (19) 3-Methylheptane, (20) 2-Methylheptane, (21)
2,3,4-Trimethylpentane, (22) 1-Heptane, (23) 2-Methylhexane, (24) 3-Methylhexane,
(25) 2,2,4-Trimethylpentane vertically displaced by 2.8, 2.6, 2.4, 2.1, 1.8, 1.6, 1.4, 1.1,
0.9, 0.7, 0.5, 0.3, 0.1, 1.2, 1.1, 1, 0.9, 0.8, 0.7, 0.6, 0.5, 0.4, 0.3, 0.25, 0.1 units,
respectively; (Air) Each alkane (in the above order) vertically displaced by 3, 2.8, 2.6,

268      2.4, 2.1, 1.9, 1.6, 1.4, 1, 0.8, 0.6, 0.4, 0.1, 1.2, 1.1, 1, 0.9, 0.8, 0.7, 0.6, 0.5, 0.4, 0.3, 0.2,

0.1 units, respectively; ($O_2$) Each alkane (in the above order) vertically displaced by

270      2,8, 2.6, 1.5, 2.4, 2,1, 1.7, 1, 1.3, 1.9, 0.8, 0.5, 0.3, 0.1, 1, 0.9, 0.8, 0.7, 0.6, 0.5, 0.4, 0.3,

0.1 units, respectively (Not mentioned defaults to 0).

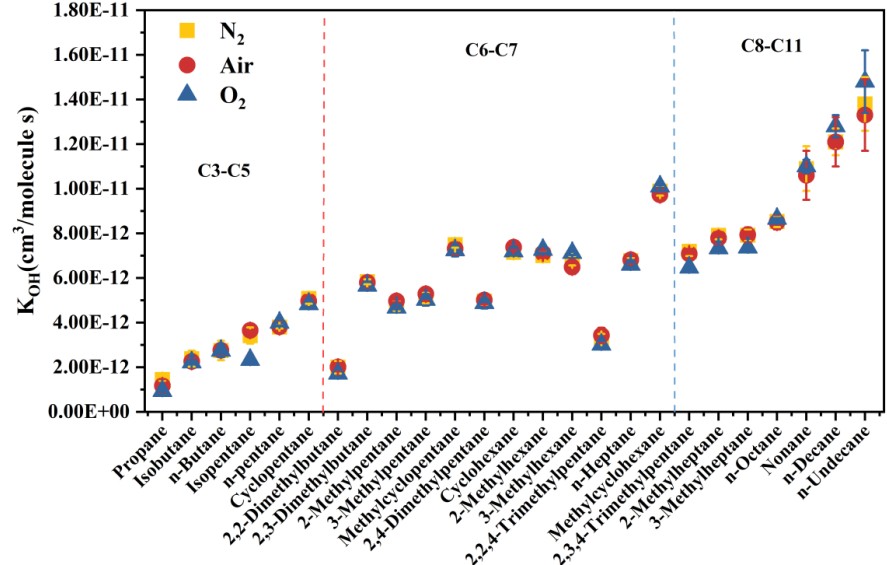


Figure 3. Rate constants of C3-C11 alkanes in different bath gases ($N_2$, Air, $O_2$)

at 298±1 K. The error bar was taken as σ.

The obtained $K_{OH}$ values for C3-C11 alkanes were compared with literature-
reported values (Table 1). For several n-alkanes, such as n-butane, the average rate
constants obtained in different bath gases are (2.75±0.43), (2.76±0.27), (2.74±0.29),
respectively, the unit is $10^{-12}$ $cm^3 \cdot molecule^{-1} \cdot s^{-1}$. The result is highly consistent with the
value (2.72±0.27) obtained by Perry et al using flash photolysis resonance fluorescence
technique in an Ar system, with a consistency of 2% or better (Perry et al., 1976).
Although slightly higher by 6% and 10% compared to the values obtained by Greiner



(Greiner, 1970a) and Talukdar et al. (Talukdar et al., 1994) using absolute techniques
in inert gases (2.56±0.25, 2.46±0.15), when considering the errors, they still exhibit
consistency within a certain range. Compared to the value obtained by DeMore et al.
(Demore and Bayes, 1999) using the relative rate method (2.36±0.25), these values are
higher by 13%, it is considered that be caused by experimental error.
**n-pentane (n-Heptane).** As in the n-butane case, the derived $K_{OH}$ for these compounds
is excellent consistent (within 3%) in different bath gases. The derived rate constants
for n-pentane and n-heptane are in excellent agreement (4% or better at 298 K) with
previous studies (Donahue et al., 1998; Atkinson, 2003; Atkinson and Arey, 2003;
Wilson et al., 2006; Crawford et al., 2011; Calvert et al., 2015; Morin et al., 2015).
**n-Octane (Nonane).** There is little difference in the rate constants of n-Octane and
Nonane in 3 bath gases, within a consistency of 2% or better. The reaction rate constants
of n-Octane and OH radicals are in extremely good agreement with the values reported
in the literature (within 5%). Same for Nonane, consistency with previous studies is
less than 8% (Greiner, 1970a; Atkinson et al., 1982; Ferrari et al., 1996; Atkinson and
Arey, 2003; Li et al., 2006).
**n-Decane.** The obtained average $K_{OH}$ for n-decane in $N_2$/Air/$O_2$ systems were
(1.21±0.06), (1.21±0.11) and (1.28±0.05), respectively, the unit is $10^{-11}$ $cm^3 \cdot molecule^{-1} \cdot s^{-1}$.
$^{-1} \cdot s^{-1}$. When considering experimental error, these results are consistent with the relative
value (1.29±0.10) obtained by Li et al. (Li et al., 2006) in a He system and the reviewed
value (1.10) of Atkinson and Arey (Atkinson and Arey, 2003), with about a consistency
of 6%-9%.
**n-Undecane.** The measured average $K_{OH}$ for n-decane in 3 bath gas systems were
(1.38±0.05), (1.33±0.16) and (1.48±0.14), respectively, the unit is $10^{-11}$ $cm^3 \cdot molecule^{-1}$
$^{-1} \cdot s^{-1}$. The data in the oxygen system is about 11% higher than that in the air system. It
is about 8% higher than the previous research (Atkinson and Arey, 2003;
Sivaramakrishnan and Michael, 2009; Calvert et al., 2015).
For the cycloalkanes, like cyclopentane, the average rate constants are 5.08±0.24,
4.96±0.27, 4.82±0.14, respectively, the unit is $10^{-12}$ $cm^3 \cdot molecule^{-1} \cdot s^{-1}$. The results are
in excellent agreement (5% or better) with the reviewed value (4.97) of Atkinson and



Arey (Atkinson and Arey, 2003) and the relative values (4.83, 4.84) of DeMore et al.
(Demore and Bayes, 1999) and Singh et al. (Singh et al., 2013) and the absolute value
(5.02) of Droege et al. (Droege and Tully, 1987). And the obtained $K_{OH}$ values for
cyclohexane are highly consistent (3% or better) with the absolute values ($7.14×10^{-12}$,
$7.19×10^{-12}$) obtained by Droege and Tully and Sprengnether et al. (Droege and Tully,
1987; Sprengnether et al., 2009). However, this result is slightly higher than the relative
value by about 5%-16%. Like the relative values measured by DeMore and Bayes
(Demore and Bayes, 1999) or Wilson et al. (Wilson et al., 2006) were $6.70×10^{-12}$ and
$6.38×10^{-12}$, respectively. It worth noting that the $K_{OH}$ value for methylcyclopentane in
this work is highly consistent (within 3% to 5%) with the absolute data reported by
Sprengnether et al. (Sprengnether et al., 2009). However, it is lower by approximately
15% to 18% compared to the relative data obtained by Anderson et al. (Andersen et al.,
2003). The $K_{OH}$ values for methylcyclohexane are excellent agreement (3% or better)
with other values reported by Atkinson and Arey (Atkinson and Arey, 2003) and Calvert
et al. (Calvert et al., 2015).
Furthermore, for several less studied branched alkanes, such as 2-Methylhexane,
3-Methylhexane, and 2-Methylheptane, there is only one study reported so far.
Sprengnether et al. (Sprengnether et al., 2009) conducted a study on 2-Methylhexane
and 3-Methylhexane and obtained $K_{OH}$ values at room temperature for the first time,
which were $6.30×10^{-12}$ and $6.69×10^{-12}$, respectively. Our results are about slightly
higher by approximately 3% to 6% compared to their values. However, the data for 2-
Methylheptane in this work is lower by about 17% compared to the value reported by
Shaw et al. (Shaw et al., 2018).





Table 1. Comparison of Experimental in this work with the reported in the literature
and Estimated Alkane Rate Constants Based on the Present SAR Calculations in the
different bath gases ($N_2$, Air, $O_2$) at 298±1 K.

| Alkanes | Bath gas | Reference | This work | | | Reference | SAR |
|---|---|---|---|---|---|---|---|
| | | | $K_{OH}/K_{reference}$ ±1σ | $K_{OH}$ ±1σ (×$10^{-12}$ cm³ molecule⁻¹ s⁻¹) | $K_{OH-av}$[a] ±1σ (×$10^{-12}$ cm³ molecule⁻¹ s⁻¹) | $K_{OH}$ (×$10^{-12}$cm³ molecule⁻¹ s⁻¹) | $K_{OH}$ (×$10^{-12}$cm³ molecule⁻¹ s⁻¹) |
| Propane | $N_2$ | n-Hexane | 0.278±0.001 | (1.45±0.01) | | | |
| | | Cyclohexane | 0.187±0.004 | (1.34±0.03) | (1.45±0.05) | | |
| | | n-Octane | 0.174±0.020 | (1.47±0.17) | | | |
| | Air | n-Hexane | 0.190±0.033 | (1.66±0.50) | | 1.11[bcd] 1.09[e] 1.91[f] (1.15±0.15)[g] | 1.27 |
| | | Cyclohexane | 0.200±0.070 | (1.10±0.20) | (1.17±0.30) | | |
| | | n-Octane | 0.172±0.057 | (1.16±0.26) | | | |
| | $O_2$ | n-Hexane | 0.178±0.002 | (0.927±0.012) | | | |
| | | Cyclohexane | 0.133±0.004 | (0.960±0.026) | (0.933±0.027) | | |
| | | n-Octane | 0.109±0.008 | (0.925±0.065) | | | |
| Isobutane | $N_2$ | n-Hexane | 0.451±0.052 | (2.35±0.27) | | | |
| | | Cyclohexane | 0.392±0.038 | (2.47±0.53) | (2.38±0.36) | | |
| | | n-Octane | 0.282±0.053 | (2.39±0.45) | | | |
| | Air | n-Hexane | 0.451±0.052 | (2.31±0.06) | | 2.12[h] 2.22[i] (2.34±0.33)[j] | 2.44 |
| | | Cyclohexane | 0.315±0.008 | (2.27±0.06) | (2.26±0.13) | | |
| | | n-Octane | 0.282±0.053 | (2.24±0.04) | | | |
| | $O_2$ | n-Hexane | 0.422±0.004 | (2.19±0.02) | | | |
| | | Cyclohexane | 0.312±0.002 | (2.24±0.02) | (2.22±0.09) | | |
| | | n-Octane | 0.262±0.006 | (2.22±0.05) | | | |
| n-Butane | $N_2$ | n-Hexane | 0.511±0.071 | (2.65±0.37) | | (2.36±0.25)[b] (2.72±0.27)[k] (2.56±0.25)[m] (2.46±0.15)[d] | 2.63 |
| | | Cyclohexane | 0.423±0.120 | (3.02±0.85) | (2.75±0.44) | | |
| | | n-Octane | 0.343±0.084 | (2.91±0.71) | | | |
| | Air | n-Hexane | 0.516±0.025 | (2.68±0.13) | | | |
| | | Cyclohexane | 0.418±0.038 | (3.01±0.27) | (2.76±0.27) | | |
| | | n-Octane | 0.345±0.042 | (2.93±0.36) | | | |





| Compound | Gas | Solvent | | | | | |
|---|---|---|---|---|---|---|---|
| | | n-Hexane | 0.517±0.032 | (2.69±0.17) | | | |
| | $O_2$ | Cyclohexane | 0.396±0.039 | (2.85±0.28) | (2.74±0.29) | | |
| | | n-Octane | 0.333±0.044 | (2.82±0.37) | | | |
| | | n-Hexane | 0.715±0.038 | (3.72±0.31) | | | |
| | $N_2$ | Cyclohexane | 0.434±0.061 | (3.12±0.44) | (3.42±0.36) | | |
| | | n-Octane | 0.363±0.054 | (3.08±0.46) | | | |
| | | n-Hexane | 0.684±0.033 | (3.56±0.17) | | 3.60 [e] | |
| Isopentane | Air | Cyclohexane | 0.512±0.026 | (3.66±0.19) | (3.65±0.25) | 3.65 [h] | 4.04 |
| | | n-Octane | 0.442±0.025 | (3.75±0.22) | | 3.50 [f] | |
| | | n-Hexane | 0.446±0.020 | (2.32±0.10) | | | |
| | $O_2$ | Cyclohexane | 0.330±0.012 | (2.38±0.09) | (2.33±0.07) | | |
| | | n-Octane | 0.275±0.001 | (2.32±0.01) | | | |
| | | n-Hexane | 0.777±0.036 | (4.04±0.19) | | | |
| | $N_2$ | Cyclohexane | 0.533±0.006 | (3.83±0.04) | (3.80±0.07) | | |
| | | n-Octane | 0.448±0.001 | (3.80±0.01) | | 3.80 [e] | |
| | | n-Hexane | 0.730±0.057 | (3.79±0.29) | | 3.98 [n] | |
| n-pentane | Air | Cyclohexane | 0.527±0.021 | (3.79±0.15) | (3.81±0.27) | 4.03 [o] | 4.05 |
| | | n-Octane | 0.454±0.029 | (3.85±0.24) | | (3.97±0.20) [p] | |
| | | | | | | (4.20±0.15) [g] | |
| | | n-Hexane | 0.754±0.011 | (3.92±0.06) | | | |
| | $O_2$ | Cyclohexane | 0.558±0.005 | (4.01±0.04) | (3.99±0.13) | | |
| | | n-Octane | 0.467±0.012 | (3.96±0.10) | | | |
| | | n-Hexane | 0.976±0.051 | (5.08±0.26) | | | |
| | $N_2$ | Cyclohexane | 0.702±0.019 | (5.05±0.14) | (5.08±0.24) | | |
| | | n-Octane | 0.605±0.019 | (5.13±0.16) | | 4.97 [e] | |
| | | n-Hexane | | (4.94±0.17) | | 4.83 [b] | |
| Cyclopentane | Air | Cyclohexane | 0.951±0.033 0.674±0.040 | (4.85±0.29) | (4.96±0.27) | 5.02 [q] | 7.07 |
| | | n-Octane | | (5.09±0.24) | | (4.90±0.20) [p] | |
| | | | | | | 4.84 [b r] | |
| | | n-Hexane | 0.924±0.007 | (4.80±0.04) | | | |
| | $O_2$ | Cyclohexane | 0.673±0.010 | (4.84±0.07) | (4.82±0.14) | | |
| | | n-Octane | 0.576±0.014 | (4.89±0.12) | | | |
| | | n-Hexane | 0.382±0.027 | (1.98±0.14) | | | |
| | $N_2$ | Cyclohexane | 0.292±0.055 | (2.10±0.39) | (2.00±0.28) | | |
| 2,2-Dimethylbutane | | n-Octane | 0.237±0.035 | (2.01±0.30) | | (2.23±0.15) [p] | |
| | | n-Hexane | 0.409±0.019 | (2.13±0.10) | | 2.15 [s] | 1.82 |
| | Air | Cyclohexane | 0.301±0.030 | (2.17±0.22) | (2.01±0.14) | 2.32 [o] | |
| | | n-Octane | 0.264±0.031 | (2.00±0.03) | | | |





| Compound | Gas | Solvent | | | | Literature | Final |
|---|---|---|---|---|---|---|---|
| | O₂ | n-Hexane | 0.327±0.015 | (1.70±0.08) | | | |
| | | Cyclohexane | 0.238±0.016 | (1.71±0.11) | (1.71±0.19) | | |
| | | n-Octane | 0.204±0.015 | (1.73±0.13) | | | |
| | N₂ | n-Hexane | 1.092±0.064 | (5.68±0.33) | | | |
| | | Cyclohexane | 0.815±0.005 | (5.86±0.03) | (5.83±0.11) | | |
| | | n-Octane | 0.687±0.002 | (5.83±0.02) | | | |
| 2,3-Dimethylbutane | Air | n-Hexane | 1.095±0.061 | (5.69±0.32) | | 5.78 [e] | |
| | | Cyclohexane | 0.798±0.035 | (5.74±0.25) | (5.80±0.27) | (6.14±0.25) [p] | 4.55 |
| | | n-Octane | 0.690±0.019 | (5.85±0.16) | | 6.03 [h] | |
| | O₂ | n-Hexane | 1.093±0.018 | (5.68±0.09) | | | |
| | | Cyclohexane | 0.786±0.008 | (5.65±0.06) | (5.65±0.17) | | |
| | | n-Octane | 0.650±0.01 | (5.52±0.17) | | | |
| | N₂ | n-Hexane | 0.913±0.017 | (4.75±0.09) | | | |
| | | Cyclohexane | 0.662±0.035 | (4.76±0.25) | (4.75±0.22) | | |
| | | n-Octane | 0.557±0.024 | (4.72±0.20) | | | |
| 2-Methylpentane | Air | n-Hexane | 0.972±0.022 | (5.06±0.11) | | 5.2 [e] | |
| | | Cyclohexane | 0.660±0.004 | (4.74±0.03) | (4.97±0.06) | (5.25±0.25) [p] | 5.45 |
| | | n-Octane | 0.586±0.001 | (4.97±0.01) | | 5.00 [f]  4.75 [s] | |
| | O₂ | n-Hexane | 0.899±0.001 | (4.67±0.01) | | | |
| | | Cyclohexane | 0.646±0.003 | (4.65±0.02) | (4.67±0.07) | | |
| | | n-Octane | 0.535±0.007 | (4.54±0.06) | | | |
| | N₂ | n-Hexane | 1.000±0.035 | (5.20±0.18) | | | |
| | | Cyclohexane | 0.707±0.015 | (5.08±0.11) | (5.10±0.23) | | |
| | | n-Octane | 0.913±0.017 | (4.92±0.29) | | | |
| 3-Methylpentane | Air | n-Hexane | 1.014±0.030 | (5.27±0.16) | | 5.20 [e] | |
| | | Cyclohexane | 0.762±0.051 | (5.37±0.41) | (5.28±0.31) | (5.54±0.25) [p] | 5.73 |
| | | n-Octane | 0.617±0.065 | (5.23±0.55) | | 4.93 [s] | |
| | O₂ | n-Hexane | 0.973±0.039 | (5.06±0.21) | | | |
| | | Cyclohexane | 0.701±0.025 | (5.04±0.18) | (5.02±0.26) | | |
| | | n-Octane | 0.582±0.028 | (4.94±0.24) | | | |
| methylcyclopentane | N₂ | n-Hexane | 1.455±0.044 | (7.56±0.23) | | | |
| | | Cyclohexane | 0.957±0.004 | (7.50±0.03) | (7.49±0.13) | | |
| | | n-Octane | 0.881±0.005 | (7.47±0.04) | | (7.65±0.10) [u] | |
| | Air | n-Hexane | 1.432±0.053 | (7.45±0.28) | | (8.60±0.30) [p] | 8.75 |
| | | Cyclohexane | 1.007±0.023 | (7.24±0.16) | (7.31±0.29) | (8.60±2.20) [t] | |
| | | n-Octane | 0.876±0.049 | (7.43±0.41) | | | |



| Compound | Atmosphere | Solvent | | | | Literature | Final |
|---|---|---|---|---|---|---|---|
| | $O_2$ | n-Hexane | 1.404±0.046 | (7.30±0.24) | | | |
| | | Cyclohexane | 1.044±0.004 | (7.26±0.20) | (7.24±0.28) | | |
| | | n-Octane | 0.881±0.005 | (7.10±0.31) | | | |
| | $N_2$ | n-Hexane | 0.967±0.025 | (5.03±0.13) | | | |
| | | Cyclohexane | 0.706±0.017 | (5.07±0.13) | (4.96±0.17) | | |
| | | n-Octane | 0.580±0.007 | (4.92±0.06) | | | |
| 2,4-Dimethylpentane | Air | n-Hexane | 0.962±0.012 | (5.00±0.06) | | 4.80 [e] | |
| | | Cyclohexane | 0.708±0.042 | (5.09±0.30) | (5.01±0.20) | 5.51 [s] | 5.02 |
| | | n-Octane | 0.596±0.026 | (5.05±0.22) | | (5.76±0.40) [p] | |
| | $O_2$ | n-Hexane | 0.944±0.032 | (4.91±0.17) | | | |
| | | Cyclohexane | 0.706±0.017 | (5.07±0.13) | (4.87±0.24) | | |
| | | n-Octane | 0.564±0.026 | (4.79±0.22) | | | |
| | $N_2$ | n-Hexane | 1.392±0.102 | (7.24±0.26) | | | |
| | | Cyclohexane | -- | -- | (7.15±0.23) | | |
| | | n-Octane | 0.842±0.008 | (7.14±0.07) | | 6.97 [e] | |
| Cyclohexane | Air | n-Hexane | 1.410±0.009 | (7.33±0.05) | | 7.14 [q] | |
| | | Cyclohexane | -- | -- | (7.38±0.13) | 6.38 [h] | 8.48 |
| | | n-Octane | 0.872±0.022 | (7.39±0.19) | | 6.70 [b] | |
| | | | | | | (7.19±0.10) [u] | |
| | $O_2$ | n-Hexane | 1.401±0.017 | (7.22±0.05) | | (6.85±0.20) [p] | |
| | | Cyclohexane | -- | -- | (7.19±0.19) | | |
| | | n-Octane | 0.830±0.013 | (7.04±0.11) | | | |
| | $N_2$ | n-Hexane | 1.366±0.055 | (7.10±0.29) | | | |
| | | Cyclohexane | 0.996±0.011 | (7.16±0.17) | (7.01±0.22) | | |
| | | n-Octane | 0.820±0.011 | (6.95±0.09) | | | |
| 2-Methylhexane | Air | n-Hexane | 1.369±0.004 | (7.12±0.02) | | | |
| | | Cyclohexane | 0.986±0.032 | (7.04±0.23) | (7.11±0.13) | (6.69±0.10) [u] | 6.86 |
| | | n-Octane | 0.820±0.025 | (6.95±0.13) | | | |
| | $O_2$ | n-Hexane | 1.415±0.015 | (7.36±0.08) | | | |
| | | Cyclohexane | 1.020±0.022 | (7.34±0.15) | (7.26±0.16) | | |
| | | n-Octane | 0.852±0.006 | (7.22±0.05) | | | |
| | $N_2$ | n-Hexane | 1.310±0.022 | (6.81±0.11) | | | |
| | | Cyclohexane | 0.938±0.023 | (6.74±0.16) | (6.77±0.21) | | |
| 3-Methylhexane | | n-Octane | 0.794±0.015 | (6.73±0.13) | | (6.30±0.10) [u] | 7.15 |
| | Air | n-Hexane | 1.248±0.025 | (6.49±0.13) | | | |
| | | Cyclohexane | 0.892±0.098 | (6.41±0.71) | (6.49±0.31) | | |
| | | n-Octane | 0.807±0.122 | (6.84±1.03) | | | |



| Compound | Gas | Solvent | | | | Literature | Final |
|---|---|---|---|---|---|---|---|
| | O₂ | n-Hexane | 1.401±0.017 | (7.28±0.09) | | | |
| | | Cyclohexane | 1.007±0.019 | (7.24±0.14) | (7.12±0.10) | | |
| | | n-Octane | 0.840±0.002 | (7.12±0.02) | | | |
| | N₂ | n-Hexane | 0.655±0.030 | (3.41±0.15) | | | |
| | | Cyclohexane | 0.458±0.026 | (3.29±0.18) | (3.30±0.19) | | |
| | | n-Octane | 0.384±0.018 | (3.26±0.15) | | | |
| 2,2,4-Trimethyl pentane | Air | n-Hexane | 0.674±0.057 | (3.50±0.30) | | 3.34 [e] | |
| | | Cyclohexane | 0.471±0.051 | (3.38±0.36) | (3.43±0.34) | 3.64 [s] | 4.64 |
| | | n-Octane | 0.396±0.043 | (3.36±0.37) | | (3.34±0.25) [p] | |
| | | | | | | (3.71±0.10) [v] | |
| | O₂ | n-Hexane | 0.587±0.019 | (3.05±0.10) | | | |
| | | Cyclohexane | 0.421±0.018 | (3.03±0.13) | (3.01±0.17) | | |
| | | n-Octane | 0.352±0.008 | (2.98±0.07) | | | |
| | N₂ | n-Hexane | 1.302±0.004 | (6.77±0.02) | | | |
| | | Cyclohexane | 0.937±0.029 | (6.74±0.21) | (6.77±0.13) | | |
| | | n-Octane | 0.789±0.017 | (6.69±0.14) | | | |
| n-Heptane | Air | n-Hexane | 1.280±0.066 | (6.66±0.34) | | 6.76 [e] | |
| | | Cyclohexane | 0.941±0.021 | (6.77±0.15) | (6.81±0.17) | 6.68 [y] | 6.87 |
| | | n-Octane | 0.804±0.005 | (6.81±0.04) | | 6.80 [h] | |
| | | | | | | (6.70±0.15) [g] | |
| | O₂ | n-Hexane | 1.271±0.004 | (6.61±0.02) | | | |
| | | Cyclohexane | 0.912±0.004 | (6.56±0.03) | (6.59±0.11) | | |
| | | n-Octane | 0.760±0.012 | (6.45±0.10) | | | |
| | N₂ | n-Hexane | 1.914±0.070 | (9.95±0.37) | | | |
| | | Cyclohexane | 1.381±0.010 | (9.93±0.07) | (9.89±0.20) | | |
| | | n-Octane | 0.789±0.017 | (9.80±0.11) | | 9.60 [e] | |
| Methylcyc lohexane | Air | n-Hexane | 1.906±0.098 | (9.91±0.51) | | (9.64±0.30) [p] | |
| | | Cyclohexane | 1.349±0.012 | (9.70±0.09) | (9.73±0.25) | (11.8±1.00) [F] | 10.20 |
| | | n-Octane | 1.190±0.042 | (10.10±0.40) | | (9.50±0.14) [D] | |
| | O₂ | n-Hexane | 1.944±0.025 | (10.10±0.20) | | (9.29±0.10) [u] | |
| | | Cyclohexane | 1.400±0.007 | (10.10±0.50) | (10.10±0.60) | | |
| | | n-Octane | 1.165±0.023 | (9.88±0.20) | | | |
| | N₂ | n-Hexane | 1.383±0.013 | (7.19±0.07) | | | |
| | | Cyclohexane | 0.997±0.043 | (7.17±0.31) | (7.19±0.21) | | |
| 2,3,4-Trimethyl pentane | | n-Octane | 0.839±0.028 | (7.16±0.24) | | 6.60 [e] | |
| | Air | n-Hexane | 1.381±0.021 | (7.18±0.11) | | 6.50 [h] | 8.54 |
| | | Cyclohexane | 0.968±0.037 | (6.96±0.26) | (7.08±0.22) | (6.60±0.26) [p] | |
| | | n-Octane | 0.823±0.014 | (6.98±0.12) | | | |





| | | n-Hexane | 1.266±0.032 | (6.58±0.16) | | | |
|---|---|---|---|---|---|---|---|
| | $O_2$ | Cyclohexane | 0.908±0.031 | (6.53±0.22) | (6.46±0.21) | | |
| | | n-Octane | 0.757±0.010 | (6.42±0.09) | | | |
| | | n-Hexane | 1.521±0.009 | (7.91±0.05) | | | |
| | $N_2$ | Cyclohexane | 1.123±0.053 | (8.07±0.38) | (7.91±0.18) | | |
| | | n-Octane | 0.856±0.033 | (7.83±0.20) | | | |
| 2-Methylheptane | | n-Hexane | 1.532±0.062 | (7.97±0.32) | | | |
| | Air | Cyclohexane | 1.061±0.029 | (7.63±0.21) | (7.79±0.28) | 9.10[L] | 8.28 |
| | | n-Octane | 0.931±0.025 | (7.89±0.21) | | | |
| | | n-Hexane | 1.444±0.017 | (7.51±0.09) | | | |
| | $O_2$ | Cyclohexane | 1.037±0.021 | (7.45±0.15) | (7.33±0.04) | | |
| | | n-Octane | 0.865±0.001 | (7.33±0.01) | | | |
| | | n-Hexane | 1.525±0.022 | (7.93±0.11) | | | |
| | $N_2$ | Cyclohexane | 1.099±0.054 | (7.84±0.38) | (7.90±0.25) | | |
| | | n-Octane | 0.921±0.026 | (7.81±0.22) | | | |
| 3-Methylheptane | | n-Hexane | 1.532±0.070 | (7.97±0.37) | | | |
| | Air | Cyclohexane | 1.094±0.068 | (7.87±0.49) | (7.93±0.33) | -- | 8.90 |
| | | n-Octane | 0.935±0.270 | (7.93±0.23) | | | |
| | | n-Hexane | 1.448±0.001 | (7.53±0.10) | | | |
| | $O_2$ | Cyclohexane | 1.040±0.024 | (7.48±0.17) | (7.36±0.11) | | |
| | | n-Octane | 0.867±0.002 | (7.35±0.02) | | | |
| | | n-Hexane | 1.651±0.043 | (8.58±0.22) | | | |
| | $N_2$ | Cyclohexane | 1.186±0.012 | (8.53±0.08) | (8.53±0.25) | | |
| | | n-Octane | -- | -- | | | |
| | | n-Hexane | 1.680±0.038 | (8.74±0.20) | | 8.11[e] | |
| n-Octane | Air | Cyclohexane | 1.142±0.030 | (8.21±0.22) | (8.50±0.32) | 8.42[m] | 8.28 |
| | | n-Octane | -- | -- | | (8.48±0.10)[z] | |
| | | n-Hexane | 1.666±0.013 | (8.66±0.07) | | | |
| | $O_2$ | Cyclohexane | 1.199±0.019 | (8.62±0.14) | (8.65±0.22) | | |
| | | n-Octane | -- | -- | | | |
| | | n-Hexane | 2.124±0.057 | (11.00±0.30) | | | |
| | $N_2$ | Cyclohexane | 1.505±0.032 | (10.80±0.30) | (10.90±1.00) | 9.70[e] | |
| | | n-Octane | 1.241±0.063 | (10.50±0.50) | | 10.20[A] | |
| Nonane | | | | | | 10.70[w] | 9.70 |
| | | n-Hexane | 2.166±0.079 | (11.30±0.40) | | (11.30±1.10)[z] | |
| | Air | Cyclohexane | 1.406±0.040 | (10.10±0.30) | (10.60±1.10) | | |
| | | n-Octane | 1.263±0.046 | (10.70±0.40) | | | |



|  |  | n-Hexane | 2.117±0.002 | (11.00±0.10) |  |  |  |
| | O₂ | Cyclohexane | 1.525±0.011 | (11.00±0.10) | (11.00±0.30) |  |  |
| | | n-Octane | 1.269±0.012 | (10.80±0.10) |  |  |  |
| | | n-Hexane | 2.355±0.078 | (12.20±0.40) |  |  |  |
| | N₂ | Cyclohexane | 1.690±0.006 | (12.10±0.10) | (12.10±0.60) |  |  |
| | | n-Octane | 1.392±0.047 | (11.80±0.40) |  |  |  |
| | | n-Hexane | 2.371±0.073 | (12.30±0.40) |  | 11.00 [e] |  |
| n-Decane | Air | Cyclohexane | 1.601±0.059 | (11.50±0.40) | (12.10±1.10) | (12.9±1.00) [z] | 11.10 |
| | | n-Octane | 1.437±0.033 | (12.20±0.30) |  |  |  |
| | | n-Hexane | 2.506±0.028 | (13.00±0.20) |  |  |  |
| | O₂ | Cyclohexane | 1.804±0.034 | (13.00±0.20) | (12.80±0.50) |  |  |
| | | n-Octane | 1.503±0.004 | (12.70±0.10) |  |  |  |
| | | n-Hexane | 2.685±0.042 | (14.00±0.30) |  |  |  |
| | N₂ | Cyclohexane | 1.843±0.092 | (13.30±0.70) | (13.80±1.20) |  |  |
| | | n-Octane | 2.685±0.042 | (13.00±0.90) |  |  |  |
| | | n-Hexane | 2.594±0.251 | (13.50±1.30) |  | 12.30 [e] |  |
| n-Undecane | Air | Cyclohexane | 1.797±0.100 | (12.90±0.70) | (13.30±1.60) | 12.50 [B] | 12.50 |
| | | n-Octane | 1.588±0.076 | (13.50±0.60) |  | (11.90±2.00) [p] |  |
| | | n-Hexane | 2.805±0.179 | (14.60±0.90) |  |  |  |
| | O₂ | Cyclohexane | 2.079±0.118 | (15.00±0.80) | (14.80±1.40) |  |  |
| | | n-Octane | 1.738±0.046 | (14.70±0.40) |  |  |  |

a: Weighted average $k_{av} = (w_{ref1}k_{ref1} + w_{ref2}k_{ref2} + …) / (w_{ref1} + w_{ref2} + …)$, where
$w_{ref1} = 1/\sigma_{ref1}^2$, etc. The error, $\sigma_{av}$, was given by: $\sigma_{av} = (1/\sigma_{ref1} + 1/\sigma_{ref2} + …)^{-0.5}$.
b: (Demore and Bayes, 1999); c: (Mellouki et al., 1994); d: (Talukdar et al., 1994); e:
(Atkinson and Arey, 2003); f: (Cox et al., 1980); g: (Morin et al., 2015); h: (Wilson et
al., 2006); i: (Tully et al., 1986); j: (Edney et al., 1986); k: (Perry et al., 1976); m:
(Greiner, 1970a) ; n: (Donahue et al., 1998); o: (Harris and Kerr, 1988); p: (Calvert et
al., 2015); q: (Droege and Tully, 1987); r: (Singh et al., 2013); s: (Badra and Farooq,
2015) u: (Sprengnether et al., 2009); t: (Anderson et al., 2004); v: (Greiner, 1970b), y:
(Crawford et al., 2011) ; z: (Li et al., 2006); L: (Shaw et al., 2020); w: (Atkinson et al.,
1982); A: (Ferrari et al., 1996); B: (Sivaramakrishnan and Michael, 2009); D: (Bejan et
al., 2018); F: (Ballesteros et al., 2015).
**3.2 Comparisons to structure–activity relationships**



Based on an extensive review of kinetic literature values for linear alkanes at room
temperature, Atkinson and Kwok et al derived the values of $K_{prim}^0$, $K_{sec}^0$, $K_{tert}^0$ at room
temperature, $K_{prim}^0=0.136\times10^{-12}$ , $K_{sec}^0=0.934\times10^{-12}$ , $K_{tert}^0=1.94\times10^{-12}$ , the unit is
$cm^3\cdot molecule^{-1}\cdot s^{-1}$. Figure 4. compared rate constants for OH radical reactions with 25
alkanes across the Air system along with estimated SAR values (Atkinson, et al and
Wilson et al) at 298±1 K. The shaded area demonstrates a 20 % uncertainty in the 1:1
black gradient line. Most n-alkanes fall into the shaded area, indicating high agreement
for n-alkanes' $K_{OH}$ rate coefficients with the SAR values, especially C3-C8 n-alkanes
(about within 8%). Some longer straight-chain alkanes like Nonane, n-Decane, and n-
Undecane exhibited slightly higher $K_{OH}$ values (around 10%) compared to the
estimated SAR values, implying that longer R-terminal alkyl chains offer more
hydrogen extraction sites than SAR estimates. For branch alkanes, such as 3-
Methylheptane and 2,3,4-Trimethylpentane, the SAR values were about 12% and 20%,
respectively higher than the $K_{OH}$ values obtained in air bath gas. On the other hand, the
obtained $K_{OH}$ values for methylcyclopentane and cyclohexane were about 14% and
16%, respectively, lower than the SAR values, indicating that the reaction activity of
these cycle-chain alkanes estimated using SAR might be overestimated.
As shown in Fig. 4, the outliers are cyclopentane, 2,3-Dimethylbutane and 2,2,4-
Trimethylpentane, respectively. The obtained $K_{OH}$ values of Cyclopentane and 2,2,4-
Trimethylpentane were about 30%, 26%, respectively, lower than the corresponding
SAR values. Nevertheless, the SAR estimate for 2,3-Dimethylbutane is approximately
22% lower than the experimental value obtained in this study. Interestingly, in the
estimation by Wilson et al. (Wilson et al., 2006), it was also found that the $K_{OH}$ of this
compound (at 298 K) could not be accurately estimated by the same methodology due
to unknown reasons. This phenomenon indicates that our understanding for the
oxidation chemistry of these compounds is still limited, still need a lot of experimental
data for alkanes with this structure to confirm.



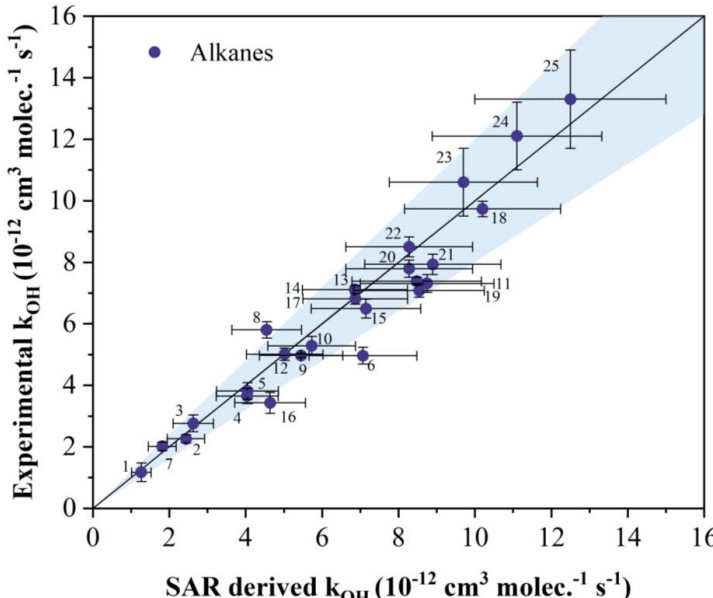

Figure 4. Measured Alkanes + OH rate coefficients plotted against SAR-derived rate coefficients for all compounds. The shaded area demonstrates a 20 % uncertainty in the 1:1 black gradient line. The alkanes represented by serial number can be identified as follows: (1) Propane; (2) Isobutane; (3) n-Butane; (4) Isopentane; (5) n-pentane; (6) Cyclopentane; (7) 2,2-Dimethylbutane; (8) 2,3-Dimethylbutane; (9) 2-Methylpentane; (10) 3-Methylpentane; (11) Methylcyclopentane; (12) 2,4-Dimethylpentane; (13) Cyclohexane; (14) 2-Methylhexane; (15) 3-Methylhexane; (16) 2,2,4-Trimethylpentane; (17) n-Heptane; (18) Methylcyclohexane; (19) 2,3,4-Trimethylpentane; (20) 2-Methylheptane; (21) 3-Methylheptane; (22) n-Octane; (23) Nonane; (24) n-Decane; (25) n-Undecane.

## 3.3 Temperature dependence (273-323 K)

This study also explored kinetic temperature dependence in the tropospheric range (273-323 K), and n-hexane (Arrhenius expression: $K=(2.43\pm0.52)\times10^{-11}$ exp [$-(481.2\pm60)/T$] at 240-340 K was used as the reference compound. Measured values for 24 C3-C10 alkanes in $N_2$/air systems were provided at different temperatures (273-323



K) in Table S2. And the preexponential factor A and activation energy $E_a/R$ obtained by
linear regression along with the values of the literature were listed in Table 2. The value
of preexponential factor A increases with the increase of the number of carbon atoms,
which is consistent with the law of its reactivity. Additionally, Arrhenius plots were
linearly fitted using this data along with literature data. The following is a detailed
analysis for several components that are important or temperature dependence data has
been less or no studied, the Arrhenius plots are shown in Figure 4-5, other components
are listed in the Supplement (Fig. S3-S15).
Table 2. Summary of Arrhenius Expression of the Reaction of OH· with C3-C11
alkanes in this work and other studies.

| Alkanes | Temperature (K) | A-factor [a] ($\times 10^{-11}$) | $E_a/R$ [b] (K) | Bath gas | Technique [c] | Reference |
|---|---|---|---|---|---|---|
| 2,3-Dimethylbutane | 273-323 | 1.15±0.09 1.17±0.08 | 219±24 227±20 | Air N$_2$ | RR/DP/GC-FID | this work |
| | 240-1220 | $1.66\times10^{-17}$ T$^2$ | 407 | | Review | (Atkinson and Arey, 2003) |
| | 250-1366 | $1.3\times10^{-12}$ (T/298)$^{2.08}$ | 426 | Ar | AR/DF/LIF | (Badra and Farooq, 2015) |
| | 220-1292 | $2.287\times10^{-17}$ T$^{1.958}$ | 365 | -- | Review | (Sivaramakrishnan and Michael, 2009) |
| Methylcyclopentane | 273-323 | 1.65±0.19 1.62±0.14 | 262±33 256±25 | Air N$_2$ | RR/DP/GC-FID | this work |
| | 230-370 | -- | -- | -- | AR/DF/LIF | (Sprengnether et al., 2009) |
| n-Heptane | 273-323 | 3.96±0.37 2.59±0.38 | 544±28 422±43 | Air N$_2$ | RR/DP/GC-FID | this work |
| | 290-1090 | 1.28±0.21 | 190 | -- | Review | (Atkinson and Arey, 2003) |
| | 241-406 | 3.38±0.17 | 497±16 | He | RR/DF/MS | (Wilson et al., 2006) |
| | 240-340 | 2.25±0.14 | 293±37 | He | RR/DF/MS | (Crawford et al., 2011) |
| | 248-896 | $2.7\times10^{-16}$ T$^{1.7}$ | 138 | He/H$_2$/NO$_2$ | AR/DF/LIF | (Morin et al., 2015) |
| | 298-500 | 0.0986 | 600 | -- | Theory | (Cohen, 1991) |
| 3-Methylheptane | 273-323 | 3.54±0.34 2.72±0.45 | 456±28 374±49 | Air N$_2$ | RR/DP/GC-FID | this work |



| | 273-323 | 4.22±0.49 | 497±34 | Air | RR/DP/GC-FID | this work |
|---|---|---|---|---|---|---|
| | | 4.12±0.77 | 487±55 | $N_2$ | | |
| | 240-340 | 2.27±0.21 | 296±27 | He | RR/DF/MS | (Li et al., 2006) |
| | 284-384 | 4.52±0.37 | 538±27 | He | RR/DF/MS | (Wilson et al., 2006) |
| n-Octane | 290-1080 | 1.78 | 235 | -- | Review | (Atkinson and Arey, 2003) |
| | 296-497 | 2.57 | 332±65 | He | AR/FP/KS | (Greiner, 1970a) |
| | 298-1000 | 0.0986 | 600 | -- | Theory | (Cohen, 1991) |
| | 273-323 | 2.38±0.90 | 952±110 | Air | RR/DP/GC-FID | this work |
| | | 2.31±0.81 | 947±102 | $N_2$ | | |
| | 296-908 | $2.72\times10^{-12}\,T^{1.46}$ | 270 | $NO_2/H_2O$ | AR/FP/LIF | (Bryukov et al., 2004) |
| Propane | 227-428 | 1.29 | 730 | Ar | RR/DP/GC | (Demore and Bayes, 1999) |
| | 233-376 | 1.01 | 660 | He | AR/FP/LIF | (Talukdar et al., 1994) |
| | 300 - 390 | 1.12 | 692 | $N_2$ | AR/EB/LIF | (Donahue et al., 1998) |
| | 273-323 | 2.29±0.74 | 739±94 | Air | RR/DP/GC-FID | this work |
| | | 3.56±0.88 | 871±73 | $N_2$ | | |
| | 300 - 390 | 0.626 | 321 | $N_2$ | AR/EB/LIF | (Donahue et al., 1998) |
| Isobutane | 213-372 | 0.572 | 293 | He | AR/FP/LIF | (Talukdar et al., 1994) |
| | 297-498 | 0.347 | 192 | He | AR/FP/GC | (Greiner, 1970a) |
| | 220-407 | $5.24\times10^{-15}T^{1.125}$ | -- | He | RR/DF/MS | (Wilson et al., 2006) |
| | 273-323 | 3.78±0.66 | 867±52 | Air | RR/DP/GC-FID | this work |
| | | 3.90±0.67 | 860±51 | $N_2$ | | |
| | 235 - 361 | 1.68 | 584 | Ar | RR/DP/GC | (Demore and Bayes, 1999) |
| | 300 - 390 | 1.34 | 513 | $N_2$ | AR/EB/LIF | (Donahue et al., 1998) |
| n-Butane | 231-378 | 1.18 | 470 | He | AR/ DF/LIF | (Talukdar et al., 1994) |
| | 294-509 | $0.156\,T^{1.95}$ | 133 | He | AR/ DF/LIF | (Droege and Tully, 1987) |
| | 298-420 | 1.76 | 559 | Ar | AR/ DF/RF | (Perry et al., 1976) |





| | 298-416 | 0.629 | 126 | $H_2O$ | AR-UV | (Gordon and Mulac, 1975) |
|---|---|---|---|---|---|---|
| Isopentane | 273-323 | 1.46±0.17 / 1.20±0.21 | 443±34 / 388±52 | Air / $N_2$ | RR/DP/GC-FID | this work |
| | 213-407 | 1.52 | 432 | $N_2$ | RR/DP/GC | (Wilson et al., 2006) |
| | 273-323 | 0.90±0.05 / 1.73±0.20 | 310±17 / 502±35 | Air / $N_2$ | RR/DP/GC-FID | this work |
| | 233-364 | 1.94 | 494 | Ar | RR/DP/GC | (Demore and Bayes, 1999) |
| n-pentane | 300-390 | 2.97 | 608 | $N_2$ | AR/EB/LIF | (Donahue et al., 1998) |
| | 224-372 | $3.13\times10^{-17}T^2$ | -115 | He | AR/FP/LIF | (Talukdar et al., 1994) |
| | 243-325 | -- | -- | $N_2/O_2/NO$ | RR/DP/GC | (Harris and Kerr, 1988) |
| | 273-323 | 3.67±0.63 / 3.48±0.51 | 619±51 / 608±43 | Air / $N_2$ | RR/DP/GC-FID | this work |
| | 288-407 | 2.71 | 526 | $N_2/H_2O$ | RR/DP/GC | (Wilson et al., 2006) |
| Cyclopentane | 240-340 | 2.43±0.50 | 481±58 | He | RR/DF/MS | (Singh et al., 2013) |
| | 273 - 423 | 2.57 | 498 | Ar | RR/DP/GC | (Demore and Bayes, 1999) |
| | 300-390 | 1.88 | 352 | $N_2$ | AR/EB/LIF | (Donahue et al., 1998) |
| | 295-491 | $4.50\times10^{-15}T^{1.21}$ | 511 | He | AR/FP/LIF | (Droege and Tully, 1987) |
| | 273-323 | 3.53±1.28 / 4.76±1.21 | 899±106 / 986±74 | Air / $N_2$ | RR/DP/GC-FID | this work |
| 2,2-Dimethylbutane | 240-330 | 3.37 | 809 | | Review | (Atkinson and Arey, 2003) |
| | 243-328 | -- | -- | $N_2/O_2/NO$ | RR/DP/GC | (Harris and Kerr, 1988) |
| | 254-1327 | $1.11\times10^{-17}T^{2.09}$ | 79 | Ar | AR/DF/LIF | (Badra and Farooq, 2015) |
| 2-Methylpentane | 273-323 | 2.30±0.29 / 2.27±0.34 | 479±38 / 478±44 | Air / $N_2$ | RR/DP/GC-FID | This work |
| | 283-387 | 2.07 | 413 | $N_2$ | RR/DP/GC | (Wilson et al., 2006) |
| 3-Methylpentane | 273-323 | 2.44±0.39 / 2.45±0.56 | 511±17 / 500±67 | Air / $N_2$ | RR/DP/GC-FID | this work |



| | | | | | | |
|---|---|---|---|---|---|---|
| | 284-381 | 2.16 | 375 | $N_2$ | RR/DP/GC | (Wilson et al., 2006) |
| | 297 - 1362 | $9.75\times10^{-18}T^{2.1}$ | -348 | Ar | AR/DF/LIF | (Badra and Farooq, 2015) |
| | 273-323 | 2.03±0.17 1.60±0.26 | 452±24 382±48 | Air $N_2$ | RR/DP/GC-FID | this work |
| 2,4-Dimethylpentane | 272-410 | 2.25 | 408 | $N_2$ | RR/DP/GC | (Wilson et al., 2006) |
| | 271-1311 | $2.00\times10^{-16}T^{1.71}$ | -143.5 | Ar | AR/DF/LIF | (Badra and Farooq, 2015) |
| | 273-323 | 3.62±0.59 | 522±48 | Air $N_2$ | RR/DP/GC-FID | this work |
| cyclohexane | 240-340 | 3.96±0.60 | 554±42 | He | RR/DF/MS | (Singh et al., 2013) |
| | 288-408 | 3.40 | 513 | $N_2$ | RR/DP/GC | (Wilson et al., 2006) |
| 2-Methylhexane | 273-323 | 1.30±0.08 1.22±0.04 | 222±19 206±9 | Air $N_2$ | RR/DP/GC-FID | this work |
| | 230 - 385 | -- | -- | -- | AR/ DF/LIF | (Sprengnether et al., 2009) |
| 3-Methylhexane | 273-323 | 2.53±1.45 2.27±0.31 | 575±161 559±42 | Air $N_2$ | RR/DP/GC-FID | this work |
| | 230-379 | -- | -- | -- | AR/ DF/LIF | (Sprengnether et al., 2009) |
| 2,2,4-Trimethylpentane | 273-323 | 1.61±0.22 1.23±0.11 | 499±40 418±27 | Air $N_2$ | RR/DP/GC-FID | this work |
| | 240-500 | 1.62 | 443 | | AR/ DF/LIF | (Atkinson, 1986) |
| | 230-385 | 1.54 | 456 | | AR/ DF/LIF | (Atkinson, 2003) |
| Methylcyclohexane | 273-323 | 4.39±0.58 2.99±0.30 | 475±29 364±39 | Air $N_2$ | RR/DP/GC-FID | this work |
| | 273-343 | 1.85±0.27 | 195±20 | Air | RR/DP/FTIR | (Bejan et al., 2018) |
| | 230-379 | -- | -- | -- | AR/ DF/LIF | (Sprengnether et al., 2009) |
| 2,3,4-Trimethylpentane | 273-323 | 1.34±0.07 1.22±0.08 | 203±15 175±19 | Air $N_2$ | RR/DP/GC-FID | this work |
| | 287-373 | 1.3 | 221 | $N_2$ | RR/DP/GC | (Wilson et al., 2006) |
| 2-Methylheptane | 273-323 | 3.93±1.33 1.62±0.37 | 536±102 265±70 | Air $N_2$ | RR/DP/GC-FID | this work |



| | 273-323 | 5.29±0.63 | 520±35 | Air | RR/DP/GC- | this work |
| Nonane | | 2.75±0.27 | 325±29 | N₂ | FID | |
| | 240-340 | 4.35±0.49 | 411±32 | He | RR/DF/MS | (Li et al., 2006) |
| | 273-323 | 5.78±0.49 | 499±25 | Air | RR/DP/GC- | this work |
| n-Decane | | 3.59±0.40 | 353±33 | N₂ | FID | |
| | 240-340 | 2.26±0.28 | 160±36 | He | RR/DF/MS | (Li et al., 2006) |

[a, b]The error bar was taken as σ.
[c]RR: relative rate; AR: absolute rate; DF: discharge flow; DP: direct photolysis; FP:
flash photolysis; EB: electron beam; UV: Ultraviolet; GC: gas chromatography; FID:
flame ionization detection; LIF: laser induced fluorescence; FTIR: fourier transform
infrared spectrometer; MS: mass spectrometry; KS: kinetic-spectroscopy.

**A. OH+ n-Octane.** Figure 5 (a) exhibits the Arrhenius plot for the reaction
between n-Octane and OH radicals in both the nitrogen and air systems, covering a
temperature range of 273 to 323 K. At high temperatures, our data align well with
previous    studies.    The    derived    Arrhenius    expressions    are    as    follows:
$K_{3\text{-Methylheptane}}^{N_2}=(4.12\pm0.77)\times10^{-11}\exp[-(487\pm55)/T]$           $cm^3 \cdot molecule^{-1} \cdot s^{-1}$,
$K_{3\text{-Methylheptane}}^{Air}=(4.21\pm0.49)\times10^{-11}\exp[-(497\pm34)/T]$ $cm^3 \cdot molecule^{-1} \cdot s^{-1}$. These results
agree well with the Arrhenius expression of $(4.52\pm0.37)\times10^{-11}\exp[-(538\pm27)/T]$
$cm^3 \cdot molecule^{-1} \cdot s^{-1}$ reported by Wilson et al. (Wilson et al., 2006) between 284 and 384
K, but contrast the expressions of $(2.27\pm0.21)\times10^{-11}\exp[-(296\pm27)/T]$ $cm^3 \cdot molecule^{-1} \cdot s^{-1}$
$^1 \cdot s^{-1}$ reported by Li et al. between 240 and 340 K (Li et al., 2006) and
$(2.57)\times10^{-11}\exp[-(332\pm65)/T]$ $cm^3 \cdot molecule^{-1} \cdot s^{-1}$ reported by Greiner (Greiner, 1970a)
between 296 and 497 K. Further investigations are necessary to understand the
discrepancies amongst these studies.
**B. OH+ n-Heptane.** The Arrhenius plot in Fig. 5 (b) displays the reaction between
n-Heptane and OH radicals in both the nitrogen and air systems, covering a temperature
range of 273 to 323 K. Our experimental data align closely with previous studies, with
differences ranging from 9% to 15% lower than the transition state theory data reported
by Cohen (Cohen, 1991) between 298 and 500 K. By fitting our data to the Arrhenius
equation,    the    resulting    Arrhenius    expressions    are    as    follows:



428   $K_{n\text{-Heptane}}^{N_2}=(2.59\pm0.37)\times10^{-11}\exp\left[-(422\pm43)/T\right]$   $cm^3\cdot molecule^{-1}\cdot s^{-1}$,

429   $K_{n\text{-Heptane}}^{Air}=(3.96\pm0.38)\times10^{-11}\exp\left[-(544\pm28)/T\right]$ $cm^3\cdot molecule^{-1}\cdot s^{-1}$. These results agree

430   well with the Arrhenius expression of $(3.38\pm0.17)\times10^{-11}\exp\left[-(497\pm16)/T\right]$

431   $cm^3\cdot molecule^{-1}\cdot s^{-1}$ reported by Wilson et al. (Wilson et al., 2006) between 241 and 406

432   K.

433   **C. OH+ Isopentane.** As Fig. 5 (c), isopentane was extensively studied in both the

434   nitrogen and air systems over a temperature range (273-323 K). As far as we know, at

435   present, only Wilson et al. has reported this compound in the range of 213-407 K

436   (Wilson et al., 2006). Our data is slightly 10% lower than that reported by Wilson et al.,

437   but this is still within the margin of error, especially at high temperatures. The Arrhenius

438   expression obtained by fitting the data points in the figure is as follows:

439   $K_{Isopentane}^{N_2}=(1.20\pm0.21)\times10^{-11}\exp\left[-(443\pm34)/T\right]$   $cm^3\cdot molecule^{-1}\cdot s^{-1}$,

440   $K_{Isopentane}^{Air}=(1.46\pm0.17)\times10^{-11}\exp\left[-(497\pm34)/T\right]$ $cm^3\cdot molecule^{-1}\cdot s^{-1}$. The results are

441   similar to the relative experimental results of Wilson et al.

442   $(1.52\pm0.21)\times10^{-11}\exp\left[-(432\pm27)/T\right]$ $cm^3\cdot molecule^{-1}\cdot s^{-1}$.

443   **D. OH+ 2,3-Dimethylbutane.** Figure 5 (d) shows the Arrhenius plot for the

444   reaction of 2,3-Dimethylbutane with OH radicals in the nitrogen and air systems over

445   the temperature range of 273 K to 323 K. The temperature-dependent values obtained

446   in this study align closely with those reported by Badra and Farooq (Badra and Farooq,

447   2015), who used the absolute rate technique in an inert gas system (Ar), as well as the

448   work of Sivaramakrishnan and Michael with a three-parameter fit (Sivaramakrishnan

449   and Michael, 2009). However, in comparison to the reviewed data from Atkinson and

450   Arey (Atkinson and Arey, 2003), our results were found to be approximately 3% to 7%

451   lower. Tate constants are subjective and are in the range ±20-30%. This discrepancy can

452   primarily be attributed to differences in the selected rate constants for reference

453   compounds. Take 298k as an example, the reference value selected in the reviewed data

454   of Atkinson (Atkinson, 1986) is in the range of $5.02\times10^{-12}$-$5.45\times10^{-12}$, while in this

455   work, we choose n-hexane as the reference, its $K_{OH}$ value is $4.84\times10^{-12}$, and the



reference k value is reduced by about 4%-13%. However, since 1986, the rate constants
of most of the alkanes obtained have decreased by about 10%. Linear regression applied
to our data yields the Arrhenius expressions as follows:
$K^{N_2}_{2,3\text{-Dimethylbutane}} = (1.17 \pm 0.08) \times 10^{-11} \exp[-(227 \pm 20)/T]$  $cm^3 \cdot molecule^{-1} \cdot s^{-1}$,
$K^{Air}_{2,3\text{-Dimethylbutane}} = (1.15 \pm 0.09) \times 10^{-11} \exp[-(219 \pm 224)/T]$  $cm^3 \cdot molecule^{-1} \cdot s^{-1}$.  The
results show that within the error range, the Arrhenius expressions of OH+2,3-
Dimethylbutane in the nitrogen and air systems are almost consistent. However, as
shown in Table S2, although the rate constants are very consistent, the activation energy
are quite different than those in the wide temperature range.

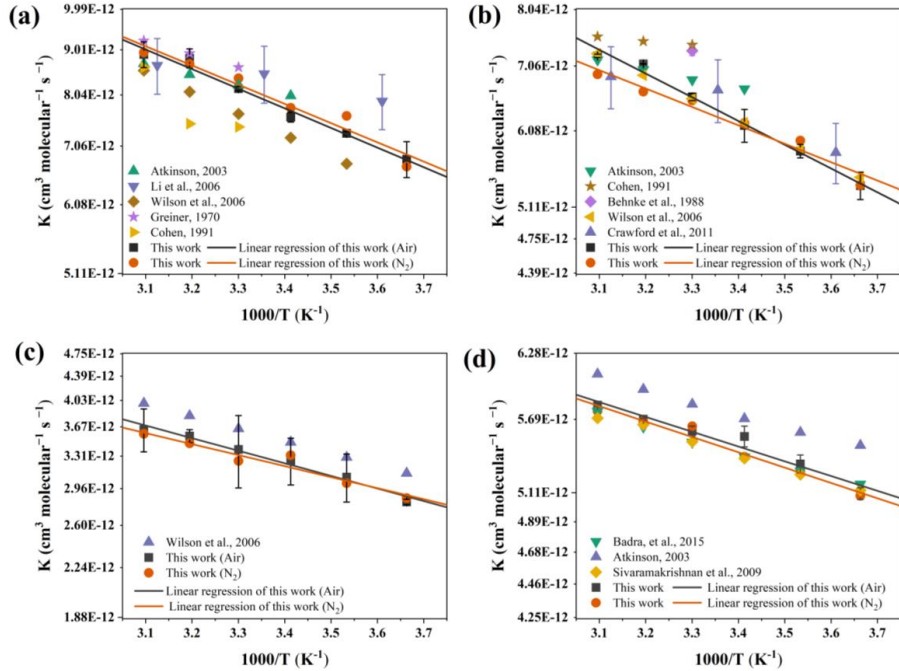


Figure 5. Arrhenius plots for the reaction of n-Octane (a), n-Heptane (b), Isopentane (c)
and 2,3-Dimethylbutane (d) with OH· at 273-323 K along with available literature data.
The error bar was taken as 2σ.
**E. OH+ Methylcyclopentane (2-Methylhexane).** Figure 6 (a) and (b) illustrate
the Arrhenius plot for the reaction of methylcyclopentane and 2-methylhexane with OH
radicals in both nitrogen and air systems, spanning a temperature range of 273 to 323



K. Literature data from Sprengnether et al. (Sprengnether et al., 2009) and Anderson et
al. (Anderson et al., 2004) are available for comparison purposes. Notably, for
methylcyclopentane, Anderson et al. (Anderson et al., 2004) reported absolute data that
is 26% higher than the relative data obtained in this study at 298 K. However, this
difference falls within the margin of error. The absolute data from Sprengnether et al.
(Sprengnether et al., 2009) is slightly higher, ranging from 10% to 20%, compared to
this study. Additionally, they derived an alternative Arrhenius expression to
accommodate the curved behavior of the rate constant between 230 and 370 K, making
it difficult to directly compare with our Arrhenius expression. The resulting Arrhenius
expressions of methylclopentane and 2-methylhexane they derived an alternative
Arrhenius expression to accommodate the curved behavior of the rate constant between
230 and 370 K, making it difficult to directly compare with our Arrhenius expression.
are as follows: $K^{N_2}_{Methylcyclopentane}=(1.62\pm0.14)\times10^{-11}\exp[-(256\pm25)/T]$ $cm^3\cdot molecule^{-1}\cdot s^{-1}$,
$K^{Air}_{Methylcyclopentane}=(1.65\pm0.19)\times10^{-11}\exp[-(262\pm33)/T]$ $cm^3\cdot molecule^{-1}\cdot s^{-1}$.
$K^{N_2}_{2-Methylhexane}=(1.22\pm0.04)\times10^{-11}\exp[-(206\pm9)/T]$ $cm^3\cdot molecule^{-1}\cdot s^{-1}$,
$K^{Air}_{2-Methylhexane}=(1.30\pm0.08)\times10^{-11}\exp[-(222\pm19)/T]$ $cm^3\cdot molecule^{-1}\cdot s^{-1}$. To the best of
our knowledge, this is the first investigation of the temperature-dependent kinetics for
the reaction of methylcyclopentane and 2-methylhexane with OH radicals utilizing the
relative rate technique. The consistency of the Arrhenius expressions in both the
nitrogen and air systems implies that the bath gas does not significantly impact the
reaction between OH and methylcyclopentane and 2-methylhexane.
**F. OH+ 3-Methylheptane.** In Figure 6 (c), the Arrhenius plot presents the reaction
between 3-Methylheptane and OH radicals in both the nitrogen and air systems,
spanning a temperature range of 273 to 323 K. A linear regression analysis of our data
yields the following Arrhenius expressions:
$K^{N_2}_{3-Methylheptane}=(3.54\pm0.45)\times10^{-11}\exp[-(374\pm49)/T]$ $cm^3\cdot molecule^{-1}\cdot s^{-1}$,
$K^{Air}_{3-Methylheptane}=(2.72\pm0.34)\times10^{-11}\exp[-(456\pm28)/T]$ $cm^3\cdot molecule^{-1}\cdot s^{-1}$. Within the
margin of error, the expression in the nitrogen system is consistent with that in the air



system between 273 and 323 K. We believe this study to be the first investigation of the temperature-dependent kinetics for the reaction between 3-Methylheptane and OH radicals. The only previous study on this reaction, reported by Shaw et al. (Shaw et al., 2020) utilizing the relative rate method in nitrogen at 323 K, demonstrates significantly higher data (>65%) compared to our results. Possible explanations for this discrepancy lie in the different reference compounds used and potential sample loss during sampling in the enrichment tube in Shaw et al.

**G. OH+ 3-Methylhexane (Figure 6 (d))**. This is the first temperature-dependence relative data. It can be seen from the figure that this data is this data is significantly lower by approximately 80% compared to the absolute data. Under low-temperature (273, 283 K) in a nitrogen gas system, it does not conform to the Arrhenius fit. This enlightens us that for research below 283 K, we still need to carry out experiments in a larger low temperature range for analysis. When excluding the low-temperature data in the nitrogen system, the Arrhenius expression is as follows:

$K^{N_2}_{\text{3-Methylhexane}}=(2.27\pm0.31)\times10^{-11}\exp[-(559\pm42)/T]$ $cm^3 \cdot molecule^{-1} \cdot s^{-1}$,

$K^{Air}_{\text{3-Methylhexane}}=(2.53\pm1.45)\times10^{-11}\exp[-(575\pm161)/T]$ $cm^3 \cdot molecule^{-1} \cdot s^{-1}$.

**H. OH+ 2-Methylheptane (Figure 6 (e))**. There are no previous temperature dependence data on this compound. Similar to 3-Methylhexane, this data is lower by approximately 37% compared to Shaw et al. at room temperature. Furthermore, the data obtained in nitrogen and air systems at 273-283 K shows an increase. Within the range of 293-323 K, the obtained Arrhenius expression is as follows:

$K^{N_2}_{\text{2-Methylheptane}}=(1.62\pm0.37)\times10^{-11}\exp[-(265\pm70)/T]$ $cm^3 \cdot molecule^{-1} \cdot s^{-1}$,

$K^{Air}_{\text{2-Methylheptane}}=(3.93\pm1.33)\times10^{-11}\exp[-(536\pm102)/T]$ $cm^3 \cdot molecule^{-1} \cdot s^{-1}$. The pre-exponential factor A and activation energy Ea of the air system are slightly higher than those of the nitrogen system.

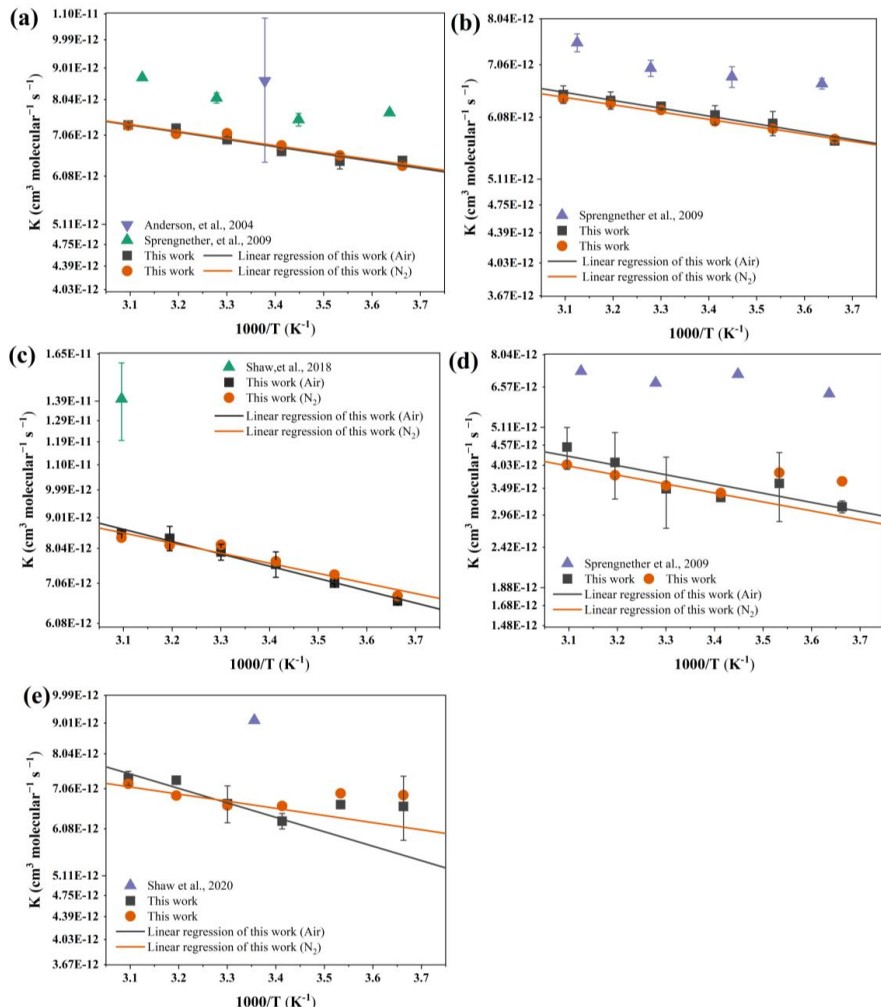

525

Figure 6. Arrhenius plots for the reaction of Methylcyclopentane (a), 2-Methylhexane

(b), 3-Methylhepane (c), 3-Methylhexane (d) and 2-Methylhepane (e) with OH· at 273-

323 K along with available literature data. The error bar was taken as 2σ.

## 4. Conclusions

The use of the multivariate relative rate method in this study allowed for the

simultaneous determination of reaction rate constants of C3-C11 alkanes and OH

radicals in different bath gases, which significantly improved the efficiency of



determination. New data and Arrhenius expressions for the reaction of Methylcyclopentane, 2-Methylhepane, 3-Methylheptane, 2-Methylhexane and 3-Methylhexane with OH radicals were obtained for the first time in the temperature range of 273-323 K, expanding the existing database. The measured relative rate constants of air bath gases in the temperature range studied were found to be highly consistent with values obtained in $N_2$, suggesting that the rate constants obtained in this experiment can reasonably represent the rate constants in the actual atmosphere. The structure-additivity method for rate constant estimation is mostly consistent for the prediction of $K_{OH}$ (298 K) for the studied n-alkanes, but its methodology and parameters do not seem to be able to reasonably estimate the rate constant of 2,3-dimethylbutane. Additionally, there is a big discrepancy in the case of several cycloalkanes (cyclopentane, methylcyclopentane, cyclohexane) and branch alkanes (2,2,4-Trimethylpentane and 2,3,4-Trimethylpentane) with this experiment for estimation parameters' overestimate. There is a reasonable suspicion that this method is still lacking some additional factors.

## Data availability

Raw data are available upon request.

## Author contributions

Yujing Mu and Chengtang Liu planned the campaign; Yanyan Xin performed the measurements; Yanyan Xin, Chengtang Liu, Yujing Mu and Xiaoxiu Lun analyzed the data; Yanyan Xin and Chengtang Liu wrote the manuscript draft. Shuyang Xie and Junfeng Liu provided technical support.

## Competing interests

The authors declare that they have no conflict of interest.



## Acknowledgements

This work was supported by the National Natural Science Foundation of China (Nos. 22076202, 42077454 and 41975164).

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
