# Peer review of "determined by the relative rate technique"

_EGUsphere, 2023_

## Referee Comment (RC2)

**Manuscript ID:** EGUsphere 2023-2802
**Authors:** Yanyan Xin, Xiaoxiu Lun, Shuyang Xie, Junfeng Liu, Chengtang Liu, Yujing Mu
**Ms. Title:** Rate coefficients for the reactions of OH radicals with C3-C11 alkanes determined by the relative rate technique.

**General Comments:**

The authors report room temperature and temperature-dependent rate coefficients for OH radicals with a series of C3 – C11 alkanes (straight-, branch-chain, and cycloalkanes), using a relative rate technique. They have also looked into the bath gas effect on the measured rate coefficients and they have compared their results with literature data, where available, and structure reactivity relationships (SAR) estimates. Although the results from this work are worth to be published there are some major and some minor issues that need to be addressed before acceptance. One of the major concerns of the present reviewer is related to the lack of physical interpretation of the observed differences in $k$ values in some reactions when $N_2$ was used as bath gas. Further, for the majority of the studied reactions, there are available data in the literature. It is not a problem to provide additional kinetic data, but this makes sense when there is a thorough study that provides more accurate data or addresses problems of previous studies. In this work, the reason for doing that is not justified, and studying so many compounds at once does not improve data quality. **Suggestion**: Maybe publishable after major revisions.

**General Comments:**

1. *Abstract*, ln. 15. Please correct "reported were determined" with "reported was determined"

2. It is highly recommended the authors to correct the rate coefficient capital K throughout the manuscript to avoid confusion with the reaction equilibrium constant. Please use $k$ to refer to the reaction rate coefficient.

3. Please avoid using rate constant since $k$ is not a constant, particularly since the authors have also measured the rate coefficient temperature dependence ($k(T)$). Although rate constant is commonly used it would be better to be more accurate.

4. Please replace '$cm^3 \cdot molecule^{-1} \cdot s^{-1}$' with '$cm^3\ molecule^{-1}\ s^{-1}$' throughout the text.

5. Please replace OH· and $NO_3$· with OH and $NO_3$ radical throughout the text.

6. Pg. 2, ln. 47–50. The reason that OH abstraction reactions dominate atmospheric degradation is related to both the faster reactions OH radical initiate along with the relative abundance of the oxidants. So, the at least 5 orders of magnitude slower reactions that $NO_3$ radical initiate

need to be combined with the relative abundance of OH and $NO_3$ radical during daytime. Please include a sentence to address that or, even better, remove the whole discussion with $NO_3$ chemistry, since it is out of the scope of this study.

7. Pg 3. Ln. 63–67. The statement in which relative rate and absolute methods are compared (stated twice in the document) is misleading. First, absolute rate measurements are not that rare, and time-resolved measurements have been extensively and very efficiently used in the past by some of the leading groups on kinetics studies. The recommendation of the present reviewer is to either rephrase or delete this argument (in both places), particularly since the two different techniques have advantages and disadvantages that are not only related to the OH detection difficulties or the accurately measured compound concentrations. Both of the latter should not be an issue nowadays! Secondary photochemistry of different sources is one of the issues that both techniques need to test and combat.

**Major Concerns:**

1. Why the authors didn't use the recommended rate coefficients from the two evaluation panels (IUPAC, NASA/JPL) for the reference reactions and they preferably, where exist? There are some issues with reproducing the quoted data in the tables. Also, in the section where they describe the criteria for reference reaction selections, they have fully omitted one of the most important criteria, which that is the reference reaction rate coefficient needs to be similar to the one under study, to increase measurement sensitivity. Please include. This way both axes range are similar and the concentration variations can be measured with similar precision. Also please include $1\sigma$, not just $\sigma$ in error analysis and describe if this includes systematic uncertainties (reference reaction rate coefficients are one of the major sources of that in relative rate determinations). In general, error analysis and references to that are vague. What is the meaning of $2\sigma$ levels on SAR? What are the major sources of errors (random and systematic) in their measurements?

2. Although the authors have used $O_2$, air ($N_2/O_2$), and $N_2$, as bath gases test measurements and interpretation analysis is incomplete. In general, it is common that when not enough/excess of $O_2$ is present in relative rate measurements, it is likely that the radicals are not efficiently scavenged and might initiate secondary chemistry, e.g., reactants – targeted or/and reference – regeneration or even chain chemistry that will result in rate coefficient underestimates or overestimates. Radicals react with $O_2$ pretty fast, e.g., typically in the order of $10^{-12}$ $cm^3$ $molecule^{-1}$ $s^{-1}$ which is in the same range as the understudied reactions. In general, to test this, people run experiments at different $O_2$ levels to ensure that the rate coefficient is not altered. At pressures close to atmospheric pressure where ~150 Torr of $O_2$ is present, there is enough

of $O_2$ to scavenge the produced radicals, in most cases, which is not the case when the non-reactive $N_2$ is used as bath gas – there is always some small amount of $O_2$ even then. Although in the case of hydrogen metathesis, this does not always result in a problem, depending also on the rate coefficient of the studied reaction and the chemistry involved, it might be an issue for slower reactions that compete with radical oxidation. It would have been nice if the authors had experimentally demonstrated that and if not acknowledged this effect in the interpretation of their results.

---

## Author Comment (AC1)

**RC2: General Comments**

**1.   Abstract, ln. 15. Please correct "reported were determined" with "reported was determined"**

**Reply:** Sorry for the mistake! The "reported were determined" have been modified to "reported was determined" in the revised manuscript.

…a new room temperature relative rate constant for 3-methylheptane that had not been previously reported was determined…

**2.   It is highly recommended the authors to correct the rate coefficient capital K throughout the manuscript to avoid confusion with the reaction equilibrium constant. Please use k to refer to the reaction rate coefficient.**

**Reply:** Yes! All rate constant symbols K and $K_{OH}$ have been modified to $k$ or $k_{OH}$ in the revised manuscript.

For example: Table 1 listed the obtained $k_{OH}$ for C3-C11 alkanes…

…the obtained $k_{OH}$ values all fall within the shadow range.

…

**3.   Please avoid using rate constant since k is not a constant, particularly since the authors have also measured the rate coefficient temperature dependence (k(T)). Although rate constant is commonly used it would be better to be more accurate.**

**Reply:** Thanks for your valuable suggestions! We have modified the rate coefficient symbol k temperature dependence to k(T) in the revised manuscript.

For example:  …the derived Arrhenius expressions are as follows: $k_{n\text{-}Octane}(T)=(5.17\pm0.97)\times10^{-11}\exp[-(546\pm60)/T]$ cm$^3$ molecule$^{-1}$ s$^{-1}$…

…213-407 K obtained by fitting our data and those of Wilson et al. is as follows: $k_{Isopentane}(T)=(1.39\pm0.12)\times10^{-11}\exp[-(424\pm25)/T]$ cm$^3$ molecule$^{-1}$ s$^{-1}$…

…

**4.   Please replace 'cm$^3$·molecule$^{-1}$·s$^{-1}$' with 'cm$^3$ molecule$^{-1}$ s$^{-1}$' throughout the text.**

**Reply:** Thanks for your valuable suggestions! All 'cm$^3$·molecule$^{-1}$·s$^{-1}$' have been

replaced with 'cm$^3$ molecule$^{-1}$ s$^{-1}$' in the revised manuscript.

**5.    Please replace OH· and NO$_3$· with OH and NO$_3$ radical throughout the text. in the revised manuscript**

**Reply:** Thanks for your valuable suggestions! All instances of OH· and NO$_3$· have been replaced with OH and NO$_3$ radical in the revised manuscript.

**6.    Pg. 2, ln. 47–50. The reason that OH abstraction reactions dominate atmospheric degradation is related to both the faster reactions OH radical initiate along with the relative abundance of the oxidants. So, the at least 5 orders of magnitude slower reactions that NO$_3$ radical initiate need to be combined with the relative abundance of OH and NO$_3$ radical during daytime. Please include a sentence to address that or, even better, remove the whole discussion with NO$_3$ chemistry, since it is out of the scope of this study.**

**Reply:** Thanks for your valuable suggestions! Following your suggestions and those of other reviewers, the discussion involving NO$_3$ chemistry has been removed from the revised manuscript.

**7.    Pg 3. Ln. 63–67. The statement in which relative rate and absolute methods are compared (stated twice in the document) is misleading. First, absolute rate measurements are not that rare, and time-resolved measurements have been extensively and very efficiently used in the past by some of the leading groups on kinetics studies. The recommendation of the present reviewer is to either rephrase or delete this argument (in both places), particularly since the two different techniques have advantages and disadvantages that are not only related to the OH detection difficulties or the accurately measured compound concentrations. Both of the latter should not be an issue nowadays! Secondary photochemistry of different sources is one of the issues that both techniques need to test and combat.**

**Reply:** Thanks for your valuable suggestions! The section comparing relative rate and absolute methods has been revised in the updated manuscript.

…Unlike the absolute rate constant method, the relative rate method relied on the

known rate constant for the reaction of a reference compound with OH radicals, with the reference reaction rate coefficient needing to be similar to that of the compound under study to enhance measurement sensitivity. By monitoring the simultaneous decay of the target and reference compounds in the presence of OH radicals due to competitive response mechanisms, the rate constant for the reaction of OH radicals with the target compound can be determined (Atkinson and Arey, 2003; Shaw et al., 2018). From 1980s to 2020s, dozens of papers for the rate coefficients of alkanes with OH radical measured by relative rate mehod have been published…

**Major comments:**

**1.  Why the authors didn't use the recommended rate coefficients from the two evaluation panels (IUPAC, NASA/JPL) for the reference reactions and they preferably, where exist? There are some issues with reproducing the quoted data in the tables. Also, in the section where they describe the criteria for reference reaction selections, they have fully omitted one of the most important criteria, which that is the reference reaction rate coefficient needs to be similar to the one under study, to increase measurement sensitivity. Please include. This way both axes range are similar and the concentration variations can be measured with similar precision. Also please include 1σ, not just σ in error analysis and describe if this includes systematic uncertainties (reference reaction rate coefficients are one of the major sources of that in relative rate determinations). In general, error analysis and references to that are vague. What is the meaning of 2σ levels on SAR? What are the major sources of errors (random and systematic) in their measurements?**

**Reply:** Thanks for the valuable suggestions! The recommended rate coefficients from the panels of IUPAC and NASA/JPL have been added in the revised manuscript. For instance, Table 1 includes database recommendation data, such as, $k_{n\text{-}Heptane}$=6.80×$10^{-12}$ $cm^3$ molecule$^{-1}$ s$^{-1}$ (Anderson et al., 2004). We apologize for the error in Table 1 presentation and the error analysis in the manuscript. We have now utilized the rate constants recommended by the McGillen et al., (2020) database as reference rate

constants to accurately recreate the table and have meticulously reviewed its content. Additionally, the format of Table 2 has been adjusted and standardized. For the error analysis mentioned in the manuscript, σ has been modified to 1σ, including systematic uncertainties. As you mentioned, selecting reference compounds with rate coefficients similar to the target reaction rate coefficient is crucial. Further details on the conditions required for relative rate techniques have been included in our revised manuscript.

[revised manuscript text omitted]

**2. Although the authors have used $O_2$, air ($N_2/O_2$), and $N_2$, as bath gases test measurements and interpretation analysis is incomplete. In general, it is common**

that when not enough/excess of O₂ is present in relative rate measurements, it is likely that the radicals are not efficiently scavenged and might initiate secondary chemistry, e.g., reactants – targeted or/and reference – regeneration or even chain chemistry that will result in rate coefficient underestimates or overestimates. Radicals react with O₂ pretty fast, e.g., typically in the order of $10^{-12}$ cm₃ molecule⁻¹ s⁻¹ which is in the same range as the understudied reactions. In general, to test this, people run experiments at different O₂ levels to ensure that the rate coefficient is not altered. At pressures close to atmospheric pressure where ~150 Torr of O₂ is present, there is enough of O₂ to scavenge the produced radicals, in most cases, which is not the case when the nonreactive N₂ is used as bath gas – there is always some small amount of O₂ even then. Although in the case of hydrogen metathesis, this does not always result in a problem, depending also on the rate coefficient of the studied reaction and the chemistry involved, it might be an issue for slower reactions that compete with radical oxidation. It would have been nice if the authors had experimentally demonstrated that and if not acknowledged this effect in the interpretation of their results.

**Reply:** Thanks for your valuable suggestions! As you and other reviewer mentioned, there is always some small amount of O₂ in the N₂ bath gas, the impact of bath gas has been reduced in our revised manuscript. For example, the study on relative rate experiments and temperature dependence in different bath gases in Section 3.1 and 3.3 have been removed, with the discussion now centered on the temperature dependence relationship in the air system.

[revised manuscript text omitted]

---

## Author Comment (AC2)

**Reply to Reviewer's Comments**

RC1: Major reply:

**1. The authors assessment of the kinetics literature is currently incomplete. This is made clear by the fact that they take credit for making the first temperature-dependent measurements in cases where measurements are clearly available, as well as the incomplete literature data presented in Table 2. I would suggest that the authors make use of the database paper of McGillen et al. (2020), and download the accompanying database. This will achieve two things: 1. It will give the authors a more comprehensive knowledge of the kinetics literature for OH + hydrocarbon reactions. 2. It will provide these authors with critically evaluated rate coefficients for many of the species that are contained within their paper. Regarding the latter, I would strongly encourage the authors to use these recommendations as their reference rate constants where applicable and reanalyze their data accordingly, and if not, I would expect the authors to justify why they do not accept these recommendations**.

**Reply**:Thanks for your valuable suggestions! As your suggestion, we looked at the study reported by McGillen et al. (2020) and used the accompanying version 2.1.0 database, specifically the reference rate constants using the data recommended by this database and the selection part of the $k$ value had been revised in Section 2.1.4 in the revised manuscript.

... The selection of $k$ values for reference compounds and the literature data assessment and comparison gives priority to the available expert-evaluated rate constants wherever possible. Here we used the recommended expert-evaluated data of database for Version 2.1.0 of McGillen et al. (Database for the Kinetics of the Gas-Phase Atmospheric Reactions of Organic Compounds – Eurochamp Data Center), which is relatively comprehensive and provides rigorously evaluated rate coefficients for many species. Among them, at $298 \pm 1$ K, the $k$ values (in units of $cm^3$ molecule$^{-1}$ s$^{-1}$) of the three reference compounds selected respectively are expert-evaluated rate constants: $k_{OH+n\text{-}Hexane}=4.97\times10^{-12}$, $k_{OH+Cyclohexane}=6.69\times10^{-12}$, $k_{OH+n\text{-}Octane}=8.48\times10^{-12}$, which is fitted or manually entered data from multiple sources. However, the value of the reference

compound at different temperatures (273-323 K) is different than the room temperature. A detailed explanation is reflected in Sec. 3.3. At the same time, we updated the data in Table 1 accordingly and reanalyzed our data, such as Section 3.1.

Sec.3.1…the $k_{OH}$ obtained for propane with n-hexane, cyclohexane and n-octane as the reference compound were $(1.38\pm0.01)\times10^{-12}$, $(1.25\pm0.03)\times10^{-12}$ and $(1.34\pm0.04)\times10^{-12}$ (the units are $cm^3$ $molecule^{-1}$ $s^{-1}$), respectively (within 10%).....

In addition, a comparison of all species with the recommended reaction rate constants of the database has been added (Figure 3) and discussed.

As shown in Fig. 3, for the different bath gases, the obtained $k_{OH}$ for C3-C11 alkanes showed high agreement. Meanwhile, it can also be observed from the figure that most of the rate coefficients obtained are very similar to the expert-evaluated values of the database by the McGillen et al. However, 2,4-Dimethylpentane is an exception, the $k_{OH}$ value obtained in this study is about 20% lower than the recommended value, but it is similar to expert-evaluated value by Atkinson and Arey (Atkinson and Arey, 2003). Additionally, it can be clearly seen in the figure that the reactivity of linear alkanes ($RCH_2R$) with OH radicals increasing…

[Figure]

Figure 3. Comparison of rate constants of C3-C11 alkanes in different bath gases ($N_2$, Air, $O_2$) with expert-evaluated data at 298±1 K. The error bar was taken as 1σ.

**2. The presentation of the data/ quality of the data is unsatisfactory. When I inspect the contents of Table 1, taking the reference rate constants as provided on page 7 of the manuscript, I am able to reproduce $k_{OH}$ for the first 3 entries (i.e. propane in N2 with n-hexane, cyclohexane and n-octane as references). Following this, the next 3 entries (propane in air) are inconsistent. I noticed that throughout this table there are many problems of this type. In my judgement, this is not acceptable for a paper whose principle subject is kinetic data and it undermines your experimental work. What is the purpose of this data, if your readers cannot trust it? For this reason, I insist that the authors return to their spreadsheets, remake this table correctly and triple check its contents.**

**Reply:** Sorry for the mistake! According to your valuable suggestions, the rate constants recommended by McGillen et al., (2020) database have been used, and the Table 1 has been corrected in the revised manuscript.

Table 1. Comparison of Experimental in this work with the reported in the literatureat 298±1 K.

[revised manuscript text omitted]

**3. The authors insistence on Arrhenius parameters for this selection of reactions in Section 3.3. is unjustified. With all the high-temperature data available for this collection of compounds, it is plainly obvious that the Arrhenius equation is insufficient to describe the temperature dependencies of any of these reactions. The only reason for using such an equation would be for datasets spanning a small temperature range (or where data precision is insufficient). I strongly encourage the authors to consider fitting their data within the context of the available measurements, because I don't think these new Arrhenius parameters add any value to your paper.**

**Reply:** Thanks for your valuable suggestions! Fitting our data to wide temperature range data from the literature for the 9 alkanes discussed in detail in the article. Detailed

description has been added in Section 3.3. in the revised manuscript.

…**A. OH+ n-Octane.** Figure 5 (a) exhibits the Arrhenius plot for the reaction between n-Octane and OH radicals, covering a temperature range of 240 to 1080 K. Within the experimental temperature range (273-323 K), our data align well with previous studies. Fit our data to expert-evaluated data (manually entered data from multiple sources), the derived Arrhenius expressions are as follows: $k_{n-Octane}$(T)=(5.07±0.97)×10$^{-11}$exp [-(543±61)/T] cm$^3$ molecule$^{-1}$ s$^{-1}$. This result agree well with the Arrhenius expression of (4.52±0.37)×10$^{-11}$exp [-(538±27)/T] cm$^3$·molecule$^{-1}$·s$^{-1}$ reported by Wilson et al. (Wilson et al., 2006) between 284 and 384 K and (4.95±0.87)×10$^{-11}$exp [-(531±56)/T] recommended Arrhenius formula obtained by experts' evaluation of data processing, but contrast the expressions of (2.27±0.21)×10$^{-11}$exp [-(296±27)/T] cm$^3$·molecule$^{-1}$·s$^{-1}$ reported by Li et al. between 240 and 340 K (Li et al., 2006) and (2.57)×10$^{-11}$exp[-(332±65)/T] cm$^3$·molecule$^{-1}$·s$^{-1}$ reported by Greiner (Greiner, 1970a) between 296 and 497 K. By comparison, our data are highly consistent with the data recommended by experts. The obtained Arrhenius expression more accurately represents the relationship between the reaction rate constant of octane and OH radicals and temperature in a wide temperature range, which has certain reference significance. Further investigations are necessary to understand the discrepancies amongst these studies.

**B. OH+ n-Heptane.** The Arrhenius plot in Fig. 5 (b) displays the reaction between n-Heptane and OH radicals in the air systems, covering a temperature range of 240 to 896 K. As shown in the figure, within the experimental temperature range (273-323 K), our data are highly similar to previous studies. By fitting our data and recommended data from multiple sources to the Arrhenius equation, the resulting Arrhenius expressions are as follows: $k_{n-Heptane}$(T)=(5.06±0.45)×10$^{-11}$exp [-(602±30)/T] cm$^3$ molecule$^{-1}$ s$^{-1}$. This result agree well with the Arrhenius expression of (5.20±0.54)×10$^{-11}$exp [-(605±39)/T] cm$^3$·molecule$^{-1}$·s$^{-1}$ reported by Morin et al. (Morin et al., 2015) between 248 and 896 K. The recommended Arrhenius equation for the reaction of OH radical and n-Heptane is in the form $k$ (T) =3.84×10$^{-12}$*exp(148/T)

* $(T/300)^{1.79}$. Rearrange the fitting data to get the Arrhenius expression in the form of $k$ (T) $=(4.82\pm0.43)\times10^{-11}\exp[-(600\pm31)/T]$ cm$^3$ molecule$^{-1}$ s$^{-1}$. Compared with the Arrhenius expression recommended in the literature, the preexponential factor A $(5.01\pm0.42)$ of this work is agree well with the one $(4.82\pm0.43)$ of recommended (the unit is $10^{-11}$ cm$^3$ molecule$^{-1}$ s$^{-1}$). However, the activation energy Ea/R of this work is about 60% higher than the recommended data.

**C. OH+ Isopentane.** As Fig. 5 (c), isopentane was extensively studied over a temperature range (213-407 K). As far as we know, at present, only Wilson et al. has reported this compound in the range of 213-407 K (Wilson et al., 2006). Within the experimental temperature range (273-323 K), our data are consistent with Wilson et al. ((273-323 K), especially in the low temperature range. The Arrhenius expression at 213-407 K obtained by fitting our data and those of Wilson et al. is as follows: $k_{Isopentane}(T)=(1.39\pm0.12)\times10^{-11}\exp[-(424\pm25)/T]$ cm$^3$ molecule$^{-1}$ s$^{-1}$. The results are similar to the relative experimental results of Wilson et al. $(1.52\pm0.21)\times10^{-11}\exp[-(432\pm27)/T]$ cm$^3$ molecule$^{-1}$ s$^{-1}$.

**D. OH+ 2,3-Dimethylbutane.** Figure 5 (d) shows the Arrhenius plot for the reaction of 2,3-Dimethylbutane with OH radicals over the temperature range of 273 K to 1366 K. The temperature-dependent values obtained in this study at high temperature (313-323 K) align closely with those reported by Badra and Farooq (Badra and Farooq, 2015), who used the absolute rate technique, as well as the work of Sivaramakrishnan and Michael with a three-parameter fit (Sivaramakrishnan and Michael, 2009). However, the data obtained at 273-293 K in this work are highly consistent with the reviewed data from Atkinson and Arey (Atkinson and Arey, 2003). Linear regression applied to our data and high temperature data in the literature (at 273-1366 K) yields the Arrhenius expression as follows: $k_{2,3-Dimethylbutane}(T)=(4.81\pm0.56)\times10^{-12}\exp[-(669\pm50)/T]$ cm$^3$ molecule$^{-1}$ s$^{-1}$. This result agrees well with the Arrhenius expression of $(4.75\pm0.71)\times10^{-11}\exp[-(664\pm77)/T]$ cm$^3\cdot$molecule$^{-1}\cdot$s$^{-1}$ reported by Badra and Farooq (Badra and Farooq, 2015).

[Figure]

Figure 5. Arrhenius plots for the reaction of n-Octane (a), n-Heptane (b), Isopentane (c) and 2,3-Dimethylbutane (d) with OH radical in wide temperature range along with available literature data. The error bar was taken as 2σ.

**4. Problems with general consistency. This is an example (but there are several others), Figure 6(a) shows temperature-dependent literature measurements for OH + methylcyclopentane. In the abstract, it states that "… Arrhenius expressions (in units of $cm^3 \cdot molecule^{-1} \cdot s^{-1}$) for the reactions of various cyclo- and branched alkanes with OH were determined for the first time: methylcyclopentane…". I can interpret this in one of two ways: 1. The authors don't appear to be aware that there is temperature-dependent data, even though they have presented it in their figures. 2. The authors are discounting the work of Sprengnether et al. because it is not presented in Arrhenius form. If the former, then the authors should organize their manuscript more carefully. If the latter, then I find this to be a strange idea (after all, if you don't like their equation, just re-fit their data with an Arrhenius equation!)…**

**Reply:** Thanks for your valuable suggestions! We are very sorry that we did not consider the different forms of temperature dependence expressions to fit the original data to obtain the same Arrhenius equation as ours. Table 2 has been updated in the revised manuscript.

[revised manuscript text omitted]

**5. It is not clear to me why the authors chose to consider the bath-gas as an important aspect of these measurements. The literature has many examples of measurements that were conducted in a wide variety of bath gases (helium, argon, nitrogen, air etc.). As far as I am aware, no dependence on bath gas has been noted. In fact, a small amount of oxygen would be necessary in relative rate experiments such as yours, otherwise alkyl radicals formed from the hydrogen abstraction reaction would themselves abstract hydrogen from other alkanes in the system, re-forming the original alkane, consuming some of the other hydrocarbons and confusing your results. In practice, it is difficult to remove oxygen to such an extent, and I would therefore assume that your experiments are not affected by this anoxic chemistry. Either way, I suggest that bath gas is an irrelevance in this paper and can be ignored.**

**Reply:** Thanks for your valuable suggestions! As you and other reviewer mentioned, there is always some small amount of $O_2$ in the $N_2$ bath gas, the impact of bath gas has been reduced in our revised manuscript. For example, the study on relative rate experiments and temperature dependence in different bath gases in Section 3.1 and 3.3 have been removed, with the discussion now centered on the temperature dependence relationship in the air system.

**….3.1 Results from relative rate experiments at 298 K**

The rate constants for the reactions involving OH radical with C3-C11 alkanes in the mixed system were determined at 298±1 K. The concentration curves of target alkanes and the reference compound (n-Hexane) were plotted in Fig. 2. As shown in Fig. 2, the decay of both target and reference compounds correlated well with eq. (7), and high correlation coefficients ($R^2$) were observed for most alkanes, exceeding 0.99. Table 1 and Table S4 listed the obtained $k_{OH}$ for C3-C11 alkanes under three bath gases using the related reference compounds. The error bars (1σ) in Table 1 accounted for reference rate constant uncertainty, and experimental parameter uncertainties (pressure, temperature, flow rate, reactant concentration). The results indicated strong agreement (within <15%) between rate constants for 25 C3-C11 straight-chain, branched-chain, and cycloalkanes, using different reference compounds. For example, the $k_{OH}$ obtained for propane with n-hexane, cyclohexane and n-octane as the reference compound were $(1.38\pm0.01)\times10^{-12}$, $(1.25\pm0.03)\times10^{-12}$ and $(1.34\pm0.04)\times10^{-12}$ (the units are $cm^3$ molecule$^{-1}$ s$^{-1}$), respectively (within 10%). This suggests that reference compound variation minimally affects results, indicating reliable experimental methods and data. Notably, the rate constant for 3-Methylheptane's reaction with OH radicals at room temperature was determined for the first time. As shown in Fig. 3, for the different bath gases, the obtained $k_{OH}$ for C3-C11 alkanes showed high agreement. Meanwhile, it can also be observed from the figure that most of the rate coefficients obtained are very similar to the expert-evaluated values of the database by the McGillen et al. However, 2,4-Dimethylpentane is an exception, the $k_{OH}$ value obtained in this study is about 20% lower than the recommended value…

**….3.3 Temperature dependence (273-323 K)**

In order to study the relationship between temperature and reaction rate constant, this study carried out experiments in the tropospheric temperature range (273-323 K), and combined with the literature data (the expert-recommended data from database for Version 2.1.0 of McGillen et al.) to study the kinetic temperature dependence of several alkanes in a wide temperature range. And n-hexane (Arrhenius expression: $k(T)=(2.43\pm0.52)\times10^{-11}$ exp $[-(481.2\pm60)/T]$ at 240-340 K was used as the reference

compound. Since the research results at room temperature show that different bath gases have little effect on the reaction rate constant, only the temperature dependence of the reaction rate constant under the air system is considered here...

**….A. OH+ n-Octane.** Figure 5 (a) exhibits the Arrhenius plot for the reaction between n-Octane and OH radicals, covering a temperature range of 240 to 1080 K. Within the experimental temperature range (273-323 K), our data align well with previous studies. Fit our data to expert-evaluated data (manually entered data from multiple sources), the derived Arrhenius expressions are as follows: $k_{n\text{-}Octane}(T)=(5.07\pm0.97)\times10^{-11}\exp[-(543\pm61)/T]$ cm$^3$ molecule$^{-1}$ s$^{-1}$. This result agree well with the Arrhenius expression of $(4.52\pm0.37)\times10^{-11}\exp[-(538\pm27)/T]$ cm$^3\cdot$molecule$^{-1}\cdot$s$^{-1}$ reported by Wilson et al. (Wilson et al., 2006) between 284 and 384 K and $(4.95\pm0.87)\times10^{-11}\exp[-(531\pm56)/T]$ recommended Arrhenius formula obtained by experts' evaluation of data processing, but contrast the expressions of $(2.27\pm0.21)\times10^{-11}\exp[-(296\pm27)/T]$ cm$^3\cdot$molecule$^{-1}\cdot$s$^{-1}$ reported by Li et al. between 240 and 340 K (Li et al., 2006) and $(2.57)\times10^{-11}\exp[-(332\pm65)/T]$ cm$^3\cdot$molecule$^{-1}\cdot$s$^{-1}$ reported by Greiner (Greiner, 1970a) between 296 and 497 K. By comparison, our data are highly consistent with the data recommended by experts. The obtained Arrhenius expression more accurately represents the relationship between the reaction rate constant of octane and OH radicals and temperature in a wide temperature range, which has certain reference significance….

**A. OH+ n-Octane.** Figure 5 (a) exhibits the Arrhenius plot for the reaction between n-Octane and OH radicals covering a temperature range of 240 to 1080 K. Within the experimental temperature range (273-323 K), our data align well with previous studies. Fit our data to expert-evaluated data (manually entered data from multiple sources), the derived Arrhenius expressions are as follows: $k_{n\text{-}Octane}(T)=(5.07\pm0.97)\times10^{-11}\exp[-(543\pm61)/T]$ cm$^3$ molecule$^{-1}$ s$^{-1}$. This result agree well with the Arrhenius expression of $(4.52\pm0.37)\times10^{-11}\exp[-(538\pm27)/T]$ cm$^3\cdot$molecule$^{-1}\cdot$s$^{-1}$ reported by Wilson et al. (Wilson et al., 2006) between 284 and 384 K and $(4.95\pm0.87)\times10^{-11}\exp[-(531\pm56)/T]$ recommended Arrhenius formula obtained by experts' evaluation of data processing, but contrast the expressions of

$(2.27\pm0.21)\times10^{-11}\exp[-(296\pm27)/T]$ cm$^3\cdot$molecule$^{-1}\cdot$s$^{-1}$ reported by Li et al. between 240 and 340 K (Li et al., 2006) and $(2.57)\times10^{-11}\exp[-(332\pm65)/T]$ cm$^3\cdot$molecule$^{-1}\cdot$s$^{-1}$ reported by Greiner (Greiner, 1970a) between 296 and 497 K. By comparison, our data are highly consistent with the data recommended by experts. The obtained Arrhenius expression more accurately represents the relationship between the reaction rate constant of octane and OH radicals and temperature in a wide temperature range, which has certain reference significance. Further investigations are necessary to understand the discrepancies amongst these studies…

**6. There are some advantages to studying so many compounds simultaneously, the main one being that it can save you some time. However, there are also some possible problems. The main one would be the formation of products which could interfere with some of your analyte peaks. I see no discussion of any sort regarding products of the reaction. Do you see any product peaks in the GC-FID? If not, why not?**

**Reply:** Yes. In our experiments, we used the GC-FID developed by the research team to accurately detect 57 compounds. The advantage of this instrument is that it can easily and quickly observe the attenuation of multiple compounds in less than 1 h. The results of the chromatogram for various species during 10-60 minutes of reaction shows the variation of acetone. For the asymmetrical peak shape of this compound in the GC-FID with OV-1 column, it is difficult to accurately quantified. In the future, we want to study those compounds by the developed GC-FID equipment with polar column.

**Minor comments:**

**General: the symbol for rate constant in the kinetics literature is an italicized lower-case k. It is not to be confused with an upper-case K (which is reserved for units of kelvin), or an italicized upper-case K (which is normally reserved for equilibrium constants).**

**Reply:** Sorry for the mistake! All rate constant symbols K and $K_{OH}$ have been modified to $k$ or $k_{OH}$ in the revised manuscript.

For example: Table 1 listed the obtained $k_{OH}$ for C3-C11 alkanes…

…the obtained $k_{OH}$ values all fall within the shadow range.

…

**Abstract: revise according to suggestions above.**

**Reply:** Thanks for your valuable suggestions! The Abstract had been revised according to suggestions above in the revised manuscript.

Abstract: Rate coefficients for the reactions of OH radicals with C3-C11 alkanes were determined using the multivariate relative rate technique. A total of 25 relative rate coefficients at room temperature and 24 Arrhenius expressions-in different temperature range were obtained. Notably, a new room temperature relative rate constant for 3-methylheptane that had not been previously reported was determined, and the obtained $k_{OH}$ values (in units of $10^{-12}$ cm$^3$ molecule$^{-1}$ s$^{-1}$) was 7.71±0.35. Interestingly, whilst results for n-alkanes agreed well with available structure activity relationship (SAR) calculations, the three cyclo-alkanes and one trimethylpentane were found to be less reactive than predicted by SAR. Conversely, the SAR estimate for 2,3-dimethylbutane were approximately 25% lower than the experimental value, highlighting that the limited understanding of the oxidation chemistry of these compounds. Arrhenius expressions (in units of cm$^3$ molecule$^{-1}$ s$^{-1}$) for the reactions of various branched alkanes with OH radical were determined for the first time: 2-methylheptane, $(1.62±0.37)×10^{-11}$exp [-(265±70)/T] , and 3-methylheptane, $(3.54±0.45)×10^{-11}$exp [-(374±49)/T]. The reactivity relation of saturated alkanes with OH radicals and chlorine atoms was obtained: $\log_{10}[k_{(Cl+alkanes)}]$ = $0.569×\log_{10}[k_{(OH+alkanes)}]$-3.111 (R$^2$ =0.86). In addition, the rate coefficients for the 24 previous studied OH + alkanes reactions were consistent with existing literature values, demonstrating the reliability and efficiency of this method for simultaneous investigation of gas-phase reaction kinetics.

**Introduction:**

**Line 44. why are you making comparisons with NO₃·? It is well known that the**

**abstraction reactions are unimportant for the alkanes. Chlorine on the other hand may become important in some environments.**

**Reply:** Thanks for your valuable suggestions! Following your suggestions and those of other reviewers, the discussion involving NO$_3$ chemistry has been removed from the revised manuscript.

**Line 44. "Dehydrogenation" of alkanes leads to alkenes. You mean to say "hydrogen abstraction".**

**Line 46. I assume by "rate constants", the authors mean "room temperature rate constants". You should specify this.**

**Line 47. Assuming that the authors have by now become more familiar with the kinetic database, you will of course know that the range of reactivity of alkanes goes from 6.36E-15 (methane) to 2.16E-11 (n-hexadecane) at the time of writing. The range provided is therefore misleading.**

**Line 47. "mol" is absolutely not an abbreviation of "molecule". "mol" is an abbreviation of "mole", which would be highly misleading.**

**Line 48. Rate constants are not faster or slower than other rate constants. They are larger or smaller.**

**Reply:** Sorry for the mistakes! These mistakes have been corrected in the revised manuscript.

**Line 66. In fact, precise measurements are highly desirable in the relative rate method. Low precision in your GC-FID measurements would lead to scatter in your relative rate plots. Therefore, this statement is misleading.**

Reply: Thanks for your valuable suggestions! The contents of relative rate method had been revised according to your suggestions and those of several other reviewers, so the description here no longer exists in the revised manuscript.

However, due to its high experimental difficulty, sensitivity to reaction conditions, and high requirements for instrumentation and equipment, the experimental conditions need to be strictly controlled and the measurement is more complicated. Alternatively, the

relative rate method, and this approach is also widely used to determine $k_{OH}$ values for organic compounds. The basic principle is that the rate constant for the reaction of the reactant used as a reference with OH radicals is known, and the reference reaction rate coefficient needs to be similar to the one under study, to increase measurement sensitivity…

**Methods:**

**General comment: several tests have been made with respect to dark losses, photolytic losses etc. However, as far as I can tell, no tests were performed regarding storage in the 1 litre sample bags. In your experiments, your samples are stored for some time before they are analysed by the GC-FID are they not? During this time, your samples are subjected to conditions of higher surface area to volume ratios, and it is here, where I would expect to see the most wall loss.**

**Reply:** Yes! The VOCs loss caused by the storage of samples in PVF bags had been evaluated in our previous work. As shown in Fig.R1, there is no obvious loss of 25 alkanes in the PVF bag for 10 hours.

[Figure]

Fig.R1 The VOC loss in PVF bags

**Line 115. Excess with respect to what?**

**Reply:** Thank you very much professor for your question! Excess with respect to what has been added in the revised manuscript.

While excess $H_2O_2$ respect to VOCs was injected through a three-way valve using a micro syringe. Initial conditions of the different species introduced into the reactor for each experiment are outlined in Table S1 in the Supplementary Material.

**Line 145. "self-developed".**

**Reply:** Sorry for the mistake! The "self-develop" has been modified to "self-developed" in the revised manuscript.

…Collected samples were subsequently analyzed using a self-developed automated injection system …

**Line 155. It is not clear to me how you have improved or expanded the work of Shaw et al. Furthermore, it is also not clear to me how the results of this work are significantly different from any other relative rate study in which several reference compounds are considered.**

**Reply:** Sorry for the mistake! We have corrected it in the revised manuscript. In addition, as you said, there have been many reports on the relative rates of alkanes and OH radicals, but most of them focus on one or several alkanes for experiments. And some data are controversial, the reaction rate constant of 2,4-Dimethylpentane with OH radical evaluated by Atkinson and Arey is $4.8 \times 10^{-12}$ $cm^3$ $molecule^{-1}$ $s^{-1}$, while the recommended expert-evaluated data of database for Version 2.1.0 of McGillen et al. is $5.76 \times 10^{-12}$ $cm^3$ $molecule^{-1}$ $s^{-1}$, which is about 20% higher than the data of Atkinson and Arey. The special feature of this work is to select the alkanes in the mixed gas of PAMs, which has an important effect on O3, and to study it in Shaw et al. Based on the application of multivariate relative rate method, and combined with the GC-FID developed by the experimental group, a large number of compounds can be studied at the same time, and the determination efficiency is improved. In addition, there is still

only one or no reported data for some alkanes, such as 3-methylheptane, which is measured for the first time, making up for the gap in the database.

The research method of this work is based on the multivariate relative rate method published by Shaw et al. (Shaw et al., 2018).

**Line 173. There is a certain irony to this statement, because I would strongly recommend that you should use the available expert-evaluated rate constants wherever possible.**

**Reply:** Thanks for your valuable suggestions! The selection of $k$ values for reference compounds has been modified in the revised manuscript.

…The selection of $k$ values for reference compounds and the literature data assessment and comparison gives priority to the available expert-evaluated rate constants wherever possible. Here we used the recommended expert-evaluated data of database for Version 2.1.0 of McGillen et al. (Database for the Kinetics of the Gas-Phase Atmospheric Reactions of Organic Compounds – Eurochamp Data Center), which is relatively comprehensive and provides rigorously evaluated rate coefficients for many species. Among them, at $298 \pm 1$ K, the $k$ values (in units of $cm^3$ molecule$^{-1}$ s$^{-1}$) of the three reference compounds selected respectively are expert-evaluated rate constants: $k_{OH+n-Hexane}=4.97\times10^{-12}$, $k_{OH+Cyclohexane}=6.69\times10^{-12}$, $k_{OH+n-Octane}=8.48\times10^{-12}$, which is fitted or manually entered data from multiple sources…

**Line 193. Was the H$_2$O$_2$ purified?**

**Reply:** Yes! $H_2O_2$ was obtained from Sinopharm Chemical Reagent Co., Ltd. as 30 wt % solution and was concentrated by bubbling helium through it prior to use.

**Lines 211-212. I don't know what the authors mean by general error is 2 sigma… Do you mean that the uncertainties for SAR estimates are generally within a factor of two? This is possibly true in a global sense, but it is considerably lower for the alkane dataset (the subject of this paper).**

**Reply:** Thanks for your valuable suggestion! As you mentioned, the 2σ was widely used in the uncertainties analysis for SAR estimates. For the relatively high carbon number of alkanes studied in this study, this value was chosen for comparation with others.

**Line 238. I don't know what an "error strip" is. I think the authors mean "error bars".**

**Reply:** Yes. The expression of "error strip" has been replaced with "error bars" in the revised manuscript.

…The error bars (1σ) in Table 1 accounted for data…

**Line 238. What is fitting dispersion?**

**Reply: Reply:** Sorry for the mistake! The "**fitting dispersion**" has been corrected in the revised manuscript.

**Results:**

**Figure 3. It would be useful to include literature recommendations for each of these rate constants where available, allowing us to see how well the experiments are performing.**

**Reply:** Thanks for your valuable suggestions. Comparison with the data of literature recommendations in figure 3 have been added in the revised manuscript.

[Figure]

Figure 3. Comparison of rate constants of C3-C11 alkanes in different bath gases ($N_2$, Air, $O_2$) with expert-evaluated data at 298±1 K. The error bar was taken as σ.

**Table 1. As noted above, the errors in this table are unacceptable at present. In addition to this, the formatting of this table is confusing and should be rethought to help the readers understand it better.**

**Reply:** We apologize for the error in Table 1 presentation and the error analysis in the manuscript. We have now utilized the rate constants recommended by the McGillen et al., (2020) database as reference rate constants to accurately recreate the table and have meticulously reviewed its content.

[revised manuscript text omitted]

**Structure-activity relationships section:**

**You only compare with one SAR (the Atkinson SAR). This is a missed opportunity, there are several others to choose from (some examples include: Neeb, 2000; Jenkin et al., 2018; McGillen et al., 2024). In the case of Neeb, this is of particular relevance because there is some critical discussion on the use of a ring-strain**

**correction factor on page 6 (300) of that study, the authors should consider this in their discussion of SAR performance for cyclic compounds. Incidentally, the recent SAR of McGillen et al. (2024) is not currently configured for cyclic compounds, however, I have assessed its performance on the selection of compounds of this paper. This is very easily done by running the Python code of that paper, from which I find that it performs marginally better than Kwok and Atkinson (1995). This, at least for this limited selection of compounds, supports Neeb's statements about ring strain corrections. However, it is most likely the case that more data on cyclic alkanes of differing ring size would be very useful in assessing this further.**

**Reply:**

Thanks for your valuable suggestions! The obtained reaction rate constants were compared 
[revised manuscript text omitted]

---

## Author Comment (AC3)

**Reply to Reviewer's Comments**

**RC3: Major comments:**

**1. The introduction provides a large number of literature data; however, a very chaotic presentation induces to readers the feeling of jumping from one study to another, all of them mostly with very general information regarding the kinetic of alkanes. The studies mentioned in the introduction are presented without an effective detail of the rate coefficient information. I would write the introduction assessing the importance of alkanes for air quality with their impact to atmosphere, then I would add information about their concentrations in troposphere and the impact on potential ozone formation, potential SOA formation and their sources and sinks. One important point of the study is to highlight the importance of accurate kinetic rate coefficients for database, global model atmospheric processes and degradation mechanisms. As one of the reviewers mentioned already, the McGillen et al., (2018) database is very important to be used as a start in the literature data assessment and comparison.**

**Reply:** Thanks for your valuable suggestions! The introduction section has been modified in the revised manuscript based on your suggestions.

… emitted into the atmospheric environment through natural and anthropogenic sources, e.g., C5-alkanes emitted from gasoline usage and C6-alkanes and higher homologous VOCs emitted as a consequence of their usage as solvents and from fuel evaporation. (Atkinson, 2000; Guenther, 2002; Atkinson and Arey, 2003). In the troposphere, alkanes are degraded and removed from the atmosphere via gas-phase oxidation reactions with OH and $NO_3$ radicals, Cl atoms and ozone ($O_3$) (Atkinson and Arey, 2003; Shi et al., 2019). These oxidation processes will form a photochemical smog in the presence of NOx and light, causing regional photochemical pollution (Fiore et al., 2005; Ling and Guo, 2014). Additionally, some secondary oxides produced by the oxidation of alkanes can form secondary organic aerosol (SOA) through homogeneous nucleation or condensation onto existing primary particles (Sun et al., 2016). To fully understand the role of alkanes in atmospheric chemistry, accurate chemical reaction rate data is an important criterion for evaluating its reactivity (Shaw

et al., 2018)….

…. Unlike the absolute rate constant method, the relative rate method relied on the known rate constant for the reaction of a reference compound with OH radicals, with the reference reaction rate coefficient needing to be similar to that of the compound under study to enhance measurement sensitivity. By monitoring the simultaneous decay of the target and reference compounds in the presence of OH radicals due to competitive response mechanisms, the rate constant for the reaction of OH radicals with the target compound can be determined (Atkinson and Arey, 2003; Shaw et al., 2018)…

**2.    The reason of selected those 25 alkanes is not presented in the study and their selection looks arbitrary. An organized evaluation based on their separation on straight-chain, branching and cyclic alkane structure would be more interesting and helpful to get valuable information. First class of alkanes including linear compounds could provide information regarding reactivity of each $CH_2$ group added to their structure and how the rate coefficient value would change over increasing alkane chain. Secondly, from the branching alkanes the authors could have more information related to CH groups added to the alkane structure. The third class in the evaluation would be cyclic alkanes where the authors could extract information regarding the reactivity increase with the cycle size from cyclobutane to cyclodecane. The authors should include in their evaluation the alkanes rate coefficients studied in present study and those existing in the literature, to release more complete discussion on their behaviour and reactivity (figure 3 and 4). As an example, there are a lot of data which could be included, mentioning here only a few ($k_{OH}$ for cyclooctane, bicyclo octane, methyl octane, etc.).**

**Reply:** Thanks for your valuable suggestions! The 25 selected alkanes are the alkanes in the PAMs mixed gas that are widely present in the atmospheric environment and contribute significantly to $O_3$ production. The classification discussion of 25 alkanes has been added in the revised manuscript based on your suggestions.

…For each additional $CH_2$ group from C3-C11, the reaction rate constant increases about 0.95-1.81 (the unit is $10^{-12}$ $cm^3$ molecule$^{-1}$ s$^{-1}$), reflects the fact that the main way is to extract the H atom from the second-order C-H bond. For branching alkanes, for example, 2,2-Dimethylbutane and 2,3-Dimethylbutane, it is obvious that the addition of CH group increases the reaction rate constants with OH radical to a great extent. For cyclic alkanes, such as cyclopentane, methylcyclopentane, cyclohexane and methylcyclohexane, it can also be seen that the reactivity increase with the increase of cycle size. By comparing the reaction rate constant of cyclopentane and cyclohexane (methylcyclopentane and methylcyclohexane), it is found that for cyclic alkanes, each $CH_2$ group reaction rate increases by about $2.37 \times 10^{-12}$ $cm^3$ molecule$^{-1}$ s$^{-1}$. It can be seen from the reaction rate constant of cyclopentane and methylcyclopentane (cyclohexane and methylcyclohexane) that the reaction rate constant increases about $2.06 \times 10^{-12}$ $cm^3$ molecule$^{-1}$ s$^{-1}$ for cycloalkanes with each increase of methyl.

**3. Since the authors highlight their first study on 3-methylheptane why they do not cover other not studied yet branched alkanes (2,2,3-trimethylpentane, etc.)**

**Reply:** This work mainly focuses on the study of 25 kinds of alkanes in PAMs mixed gases, which are widely present in the atmospheric environment and contribute significantly to the production of $O_3$. However, 2,2,3-trimethylpentane is not included in the PAMs mixed gases. The reaction rate constant of 2,2,3-trimethylpentane with OH radical can be further measured in our future studies.

**4. The data reported in the Table 2 should be reevaluated and presented in more concise and understandable form. There are multiple examples of inconsistency of the data with many average data not well calculated (i.e. isopentane in $N_2$).**

**Reply:** Sorry for the mistake! The data reported in the Table 2 had been reevaluated and presented in more concise and understandable form in the revised manuscript.

[revised manuscript text omitted]

**5.    With the extensive interpretation of data including the existing literature rate coefficients, the authors would be able to evaluate the accurate reactivity trends for the class of alkane toward OH radicals and calculate new factors for the SAR method and then to improve the SAR method. A simple comparison with the Kwok and Atkinson SAR method is not worth to do. Evaluation of existing SAR approaches in the literature, with discussion about the influence on the substituent factors, is necessary. (McGillen et al., 2024 (doi.org/10.1039/D3EA00147D), Jenkin et al., 2018 (doi.org/10.5194/acp18-9297-2018).**

**Reply:** Thanks for your valuable suggestions! Only the alkane data of this work had been classified and discussed to evaluate the reactivity trends of alkanes towards OH radicals in the revised manuscript.

…To evaluate the reliability of our experimental data, multiple comparisons were made between the obtained reaction rate constants and the SAR values of different experimental groups (Figure 4). As shown in Figure 4, most n-alkanes are fall into the shaded region, indicating a high level of agreement for $k_{OH}$ rate coefficients of most n-alkanes (experimental values) with the SAR values, particularly for C3-C11 n-alkanes (about within 10%). Although the measured values of n-butane and n-pentane were lower than the estimated values of Neeb (2000), the similar trend was observed when comparing our experimental data with the SAR values of Wilson et al, 2006, and Jenkin et al, 2018 (refer to Fig. 4 (c) and Fig. 4 (d)), suggesting a certain level of reliability in our results…

…For branch alkanes, such as monomethyl branched alkanes (2-Methylpentane, 3-Methylpentane, 2-Methylhexane, 3-Methylhexane 2-Methylheptane and 3-Methylheptane), the obtained $k_{OH}$ values all fall within the shadow range. The results indicated a relatively consistent alignment between our experimental data and the SAR estimated data within a certain margin of error, particularly for the SAR values of Neeb and Jenkin et al. (within 8%). Nevertheless, there seemed to be something different for

polymethyl branched alkanes, like 2,3-Dimethylbutane, the experimental data was about 25% higher than the estimated SAR values of Atkinson and Kwok et al. (1995) and Neeb (2000), especially 53% higher than that of Jenkin et al. (2018). This suggested a potential underestimation of $k_{OH}$ values of 2,3-dimethylbutane by these SAR estimation methods…

…For cyclic alkanes, such as cyclopentane and cyclohexane, the obtained $k_{OH}$ values in this study were approximately 32% and 15%, respectively, lower than the SAR values of Atkinson and Kwok et al., 1995; b. Neeb 2000; c. Jenkin et al. 2018. On the other hand, the obtained experimental values for methylcyclopentane and methylcyclohexane were similar to SAR values of Neeb and Wilson et al (within 5%) (Neeb, 2000; Wilson et al. 2006), However, compared with the SAR values of Atkinson and Kwok et al. and Jenkin et al., this result is about 15% and 8% lower…

**6. The reaction channel of the OH radical initiated degradation of alkanes is strictly related to hydrogen abstraction in the presence of oxygen. There are clearly correlations on the alkane reactivity with OH radicals and Cl atoms for all saturated class of VOCs. Please evaluate a log-log correlation of $k_{CL}$ and $k_{OH}$ as presented by Calvert et al., 2011 (Calvert, J., Mellouki, A., Orlando, J., Pilling, M., and Wallington, T.: Mechanisms of Atmospheric Oxidation of the Oxygenates, Oxford University Press) for alkanes, saturated alcohols and ethers and also by Tovar et al. (2022) (doi.org/10.5194/acp-22-6989-2022) for saturated epoxides.**

**Reply:** Thanks for your valuable suggestions! More discussion on correlation between the rate coefficients of the reaction of alkanes with OH radicals and chlorine atoms had been added in our revised manuscript.

**3.4 Correlation between the rate coefficients of the reaction of alkanes with OH radicals and chlorine atoms**

…Figure 7 presents a log–log correlation plot between the Cl atoms and OH radical rate coefficients with the series of C3-C11 studied above. A very clear correlation ($R^2$ =0.86) described by the relation $\log_{10}[k_{(Cl+alkanes)}] = 0.569 \times \log_{10}[k_{(OH+alkanes)}]$-3.111 was obtained. Although the correlation between propane and isobutane is relatively

discrete, the reactivity of saturated alkanes with OH radicals and chlorine atoms is still clearly related to the saturated alkane series. In addition, the log–log correlation for the series of saturated alkanes with these two oxidants presented by Calvert et al. (2011) described by the relation $\log_{10}[k_{(Cl+alkanes)}] = 0.521 \times \log_{10}[k_{(OH+alkanes)}]$-3.670 with ($R^2$=0.85) is in better agreement with the log–log correlations obtained in this study for saturated alkanes. This correlation can be utilized to predict rate coefficients for unmeasured reactions, such as the reaction of 2,2,3-trimethylpentane with chlorine…

**7. The importance of the bath-gas is over highlighted in this study and a single example for a selected alkane would be enough to prove that is no bath-gas effect on the rate coefficient value. A revaluation of the paper consistency should be performed.**

**Reply:** Thanks for your valuable suggestions! As you and other reviewer mentioned, there is always some small amount of $O_2$ in the $N_2$ bath gas, the impact of bath gas has been reduced in our revised manuscript. For example, the study on relative rate experiments and temperature dependence in different bath gases in Section 3.1 and 3.3 have been removed, with the discussion now centered on the temperature dependence relationship in the air system.

**….3.1 Results from relative rate experiments at 298 K**

[revised manuscript text omitted]

**8. Please add more information regarding the conditions needed for relative rate techniques. Also include more advantages and disadvantages of the absolute and relative techniques.**

**Reply:** Thanks for your valuable suggestions! More information regarding the conditions needed for relative rate techniques and more advantages and disadvantages

of the absolute and relative techniques in this work in the introduction section had been added in our revised manuscript.

…Unlike the absolute rate constant method, the relative rate method relied on the known rate constant for the reaction of a reference compound with OH radicals, with the reference reaction rate coefficient needing to be similar to that of the compound under study to enhance measurement sensitivity. By monitoring the simultaneous decay of the target and reference compounds in the presence of OH radicals due to competitive response mechanisms, the rate constant for the reaction of OH radicals with the target compound can be determined (Atkinson and Arey, 2003; Shaw et al., 2018)…

**9.   The authors should highlight the atmospheric implication and the impact of their research as requested by a scientific journal as "Atmospheric Chemistry and Physics". Please add information about the alkane lifetime in the atmosphere toward the OH radicals. A more extensive conclusion and atmospheric implication should be performed.**

Reply: Thanks for your valuable suggestions! The atmospheric lifetime and implications had been added in the in the revised manuscript.

…The atmospheric lifetime of alkanes in the troposphere can be estimated using the following formula:

$$\tau_{alkane} = 1/(k_{alkane+OH}[OH])$$

where $\tau_{alkane}$ is the atmospheric lifetime of the alkane due to OH removal, $k_{alkane+OH}$ is the rate constant for the reaction of the alkane with OH radical at the typical tropospheric temperature of 298 K, and [OH] is the atmospheric concentrations of the hydroxyl radicals. The average tropospheric hydroxyl radical concentration has been previously reported in the literature as $1 \times 10^6$ molecules cm$^{-3}$ (Li et al., 2018). Using the $k_{alkane+OH}$ (298 K) values determined in the present work, the atmospheric lifetime for 25 alkanes was estimated and listed on the Table S3. As can be seen from the table, the atmospheric lifetime of C3-C11 alkanes reacting with OH radicals are about 1-11 days. As the carbon chain grows, the atmospheric lifetime seems to reduce, especially for long-chain alkanes with carbon atoms of 8-11, the residence time in the atmosphere

is only about 1 day. They are emitted into the air and degraded quickly to generate alkyl radicals, which are immediately converted into alkyl peroxy radicals by reacting with abundant $O_2$ in the atmosphere. The subsequent reaction of alkyl peroxyl radicals enhances the conversion of NO to $NO_2$ by $HO_2$ radicals, leading to the production of tropospheric ozone. For short-chain alkanes that stay in the atmosphere for a long time, such as propane, the lifetime is 11d. It should be noted that because the OH concentration is the global average estimated concentration, the applicability of the lifetime may be different in the atmosphere with different OH radical concentrations...

**Minor comments:**

**Line 84: Finlayson-Pitts**

**Reply:** Thanks for your valuable suggestions! Modifications have been made in the literature in the revised manuscript.

Finlayson-pitts, B. J., Hernandez, S. K., and Berko, H. N.: A new dark source of the gaseous hydroxyl radical for relative rate measurements, Journal of Physical Chemistry, 97, 1172-1177, 10.1021/j100108a012, 1993.

**Line 127: "at253"**

**Reply:** Thanks for your valuable suggestions! Modifications have been made in the revised manuscript.

…The 25 alkanes were detected by FID at 523.15 K after programmed heating at 253.15 K, 303.15 K and 433.15 K in 30 min…

**Line 130: please avoid given values in the form of 0.00013 or 0.00048. Change the units to pptv/h.**

**Reply:** Thanks for your valuable suggestions! The form of 0.00013 or 0.00048 had been revised in the revised manuscript.

…The $K_d$ values ranged from 1.3 to 4.8 (the units are $10^{-4}$ ppbv/h)…

**Line 283 and 285: please add units**

**Reply:** Sorry for the mistake! Units have been added in the revised manuscript.

… in the air gas is (2.63±0.23), the unit is $10^{-12}$ $cm^3$ molecule$^{-1}$ s$^{-1}$ (applicable to all units involved in this paragraph).

**Line 460: please revise the rate coefficient**

**Reply:** Yes! The rate coefficient had been revised in the revised manuscript.

**…**Linear regression applied to our data and expert- recommended data (at 253-263 K) yields the Arrhenius expression as follows: $k_{2,3\text{-}Dimethylbutane}$ (T) =(4.81±0.56)×$10^{-12}$exp [-(669±50)/T] $cm^3$ molecule$^{-1}$ s$^{-1}$…

**Line 104: The figure of the simulation chamber shows a ratio of 200:50 of $N_2$:$O_2$ mixture. The study used synthetic air or this mixture shown in the figure?**

**Reply:** Thank you for your question. As shown by the figure of the simulation chamber, the air used in this study is a mixture of nitrogen (200L) and oxygen (50L) at 4:1.

[revised manuscript text omitted]

---

## Author Response (AR2)

RC1:

**Line 37-38 Please revise the sentence since the alkanes are extremely less reactive with NO₃ and ano reacting with ozone.**

**Reply:** Thanks for your valuable suggestions! Modifications have been made in the revised manuscript.

…In the troposphere, the alkanes are extremely less reactive with $NO_3$ and ano reacting with ozone, they are degraded and removed from the atmosphere via gas-phase oxidation reactions with OH radicals and chlorine atoms…..

**Line 41: Replace "some secondary oxides" with degradation products.**

**Reply:** Thanks for your valuable suggestions! Modifications have been made in the revised manuscript.

…Additionally, degradation products produced by the oxidation of alkanes can form…

**Line 47: There are not only these two methods used for absolute measurements.**

**Reply:** Thanks for your valuable suggestions! Modifications have been made in the revised manuscript.

…(such as flash photolysis and emission flow et al.)…

**Line 74: "Finlaysonpitts et al"**

**Reply:** Thanks for your valuable suggestions! Modifications have been made in the revised manuscript.

Finlaysonpitts, B. J. and Pitts, J. N., Jr.: Tropospheric air pollution: ozone, airborne toxics, polycyclic aromatic hydrocarbons, and particles, Science (New York, N.Y.), 276, https://doi.org/1045-1052, 10.1126/science.276.5315.1045, 1997.

**Line 225: "RCH2R", there is better to add R1CH2R2**

**Reply:** Thanks for your valuable suggestions! Modifications have been made in the

revised manuscript.

…the reactivity of linear alkanes $(R_1CH_2R_2)$ with OH radicals increasing…

**Line 475: Please revise the linear regression fit for figure 6e and the y axis for all figures.**

**Reply:** Thanks for your valuable suggestions! Modifications have been made in the revised manuscript.

[Figure]

**Line 490: Please give the rate coefficient for 2,2,3-trimethylpentane with 491 chlorine atoms.**

**Reply:** Thanks for your valuable suggestions! Modifications have been added in the revised manuscript.

…such as the reaction of 2,2,3-trimethylpentane with chlorine atoms. It is currently known that the rate constant for the reaction of 2,2,3-trimethylpentane with OH radical at room temperature is $4.84 \times 10^{-12}$ $cm^3$ $molecule^{-1}$ $s^{-1}$, according to the above correlation equation, it can be inferred that the rate constant with chlorine atoms is $2.72 \times 10^{-10}$ $cm^3$ $molecule^{-1}$ $s^{-1}$…

**Line 511: "11d"**

**Reply:** Thanks for your valuable suggestions! Modifications have been made in the revised manuscript.

…..such as propane, the lifetime is 11 days….

**Line 520-522: please refine the conclusions after the modification performed from the reviewers suggestions**

**Reply:** Thanks for your valuable suggestions! Modifications have been made in the revised manuscript.

The use of the multivariate relative rate method in this study allowed for the simultaneous determination of reaction rate coefficients of $C_3$-$C_{11}$ alkanes and OH radicals, which significantly improved the efficiency of determination. A total of 25 relative rate coefficients at room temperature were obtained, including the determination of a previously unreported room temperature relative rate coefficient for 3-methylheptane. For the studied n-alkanes, the obtained rate coefficients ($k_{OH}$) were found to be consistent with results estimated by the SAR methods using parameters provided by various positional groups, such as Atkinson and Kwok, Neeb, Wilson, Jenkin, and McGillen. However, it is important to note that parameters other than those provided by Wilson group do not appear to reasonably estimate the rate coefficients of

2,3-dimethylbutane. Additionally, SAR estimates for several cyclic alkanes (cyclopentane, methylcyclopentane, cyclohexane) and branched alkanes (2,2,4-trimethylpentane) appear to be overestimated compared to our measurements. This raises reasonable suspicion that these methods may still lack consideration of additional factors. Arrhenius expressions for the reaction of 2-Methylhepane and 3-Methylheptane with OH radicals were obtained for the first time in the temperature range of 273-323 K, expanding the existing database. In addition, correlation equations for the rate coefficients of alkanes reacting with OH radicals and chlorine atoms were obtained, and the rate coefficient of 2,2,3-trimethylpentane with chlorine atoms, which has not yet been reported, was deduced. The atmospheric lifetimes of the alkanes were also obtained for further prediction of their environmental impact.

**Line 523: there is no "structure-additivity method"**

**Reply:** Thanks for your valuable suggestions! Modifications have been made in the revised manuscript.

…The method of structure-activity relationship for rate constant estimation is mostly consistent for the prediction of $k_{OH}$ (298 K) for the studied n-alkanes,…

**RC2:**

**1.Major points:**

**Figure 5: There are many problems with this graph.**

**I am assuming that the black lines are Arrhenius fits. As discussed in your review, Arrhenius fits aren't very useful for this set of compounds over a large temperature range (for example, panel D shows just how poorly this approach describes the data), and I would recommend other fits such as k(T) = Aexp(B/T)(T/300)^n or k(T) = AT^n*exp(B/T), where A = A-factor, B = E/R and n = an additional term to provide curvature.**

**The units on the y-axes are wrong (cm^3 molecule^-1 s^-1).**

**There are missing data from the plots:**

**Panel A:**

Anderson et al., 2004 (DOI: https://doi.org/10.1021/jp0472008); Behnke et al., 1988 (DOI: https://doi.org/10.1016/0004-6981(88)90341-1); Nolting et al., 1988 (DOI: https://doi.org/10.1007/BF00048331); Han et al., 2018 (DOI: https://doi.org/10.3390/atmos9080320); Ferrari et al., 1996 (DOI: https://doi.org/10.1002/(SICI)1097-4601(1996)28:8%3C609::AID-KIN6%3E3.0.CO;2-Z)

**Panel B:**

Sivaramakrishnan et al., 2009 (DOI: https://doi.org/10.1021/jp810987u); Pang et al., 2011 (DOI: https://doi.org/10.1524/zpch.2011.0156); Han et al., 2018 (DOI: https://doi.org/10.3390/atmos9080320)

**Panel C:**

Atkinson et al., 1984 (DOI: https://doi.org/10.1002/kin.550160413); Cox et al., 1980 (DOI: https://doi.org/10.1021/es60161a007); Darnall et al., 1978 (DOI: https://doi.org/10.1021/j100503a001); Lloyd et al., 1976 (DOI: https://doi.org/10.1021/j100549a003)

**Panel D:**

Harris and Kerr, 1988 (DOI: https://doi.org/10.1002/kin.550201203); Wilson et al., 2006 (DOI: https://doi.org/10.1021/jp055841c) Also, the work of Sivaramakrishnan et al., 2009 is conducted at high temperature and the other points don't seem to match up with the figure legend.

Reply: Thanks for your valuable suggestions! In the article of Atkinson et al., 1984, (DOI: https://doi.org/10.1002/kin.550160413) there is no mention of isopentane, but only of isobutane. Modifications have been made in the revised manuscript.

[Figure]

**2.Conclusions:**

**The conclusions have not changed since the review. They should be updated to reflect the contents of the revised article (for example, it was already pointed out that your study does not present the first temperature-dependent data for methylcyclopentane, and yet, this is what you claim in your conclusions. This is unacceptable, this error has already been pointed out in your earlier review. Also, you have now applied several SAR methods for rate constant estimation, it would be useful to summarize the findings of these different methods.**

**Reply:** Thanks for your valuable suggestions! Modifications have been made in the revised manuscript.

The use of the multivariate relative rate method in this study allowed for the simultaneous determination of reaction rate coefficients of $C_3$-$C_{11}$ alkanes and OH radicals, which significantly improved the efficiency of determination. A total of 25 relative rate coefficients at room temperature were obtained, including the determination of a previously unreported room temperature relative rate coefficient for

3-methylheptane. For the studied n-alkanes, the obtained rate coefficients ($k_{OH}$) were found to be consistent with results estimated by the SAR methods using parameters provided by various positional groups, such as Atkinson and Kwok, Neeb, Wilson, Jenkin, and McGillen. However, it is important to note that parameters other than those provided by Wilson group do not appear to reasonably estimate the rate coefficients of 2,3-dimethylbutane. Additionally, SAR estimates for several cyclic alkanes (cyclopentane, methylcyclopentane, cyclohexane) and branched alkanes (2,2,4-trimethylpentane) appear to be overestimated compared to our measurements. This raises reasonable suspicion that these methods may still lack consideration of additional factors. Arrhenius expressions for the reaction of 2-Methylhepane and 3-Methylheptane with OH radicals were obtained for the first time in the temperature range of 273-323 K, expanding the existing database. In addition, correlation equations for the rate coefficients of alkanes reacting with OH radicals and chlorine atoms were obtained, and the rate coefficient of 2,2,3-trimethylpentane with chlorine atoms, which has not yet been reported, was deduced. The atmospheric lifetimes of the alkanes were also obtained for further prediction of their environmental impact.

**Minor points:**

**Title:**

**The numbers 3 and 11 should be in subscript. This change should be applied throughout your manuscript (not just the title).**

**Reply:** Thanks for your valuable suggestions! All $C_3$-$C_{11}$ have been revised in the revised manuscript.

**Rate coefficients for the reactions of OH radical with $C_3$-$C_{11}$ alkanes determined by the relative rate technique**

…Rate coefficients for the reactions of OH radicals with $C_3$-$C_{11}$ alkanes were determined using the multivariate relative rate technique…

…Anderson et al. obtained the $k_{OH}$ of $C_2$-$C_8$ several n-alkanes and…

**Abstract:**

**Line 16: the edit is worse than the original. I suggest to change it back to how it was.**

**Reply:** Thanks for your valuable suggestions! Modifications have been made in the revised manuscript.

...A total of 25 relative rate coefficients at room temperature and 24 Arrhenius expressions in the temperature range of 273-323 K were obtained…

**Line 18: kOH. k should be italicized. OH shouldn't be italicized. The authors should carefully go through their manuscript and correct each instance of this problem.**

**Reply:** Thanks for your valuable suggestions! Modifications about all $k_{OH}$ have been made in the revised manuscript.

…The absolute rate method (such as flash photolysis and emission flow et al.) involves calculating the reaction kinetics parameter $k_{OH}$ for organic compounds with OH radicals…

…Anderson et al. obtained the $k_{OH}$ $C_2$-$C_8$ several n-alkanes and cyclic alkanes by the relative technique in the air system at $296 \pm 4$ K….

$k_{OH+n-Hexane}$=4.97×10$^{-12}$, $k_{OH+Cyclohexane}$=6.69×10$^{-12}$, $k_{OH+n-Octane}$=8.48×10$^{-12}$

…For example, the $k_{OH}$ obtained for propane with n-hexane, cyclohexane and n-octane as the reference compound…

**Lines 20 – 26: The authors have employed several SAR methods now, and the results of these calculations should be included in your abstract.**

**Reply:** Thanks for your valuable suggestions! Modifications have been made in the revised manuscript.

Interestingly, whilst results for n-alkanes agreed well with available structure activity relationship (SAR) calculations of Atkinson and Kwok, Neeb, Wilson, Jenkin, and McGillen, the three cyclo-alkanes (cyclopentane, methylcyclopentane, cyclohexane) and one branched alkane (2,2,4-trimethylpentane) were found to be less reactive than predicted by SAR. Conversely, the SAR estimates for 2,3-dimethylbutane

were approximately 25 % lower than the experimental values, with the exception of those estimated by the Wilson group, highlighting the limited understanding of the oxidative chemistry of these compounds.

**Line 35: Remove the word "very".**

**Reply:** Thanks for your valuable suggestions! The "very" has been removed in the revised manuscript.

…The reactivity relation of saturated alkanes with OH radicals and chlorine atoms was obtained: $\log_{10}[k_{(Cl+alkanes)}] = 0.569 \times \log_{10}[k_{(OH+alkanes)}]\text{-}3.111$ ($R^2 = 0.86$)…

**Introduction:**

**Line 68: I am not familiar with the term "some secondary oxides". I recommend "oxygenated molecules".**

**Reply:** Thanks for your valuable suggestions! Combining your and other reviewer's suggestions, modifications have been made in the revised manuscript.

…Additionally, degradation products produced by the oxidation of alkanes can form…

**Lines 71 – 73: This is purely tautological. Essentially, you are stating that you need to measure how fast something reacts in order to evaluate how fast it reacts… Just delete these lines.**

**Reply:** Thanks for your valuable suggestions! Modifications have been made in the revised manuscript.

…….

**Line 90: relies, not relied.**

**Reply:** I'm sorry for the mistake! Modifications have been made in the revised manuscript.

…the relative rate method relies on the recommended rate coefficient for the reaction of a reference…

**Line 98: "method" not "mehod".**

**Reply:** I'm sorry for the mistake! Modifications have been made in the revised manuscript.

…dozens of papers for the rate coefficients of alkanes with OH radical measured by relative rate method have been published…

**Methods:**

**Line 144: "$H_2O_2$ with respect to …".**

**Reply:** Thanks for your valuable suggestions! Modifications have been made in the revised manuscript.

…$H_2O_2$ with respect to VOCs was injected through a three-way valve…

**Line 159: lower-case k for kd.**

**Reply:** Thanks for your valuable suggestions! Modifications have been made in the revised manuscript.

…The $k_d$ values ranged from 1.3 to 4.8 (the units are $\times10^{-4}$ ppbv/h)…

**Estimation of the rate constant at 298 K (SAR):**

**Mostly, this section has been improved, and I think the inclusion of the different methods is interesting. However, it was recommended by my review (referee 1) and that of referee 3 that you should try to compare with the recent SAR method of McGillen et al., 2024 (DOI: https://doi.org/10.1039/D3EA00147D). The reasons for this are that it is a distinctly different methodology to that of Atkinson, Wilson, Neeb, etc. The authors appear not to have been able to include these predictions. They are easy to do, and I list the results here:**

**I list the results here:**

**name k SAR**

**propane 1.20E-12**

**isobutane 2.17E-12**

**n-butane 2.38E-12**

**isopentane 3.37E-12**

**n-pentane 3.67E-12**

**cyclopentane 7.30E-12**

**2,2-dimethylbutane 1.94E-12**

**2,3-dimethylbutane 4.27E-12**

**2-methylpentane 4.65E-12**

**3-methylpentane 4.67E-12**

**methylcyclopentane 8.34E-12**

**2,4-dimethylpentane 5.56E-12**

**cyclohexane 8.76E-12**

**2-methylhexane 5.97E-12**

**3-methylhexane 6.00E-12**

**2,2,4-trimethylpentane 4.10E-12**

**n-heptane 6.40E-12**

**methylcyclohexane 9.76E-12**

**2,3,4-trimethylpentane 6.39E-12**

**2-methylheptane 7.34E-12**

**3-methylheptane 7.36E-12**

**n-octane 7.80E-12**

**n-nonane 9.21E-12**

**n-decane 1.06E-11**

**n-undecane 1.21E-11**

**I recommend that you include these new predictions in this section so that you can make a more complete comparison of the available SAR methods, and to discuss whether or not accounting for the cycle size is a useful parameter in these SARs. Also, why don't you compare the SAR estimates at different temperatures?**

**Reply:** Thanks for your valuable suggestions and help, it will help me a lot!

Modifications have been made in the revised manuscript.

In addition, there are a number of SAR methods that are quite different in their estimation from those of Atkinson, Wilson, et al. and Neeb, et al., for instance, the method of McGillen et al. Figure S3 shows a comparison of our measurements with the SAR estimates of McGillen et al. Similar to the results of Kwok and Atkinson, Neeb, and Jenkin et al., the obtained $k_{OH}$ values of cyclopentane and 2,3-Dimethylbutane in this study exceed the shaded area. This further illustrates that there is still a large discrepancy between the experimental values and the SAR estimates for both substances. For cycloalkanes, the SAR estimates of McGillen et al. are still overestimated to varying degrees compared to our measurements, especially for cyclopentane, where the experimentally measured $k_{OH}$ in this work is still about 34% lower than the SAR estimate. And the $k_{OH}$ values for cyclohexane, methylcyclopentane and methylcyclohexane were also lower than the estimated values by about 18%, 12% and 5%, respectively. For the branched alkanes, again the $k_{OH}$ of 2,3-Dimethylbutane is higher than the SAR estimate by about 32% or so. Similarly to the comparison with the Neeb, and Jenkin et al SAR estimates, the experimental measurements we obtained for 2,2,4-Trimethylpentane are also lower than the McGillen et al estimates by about 14%. By comparing the reaction rate coefficients of cyclopentane and cyclohexane, it is found that for cyclic alkanes of Kwok and Atkinson, Neeb, Jenkin et al., and McGillen et al, the cycle size increases by about $1.41 \times 10^{-12}$ cm$^3$ molecule$^{-1}$ s$^{-1}$. However, For the SAR estimate of Wilson et al, the cycle size increases by about $1.12 \times 10^{-12}$ cm$^3$ molecule$^{-1}$ s$^{-1}$.

Jenkin et al. obtained optimised Arrhenius parameters ($k = A\exp(-(E/R)/T)$) for the group rate coefficients for H-atom abstraction from -CH$_3$, -CH$_2$- and CH< groups of alkanes by SAR method, from this it is possible to derive reaction rate constants for different temperatures.

$k_{prim} = 2.9 \times 10^{-12} \times \exp(-(925)/T)$

$k_{sec} = 4.95 \times 10^{-12} \times \exp(-(555)/T)$

F(-CH$_3$)=1; F(-CH$_2$-)= $\exp(89/T)$

Take n-Octane e as an example:

(1) $k_{prim} \times$ F(-CH$_2$-) $\times 2 = 2.9 \times 10^{-12} \times \exp(-(925)/T) \times \exp(89/T) \times 2$

(2) $k_{sec} \times$ F(-CH$_3$) $\times$ F(-CH$_2$-) $\times 2 + k_{sec} \times$F(-CH$_2$-)$^2 \times 4 = 4.95 \times 10^{-12} \times \exp(-(555)/T) \times 1 \times$

$\exp(89/T) \times 2 + 4.95 \times 10^{-12} \times \exp(-(555)/T)$ [exp(89/T)]$^2 \times 4$

Therefore, the comparison between the estimated values of the SAR method and the experimental values of the present work at different temperatures is shown in the following figure:

[Figure]

SAR estimates are in good agreement with the experimental values for n-octane at different temperatures. In addition, the experimental values of the other components at different temperatures within the error range are in good agreement with the SAR estimates. We added supplement for different temperatures in the revised manuscript.

**Line 448:** Also, the experimental values of n-Octane obtained at different temperatures are in high agreement with the SAR estimates.

**Line 269 (and everywhere else): k SHOULD NOT BE UPPER-CASE!**

**Reply:** Thanks for your valuable suggestions! Modifications have been made in the revised manuscript.

…$k^0_{prim}$ , $k^0_{sec}$ , $k^0_{tert}$ represent the rate coefficients of each -CH$_3$,…

…Atkinson and Kwok et al derived the values of $k^0_{prim}$, $k^0_{sec}$, $k^0_{tert}$ at room temperature,

$k_{\mathrm{prim}}^0$=0.136×10$^{-12}$,  $k_{\mathrm{sec}}^0$=0.934×10$^{-12}$,  $k_{\mathrm{tert}}^0$=1.94×10$^{-12}$, the unit is cm$^3$ molecule$^{-1}$ s$^{-1}$…

$$k(CH_3\text{-}X)=k_{\mathrm{prim}}^0\mathrm{F}(X)$$

$$k(X\text{-}CH_2\text{-}Y)=k_{\mathrm{sec}}^0\mathrm{F}(X)\mathrm{F}(Y)$$

$$k(X\text{-}CH(Y)Z)=k_{\mathrm{tert}}^0\mathrm{F}(X)\mathrm{F}(Y)\mathrm{F}(Z)$$

$$k_{\mathrm{tot}} = \sum [k(CH_3\text{-}X)+k(X\text{-}CH_2\text{-}Y)+k(X\text{-}CH(Y)Z)]$$

**Line 271: "update" not "updated". "modify" not "modified".**

**Reply:** I'm sorry for the mistake! Modifications have been made in the revised manuscript.

…many researchers continued to update and modify some parameters based on…

**Line 272: It is not clear what is meant by fundamental rate constant. Maybe chose "base rate constant" or something like this.**

**Reply:** Thanks for your valuable suggestions! Modifications have been made in the revised manuscript.

…and obtained the new base rate coefficients for different positional groups,…

**Results and discussion:**

**Line 307: There is no such thing as a second-order C-H bond. You mean "secondary" perhaps.**

**Reply:** Thanks for your valuable suggestions! Modifications have been made in the revised manuscript.

…reflects the fact that the main way is to extract the H atom from the secondary C-H bond…

**Line 311: "reactivity increases" not "reactivity increase".**

**Reply:** I'm sorry for the mistake! Modifications have been made in the revised

manuscript.

…can also be seen that the reactivity increases with the increase of cycle size…

**Table 1: This table is mostly OK, but I find it confusing that you have a column entitled "reference" (which should not be confused with your reference rate constant). I suggest to rename this column to "Literature measurements".**

**Reply:** Thanks for your valuable suggestions! Modifications have been made in the revised manuscript.

| Alkanes | Reference | This work | | | Literature measurements |
|---------|-----------|-----------|---|---|-------------------------|
| | | $k_{OH}/k_{reference}$ $\pm 1\sigma$ | $k_{OH}$ $\pm 1\sigma$ ($\times 10^{-12}$ cm$^3$ molecule$^{-1}$ s$^{-1}$) | $k_{OH\text{-}av}{}^a$ $\pm 1\sigma$ ($\times 10^{-12}$ cm$^3$ molecule$^{-1}$ s$^{-1}$) | $k_{OH}$ ($\times 10^{-12}$cm$^3$ molecule$^{-1}$ s$^{-1}$) |
| Propane | n-Hexane | 0.190±0.033 | (9.43±1.66) | (1.01±0.26) | 1.11 [bcd] |
| | Cyclohexane | 0.153±0.028 | (1.03±0.18) | | 1.09 [e] |
| | n-Octane | 0.136±0.031 | (1.16±0.26) | | 1.91 [f] |
| | | | | | (1.15±0.15) [g] |

**Comparisons to structure-activity relationships:**

**Lines 453 – 455: Under the circumstances of this study, I find it strange that you would evaluate the reliability of experimental data by comparing with SAR estimates. If conducted well, an experimental measurement will be considerably more reliable than an estimation technique. Therefore, rather than assessing how reliable the experiment is, more realistically, this comparison assesses how accurate the estimation technique is.**

**Reply:** Thanks for your valuable suggestions! Modifications have been made in the revised manuscript.

To assess the accuracy of the estimation technique, multiple comparisons were made between the obtained reaction rate coefficients and the SAR…

**Line 462: "branched" not "branch".**

**Reply:** I'm sorry for the mistake! Modifications have been made in the revised manuscript.

….For branched alkanes, such as monomethyl branched alkanes….

**Line 464: what is "shadow range"? The meaning is unclear, rephrase.**

**Reply:** I'm sorry for the mistake! Modifications have been made in the revised manuscript.

….the obtained $k_{OH}$ values all fall within the shadow area…

**Line 497: "n-nonane"**

**Reply:** I'm sorry for the mistake! Modifications have been made in the revised manuscript.

(23) n-nonane;

**Temperature dependence (273-323 K):**

**Table 2: similar to Table 1, I suggest to rename this column "Reference" to "Literature measurements".**

**Reply:** Thanks for your valuable suggestions! Modifications have been made in the revised manuscript.

| Alkanes | Temperature (K) | A-factor [a] ($\times 10^{-11}$) | $E_a/R$ [b] (K) | Technique [c] | Literature measurements |
|---|---|---|---|---|---|
| Propane | 273-323 | 2.38±0.90 | 952±110 | RR/DP/GC-FID | this work |
| | 296-908 | 2.71±0.17 | 988±31 | AR/FP/LIF | (Bryukov et al., 2004) |
| | 227-428 | 1.29 | 730 | RR/DP/GC | (Demore and Bayes, 1999) |
| | 233-376 | 1.01 | 660 | AR/FP/LIF | (Talukdar et al., 1994) |

**Figure 6: Similar to Figure 5, there are several errors here. Figure caption contains misspellings of 3-methylheptane and 2-methylheptane.**

**Reply:** Thanks for your valuable suggestions! Modifications have been made in the revised manuscript.

Figure 6. Arrhenius plots for the reaction of Methylcyclopentane (a), 2-Methylhexane (b), 3-Methylheptane (c), 3-Methylhexane (d) and 2-Methylheptane (e) with OH radical along with available literature data…

**Panel A: The work of Anderson was conducted at 296 K. The high temperature measurements were conducted by Sivaramakrishnan and Michael, 2008 (so your legend is incorrect).**

**Reply:** I'm sorry for the mistake! Modifications have been made in the revised manuscript.

[Figure]

**Correlation between the rate coefficients of the reaction of alkanes with OH 672 radicals and chlorine atoms:**

**Line 677: replace "discrete" with "weak"?**

**Reply:** Thanks for your valuable suggestions! Modifications have been made in the revised manuscript.

…Although the correlation between propane and isobutane is relatively weak, the reactivity of…

**Atmospheric lifetime and implications:**

**Lines 693 – 694: The reference Li et al., 2018 did not assess the atmospheric**

**abundance of OH radicals. Choose a relevant reference for this point.**

**Reply:** I'm sorry for the mistake! Modifications have been made in the revised manuscript.

Lawrence, M. G., Jöckel, P., and von Kuhlmann, R.: What does the global mean OH concentration tell us?, Atmospheric Chemistry and Physics, 1, 37-49, https://doi.org/10.5194/acp-1-37-2001, 2001.

**Line 696: "in" not "on the". "lifetimes" not "lifetime".**

**Reply:** Thanks for your valuable suggestions! Modifications have been made in the revised manuscript.

… As can be seen from the table, the atmospheric lifetimes of $C_3$-$C_{11}$ alkanes reacting with OH radicals are about 1-11 days…

**Line 698: "lifetimes are reduced" not "lifetime seems to reduce".**

**Reply:** I'm sorry for the mistake! Modifications have been made in the revised manuscript.

…As the carbon chain grows, the atmospheric lifetimes are reduced, especially…

**Lines 701 – 702: RO$_2$ will serve to convert NO to NO$_2$ directly. The role of HO$_2$ is also important, but is not directly related to the peroxy radicals mention here.**

**Reply:** Thanks for your valuable suggestions! Modifications have been made in the revised manuscript.

…They are emitted into the air and degraded quickly to generate alkyl radicals, which are immediately converted into alkyl peroxy radicals by reacting with abundant $O_2$ in the atmosphere. Alkyl peroxyl radicals will serve to convert NO to $NO_2$ directly, leading to the production of tropospheric ozone….

**Line 703: 11 days is not a "long time".**

**Reply:** Thanks for your valuable suggestions! I sincerely apologize that the expression is not accurate enough here. Modifications have been made in the revised manuscript.

…Longer atmospheric residence time of short-chain alkanes compared to long-chain C8-C11 alkanes, such as propane…

**Lines 703 – 705: This is tautological in its present form, and should be removed.**

**Reply:** Thanks for your valuable suggestions! The tautological portions have been removed in the revised manuscript.

**Line 710: previous temperature dependent data exists for methylcyclopentane. You must change this accordingly.**

**Reply:** Thanks for your valuable suggestions! Modifications have been made in the revised manuscript.

…Arrhenius expressions for the reaction of 2-Methylhepane and 3-Methylheptane with OH radicals were obtained for the first time in the temperature range of 273-323 K,…

**Lines 716 – 717: is this true of all SAR techniques or just one of them?**

**Reply:** Thank you very much for your query, which is addressed here for the above 4 SAR techniques and the conclusions have been modified for a clearer summary.

**Line 720: More than one method has been employed. The use of the singular "method" is inappropriate and should be updated.**

**Reply:** I'm sorry for the mistake! Modifications have been made in the revised manuscript.

…This raises reasonable suspicion that these methods may still lack consideration of additional factors.

RC3:

**1. Minor: It is highly recommended the authors to preferably use rate coefficient instead of rate constant. k is not a constant - although the term is used in the literature - and they have clearly described that depends on T! This comment was also present in the first review.**

**Reply:** Apologies for not being able to complete all the changes to rate constant in the last revision! All rate constant has been replaced with rate coefficient in the revised manuscript.

…Notably, a new room temperature relative rate coefficient for 3-methylheptane that had not been previously reported was determined…

…By monitoring the simultaneous decay of the target and reference compounds in the presence of OH radicals due to competitive response mechanisms, the rate coefficient for the reaction of OH radicals with the target compound….

**2. Major: In absolute and relative rate methods comparison the authors should be very careful! They use the term known, although they should use the term recommended. The fact that a reaction rate cofficient has been measured doesn't mean that is known! It deppends on them whst k-values they would use for their reference reaction used, but they need to justify why they used those. It is strongly recommended to use the evalusted rate coefficients values from NASA/JPL and IUPAC panels, where exist and if not they need to do their own evaluation and justify their selection. Please keep in mind that absolute rate coefficients determinations although sometimes might be more demanding and challenging do not rely on other measurements and thus are not subzected on systematic errors of previous measurements, that are larger at temperatures away from room temperature. Particularly when so many reaction rate coefficients are measured at once it is very likely that reaction products etc might interfere in their subtraction analysis and the error bars that should be quoted cannot be less than 15 %. Like mentioned in the previous review, the error analysis of the kinetic data presented in this work is actually missing! What are the estimated systematic**

**uncertainties of authors measurements? How they have been estimated?**

**Reply:** Thanks for your valuable suggestions! Modifications have been made in the revised manuscript.

(1) …Unlike the absolute rate method, the relative rate method relied on the recommended rate coefficie for the reaction of a reference…(Line 60)

(2) …The basic principle is that the rate coefficient for the reaction of the reactant used as a reference with OH radicals needs to be the recommended rate coefficients values, (Line 137)

(3) The rate constant of the reference compound we used was not found in the NASA/JPL and IUPAC database, so we chose the expert evaluated data from McGillen et al.

(4) The ratio $k/k_{ref}$ is derived from ln–ln plots. The error, $\sigma$, is calculated as the standard error based on the product of $k/k_{ref}$ obtained in several experiments and the $k_{ref}$ recommended in the literature.

The derived rate coefficients are weighted average of the obtained data with various references taking into account of the uncertainties on the references rate coefficients values as: $k_{av} = (w_{ref1}k_{ref1}+w_{ref2}k_{ref2}+…)/ (w_{ref1}+w_{ref2}+…)$, where $w_{ref1}=1/\sigma_{ref1}^2$, etc. The error, $\sigma_{av}$, was given by: $\sigma_{av} = (1/\sigma_{ref1}+1/\sigma_{ref2}+…)^{-0.5}$ (Farrugia et al., 2015): unit of $cm^3$ $molecule^{-1}$ $s^{-1}$.

**3. Major: Figure 5: (a) First please use the same symbols in all 4 plots when you refer to the same literature study, for consistency purposes. For instance, for Atkinson et al. (2003) you have used three different symbols and colors, which makes the comparison confusing for the reader. (b) Include temperature as mirror to bottom axis, so as the reader to have a direct access to the temperature that the kinetic data refer to just lookin the plot. (c) do the 2σ error bars include systematic uncertainties? How the error bars shown in figure 5c, for instance, were determined? Why the error bar at 273 K is that smaller compared to 296 K and then becomesd that smaller again at 313 K? Systematic uncertainties normally**

**become larger away from room temperature, where most of the test measurements are carried out. This plot doesn't seem to include the systematic uncertainties from the reference reactions or they have mistakenly included. If this is tonly the 2σ precision of the fits, why the present measurements have so large error limits? (d) What are the fits shown in all 4 plots. First they do not fit the data! Why the authors didn't try to fit the actual data with a modified Arrhrnius expression? To present fits that are not representing this work or combined with literature measured data seems odd. The easiest way around it is to fit their data and discuss the agrrement with other literature data in the same temperature data. The k(T)-trends observed in these plots might contain important mechanistic information that the authors have disregarded!**

**Reply:** Thanks for your valuable suggestions! Modifications have been made in the revised manuscript.

**(a)** All the symbols and colours in all 4 plots have been harmonized in the revised manuscript.

**(b)** Temperature is added as a mirror image of the bottom axis in the figure.

**(c)** 2σ in the figure is the error for each temperature, which is calculated as the standard error based on the product of $k/k_{ref}$ obtained in several experiments and the $k_{ref}$ recommended in the literature. In addition, we returned to the original spectrogram and reprocessed the data, the corrected data are shown in Figure 5(c).

**(d)** We obtained the Arrhenius expression for the range 273-323 K by performing experiments in the temperature range under study (273-323 K). In addition we fitted again to the high temperature data from the literature based on the comments of other reviewers, assigning curvature to the data as it varied and fitting it with a new expression $(k(T) = A*exp(B/T)*(T/300)^n)$ to obtain the actual expression.

---

## Author Response (AR3)

**To editor:**

1.  Line 14: Please change 'values' to 'value'.

**Reply:** Thanks for your valuable suggestions! Modifications have been made in the revised manuscript.

Line 14…and the obtained $k_{OH}$ value (in units of…

Line 26: Please change 'previous' to 'previously'.

**Reply:** Thanks for your valuable suggestions! Modifications have been made in the revised manuscript.

Line 27: …the rate coefficients for the 24 previously studied OH + alkanes…

Line 41-42: please the text to the following: "less reactive with $NO_3$ and with ozone, and thus they are degraded…

**Reply:** Thanks for your valuable suggestions! Modifications have been made in the revised manuscript.

Line 41-42: …the alkanes are extremely less reactive with $NO_3$ and with ozone, and thus they are degraded and removed from the atmosphere….

Line 44: Please change 'will form a' to 'can lead to'.

**Reply:** Thanks for your valuable suggestions! Modifications have been made in the revised manuscript.

Line 44: These oxidation processes can lead to photochemical …

Line 59: Please change "in the carbon monoxide, He and $N_2$ system" to "in carbon monoxide, He and $N_2$ systems"

**Reply:** Thanks for your valuable suggestions! Modifications have been made in the revised manuscript.

Line 57: …with selected alkanes in carbon monoxide, He and $N_2$ systems, respectively.

Line 72: Please insert the word 'and', "alkanes, and more complex …"

**Reply:** Thanks for your valuable suggestions! Modifications have been made in the revised manuscript.

Line 70: …alkanes, and more complex and multifunctional alkanes…

Line 76-77: Please change current text to "Perry et al found that the rate coefficients of n-butane increased by 72% as the temperature…"

**Reply:** Thanks for your valuable suggestions! Modifications have been made in the revised manuscript.

Line 73-74: Perry et al found that the rate coefficients of n-butane increased by 72% as the temperature…

Line 81: Please change "Finlaysonpitts" to "Finlayson-Pitts"

**Reply:** Thanks for your valuable suggestions! Modifications have been made in the revised manuscript.

Line 78: Finlayson-Pitts et al., 1993

Line 135-139: I suggest changing this to read as follows: "The basic principle is that the rate coefficient for reaction of a reference compound with OH needs to be well established; then, the rate coefficient for the target compound can be determined by monitoring the …"

**Reply:** Thanks for your valuable suggestions! Modifications have been made in the revised manuscript.

Line 137-140: The basic principle is that the rate coefficient for reaction of a reference compound with OH needs to be well established; then, the rate coefficient for the target compound can be determined by monitoring the simultaneous decay of the target and reference compound…

Line 165-166: I recommend changing the current text to the following: "In this work, three different commonly used reference compounds (n-hexane, cyclohexane and noctane) were used to determine the …"

**Reply:** Thanks for your valuable suggestions! Modifications have been made in the revised manuscript.

Line 166-168: …In this work, three different commonly used reference compounds (n-hexane, cyclohexane and n-octane) were used to determine the rate coefficients for each reaction at room temperature to check the consistency…

Line 177: Please change 'reflected' to 'presented'.

**Reply:** Thanks for your valuable suggestions! Modifications have been made in the revised manuscript.

Line 180: … temperatures is presented in Sec. 3.3.

Line 216-218: I suggest the following text: "new base rate coefficients for different positional groups, and also developed independent methods for rate coefficient estimation. Some examples include: …"

**Reply:** Thanks for your valuable suggestions! Modifications have been made in the revised manuscript.

Line 214-216: …and obtained the new base rate coefficients for different positional groups, and also developed independent methods for rate coefficient estimation. Some examples include: …

Line 244: Please change 'reflect' to 'reflecting'.

**Reply:** Thanks for your valuable suggestions! Modifications have been made in the revised manuscript.

Line 244: …reflecting the fact that the main way is…

Line 254: Please change 'each increase' to 'addition'.

**Reply:** Thanks for your valuable suggestions! Modifications have been made in the revised manuscript.

Line 253: …for cycloalkanes with addition of methyl.

Line 276: Please change 'in the air gas' to 'in air'.

**Reply:** I'm sorry for the mistake! Modifications have been made in the revised manuscript.

Line 275: …the average rate coefficient obtained  is…

Line 289-290: Please change 'Same for Nonane' to "the same applies for Nonane…"

**Reply:** Thanks for your valuable suggestions! Modifications have been made in the revised manuscript.

Line 290: The same applies for Nonane, consistency with previous studies is less…

Line 306: Please change 'Like' to 'For example'.

**Reply:** Thanks for your valuable suggestions! Modifications have been made in the revised manuscript.

Line 307: …by about 5%-16%. For example, the relative values measured by…

Line 312: Please change 'are excellent agreement' to 'are in excellent agreement'.

**Reply:** Thanks for your valuable suggestions! Modifications have been made in the revised manuscript.

Line 313: …methylcyclohexane are in excellent agreement…

Line 351: Please change '2,3-dimethylbutane, the experimental' to '2,3-dimethylbutane, where the experimental'.

**Reply:** Thanks for your valuable suggestions! Modifications have been made in the revised manuscript.

Line 352: …like 2,3-Dimethylbutane, where the experimental data was about…

Line 362: I suggest the following text: "…these compounds is still limit3d, and that further data and analysis for alkanes with this structure are needed."

**Reply:** Thanks for your valuable suggestions! Modifications have been made in the revised manuscript.

Line 362: The results indicated that our understanding for the oxidation chemistry of these compounds is still limited, and that further data and analysis for alkanes with this structure are needed.

Line 370: Please change 'cycle-chain' to 'cyclic'

**Reply:** Thanks for your valuable suggestions! Modifications have been made in the revised manuscript.

Line 370: …activity of these cyclic alkanes estimated…

Line 387-388: I suggest the following text: "the increase in cycle size increases k by about 1.41 x 10-12 cm3 molecule-1 s-1. However for the SAR estimate of Wilson et al., the increase is about 1.12 x 10-12…"

**Reply:** Thanks for your valuable suggestions! Modifications have been made in the revised manuscript.

Line 387-388: …it is found that for cyclic alkanes of Kwok and Atkinson, Neeb, Jenkin et al., and McGillen et al, the increase in cycle size increases $k$ by about $1.41\times10^{-12}$ $cm^3$ molecule$^{-1}$ s$^{-1}$. However, for the SAR estimate of Wilson et al, the increase is about $1.12\times10^{-12}$ $cm^3$ molecule$^{-1}$ s$^{-1}$.

Line 416: Please change 'no' to 'not'.

**Reply:** Thanks for your valuable suggestions! Modifications have been made in the revised manuscript.

Line 419: …dependence data has been less or not studied…

Line 420: Can the authors adjust Table 2 somehow to make it more clear which data entries belong to which compound?

**Reply:** Thanks for your valuable suggestions! Modifications have been made in the revised manuscript.

Line 442: Please change "slightly' to 'reasonably'.

**Reply:** Thanks for your valuable suggestions! Modifications have been made in the revised manuscript.

Line 442: This result is reasonably consistent with…

Line 468-469: I do not know what is meant by 'finger' in this sentence. I think this word can be deleted in both places.

**Reply:** Thanks for your valuable suggestions! Modifications have been made in the revised manuscript.

Line 460-461: …the activation energy (Ea/R) obtained in this work is more in line with that of (148), however, the factor A obtained (4.48) is about 17% higher than that (3.84).

Line 538: Please delete the words 'is this data'.

**Reply:** Thanks for your valuable suggestions! Modifications have been made in the revised manuscript.

Line 528: …from the figure that this data is significantly lower by…

Line 556: I think the words "to the saturated alkanes series' can be deleted.

**Reply:** Thanks for your valuable suggestions! "to the saturated alkanes series' have been deleted in the revised manuscript (Line 547).

Line 608: Please delete the word "positional'.

**Reply:** Thanks for your valuable suggestions! "positional' have been deleted in the revised manuscript (Line 589).

Line 614: 2-methylheptane. (There is a t missing).

**Reply:** I'm sorry for the mistake! "2-methylheptane' have been revised in the revised manuscript (Line 597).

**To RC1:**

1.  Minor considerations:

Lines 20-21: in general, our understanding of alkane oxidation is good (relative to other types of compounds). I would suggest to rephrase this to something like: "… highlighting that there may be additional factors that govern the reactivity of highly branched alkanes that are not captured by current SAR techniques."

**Reply:** Thanks for your valuable suggestions! Modifications have been made in the revised manuscript (Lines 21-22).

…highlighting that there may be additional factors that govern the reactivity of highly branched alkanes that are not captured by current SAR techniques…

2.  General:

some of the new edits contain typographical and grammatical errors. An example of this would be lines 41-43, which is considerably worse than the sentence that it replaces. I won't concern myself any further with these small details, since I assume they will be fixed during the finalization of your manuscript. However, I would suggest to the authors that hasty edits such as this can degrade the quality of your presentation and distract your readers.

**Reply:** I'm sorry for the mistake! Typographical and grammatical errors have been revised in the revised manuscript. For example, the sentence in lines 41-43 has been modified as follows:

In the troposphere, the alkanes are extremely less reactive with $NO_3$ and with ozone, and thus they are degraded and removed from the atmosphere via gas-phase oxidation reactions with OH radicals and chlorine atoms.

3. Section 3.2, Figure 4: Firstly, why don't you include the predictions of McGillen et al. (2024) in this plot? It seems strange to put it in the SI. Secondly, it appears that there are fewer predictions of Wilson et al. (2006) (18 compared with 25 for the other methods). Is this a limitation of the method, or is this a mistake?

**Reply:** Thanks for your valuable suggestions! The predictions of McGillen et al. (2024)

in Figure 4 in the revised manuscript. However since the predictions of Wilson et al. include only the 18 alkanes listed in the figure, and the other 7 alkanes have not been studied, only a comparison of these 18 alkanes is included in the figure.

[Figure]

Figure 4. Measured Alkanes + OH rate coefficients plotted against SAR-derived rate coefficients for all compounds (a. (Kwok and Atkinson, 1995); b. (Neeb, 2000); c. (Jenkin et al., 2018); d. (Wilson et al., 2006); e. (McGillen et al., 2020)). The shaded

area demonstrates a 20 % uncertainty in the 1:1 black gradient line. The alkanes represented by serial number can be identified as follows: (1) Propane; (2) Isobutane; (3) n-Butane; (4) Isopentane; (5) n-pentane; (6) Cyclopentane; (7) 2,2-Dimethylbutane; (8) 2,3-Dimethylbutane; (9) 2-Methylpentane; (10) 3-Methylpentane; (11) Methylcyclopentane; (12) 2,4-Dimethylpentane; (13) Cyclohexane; (14) 2-Methylhexane; (15) 3-Methylhexane; (16) 2,2,4-Trimethylpentane; (17) n-Heptane; (18) Methylcyclohexane; (19) 2,3,4-Trimethylpentane; (20) 2-Methylheptane; (21) 3-Methylheptane; (22) n-Octane; (23) n-nonane; (24) n-Decane; (25) n-Undecane.

4. Line 468: finger activation energy? What is this?

**Reply:** I'm sorry for the mistake! Modifications have been made in the revised manuscript. The 'finger activation energy' has been changed to 'activation energy'.

Line 460-461: …the activation energy (Ea/R) obtained in this work is more in line…

5. Figure 7: To be consistent with the other figures, it would be useful to number the points in this graph according to the compound (e.g. (1) = propane, (2) = isobutane, etc.).

**Reply:** Thanks for your valuable suggestions! Modifications have been made in the revised manuscript.

[Figure]

Figure 7. Log-log plot of the rate coefficients for the reaction of Cl-atoms versus the reaction of OH radicals with the saturated alkanes ($C_3$-$C_{11}$ alkanes studied above). The solid line represents the unweighted least-squares fit to the data. The alkanes represented by serial number can be identified as follows: (1) Propane; (2) Isobutane; (3) n-Butane; (4) Isopentane; (5) n-pentane; (6) Cyclopentane; (7) 2,3-Dimethylbutane; (8) 2-Methylpentane; (9) 3-Methylpentane; (10) Methylcyclopentane; (11) 2,4-Dimethylpentane; (12) Cyclohexane; (13) 2-Methylhexane; (14) 2,2,4-Trimethylpentane; (15) n-Heptane; (16) Methylcyclohexane; (17) n-Octane; (18) n-nonane; (19) n-Decane; (20) n-Undecane.

**To RC2:**

1. The authors are advised to revise the errors for average values in Table 1 since there are used the weighted average approach and not sigma.

   **Reply:** Thanks for your valuable suggestions! Modifications have been made in the revised manuscript.

   $\sigma$ has been replaced with $\theta$ in Table 1.

   where $w_{ref1}=1/\theta_{ref1}^{2}$, etc. The error, $\theta_{ref}=\sqrt{\dfrac{\Sigma(x_i-\bar{x})^2}{n}}$, $\theta_{av}=(1/\theta_{ref1}+1/\theta_{ref2}+\ldots)^{-0.5}$.

2. Please also expand the conclusions with more information from Figure 4,5 and add conclusions about Figure 7.

   **Reply:** Thanks for your valuable suggestions! More information from Figure 4,5 and Figure 7 have been added in the conclusion in the revised manuscript.

   Line599: expanding the existing database. And the rate coefficients do not change significantly in this temperature range, especially for 2-methylheptane. Whilst the value of preexponential factor A increases with the increase of the number of carbon atoms, which is consistent with the law of its reactivity. In addition, correlation equations for the rate coefficients of alkanes reacting with OH radicals and chlorine atoms were obtained, and the rate coefficient of 2,2,3-trimethylpentane with chlorine atoms, which has not yet been reported, was deduced.

**To RC3:**

1. In several places when they refer to other kinetic studies, e.g., Introduction, line 56 "…in the Ar system at 300 K using…" they should include the total pressure and not only the bath gas, e.g., "at 300 K and xxx Torr in Ar, using…". Another example is in line 70, but it also appears in several places that it is suggested the authors to change it and get rid of Ar or Air system.

   **Reply:** Thanks for your valuable suggestions! Modifications have been made in the revised manuscript.

   …Greiner measured the first kinetic data for the reaction of OH radicals with three alkanes at 300 K and 28-149 Pa in the Ar system using….(line 55)

   …Shaw et al. and Phan and Li obtained rate coefficients of a series of alkanes  (Phan and Li, 2017; Shaw et al., 2018; Shaw et al., 2020). (line 66)

   …Anderson et al. obtained the $k_{OH}$ of C$_2$-C$_8$ several n-alkanes and cyclic alkanes by the relative technique  at 296 ± 4 K. (line 68)

   For several n-alkanes, the average rate coefficient obtained  are…(line 275)

2. Correct typo in line 42 "ano" and change to "are no", replace "the absolute rate constant method and the relative rate constant method" with "both absolute and relative rate methods" Change "et al." in line 53 with etc., line 72 replace "solely" with "limited to", in line 77 replace "multiplied by" with "increased by", in line 79 change "in varying degrees at 300-390 K" with "varying the temperature in the range 300 – 390 K", Reword the paragraph between lines 84 and 87, correct the use of "respectively" in several places (e.g., line 300 – 3 numbers for one compound) and these are only very few examples of editing needed and make difficult for the reader to follow the messages that the authors try to pass across.

   **Reply:** I'm sorry for the mistakes! Modifications have been made in the revised manuscript.

   Line 42: …the alkanes are extremely less reactive with NO$_3$ and with ozone, and thus they are degraded and removed from the atmosphere…

   Line 50: …using both absolute and relative rate methods.

Line 51: …flash photolysis and emission flow etc.

Line 69: …the majority of experiments were conducted limited to on $C_2$-$C_6$ alkanes…

Line 74: …rate coefficients of n-butane increased by 72%....

Line 76: …also increased varying the temperature in the range 300 - 390 K.

Line 81-84: In addition, there is another alkane (e.g., 3-methylheptane) for which only two or fewer measurements of OH radical rate coefficients have been reported in the above temperature range,…

Line 300-301: For the cycloalkanes, like cyclopentane, the average rate coefficient is 4.82±0.27,…

3.  Please replace the expert-evaluated data use with evaluated data throughout the manuscript.

**Reply:** Thanks for your valuable suggestions! Modifications have been made in the revised manuscript.

Line172: Here we used the recommended evaluated data of database for…

Line272: …in different bath gases ($N_2$, Air, $O_2$) with evaluated data at 298±1 K…

Line283: …and the evaluated data (2.36) of McGillen et al.'s…

…

4. Figure 1: a more comprehensive, but brief, setup description needed here, e.g., describe the acronyms.

**Reply:** Thanks for your valuable suggestions! Modifications have been made in the revised manuscript.

Figure 1. A schematic of the experimental device (MFC:Mass Flow Controlle; UV: ultraviolet lamp)

5. Lines 109-110: The way that the tests that the authors carried out are described is deceptive and confusing. $N_2$ does not react with NMHCs, so I guess that the authors want maybe to imply that they tried to ensure that no wall-loss (e.g., hydrolysis) of NMHC occurs under dark conditions and that NMHCs are stable when exposed at 254 nm. They should not present the tests here in reaction forms. Simply state what they

did.

**Reply:** Thanks for your valuable suggestions! Modifications have been made in the revised manuscript.

…Initial conditions of the different species introduced into the reactor for each experiment are outlined in Table S1 in the Supplementary Material. By varying the presence of $H_2O_2$, turning on/off the light, a series of observations were generated such as $N_2$ + NMHCs + dark reaction, $N_2$ + NMHCs + hv (254 nm), and $N_2$ + NMHCs + $H_2O_2$ + dark reaction. to ensure no wall-loss of NMHCs under dark conditions and their stability when exposed to 254 nm light.

6. Lines 135 – 139: The whole sentence needs to be rephrased. The "basic principle" of the RR method is that the monitoring of the relative loss of two reactants when they both only react with the under study radical and one of the two rate coefficients is well established can lead to the determination of the other reaction rate coefficient (reaction of interest).

**Reply:** Thanks for your valuable suggestions! Modifications have been made in the revised manuscript.

Lines 137 – 140: The basic principle is that the rate coefficient for reaction of a reference compound with OH needs to be well established; then, the rate coefficient for the target compound can be determined by monitoring the simultaneous decay of the target and reference compounds in the presence of OH radicals due to the competitive response mechanism.

7. Lines 141 – 142: The statement here is not true in most of the cases! The product of the rate coefficient and the OH concentration needs to be substantially larger (100 times or more) compared to all the rest of the competitive reactions of the reactant with the other radicals X (kx [X]).

**Reply:** Thanks for your valuable suggestions! The description of this part has been removed in the revised manuscript.

Lines 144 – 145: To ensure that the reactants only react with OH radicals, the OH

8. Lines 172 – 175: Here significant editing is needed (rate constants was used instead of rate coefficient – also in 2.2 section title), and the next sentence is not needed at all! It is well known that k depends on temperature!

**Reply:** Thanks for your valuable suggestions! Modifications have been made in the revised manuscript.

…the $k$ values (in units of $cm^3$ molecule$^{-1}$ s$^{-1}$) of the three reference compounds selected respectively are evaluated rate coefficients: $k_{OH+n\text{-}Hexane}$=4.97×10$^{-12}$, $k_{OH+Cyclohexane}$=6.69×10$^{-12}$, $k_{OH+n\text{-}Octane}$=8.48×10$^{-12}$, which is fitted or manually entered data from multiple sources. A detailed explanation at different temperatures is presented in Sec. 3.3.( Lines 175 – 180)

9. Line 224: More important than reporting the R-squares is to present and discuss the intercepts of the linear fits, that is totally missing. This can reveal potential systematic errors and possible curvatures in their data.

**Reply:** Thanks for your valuable suggestions! The intercept case of the linear fits has been added in the revised manuscript.

Line 223: …As shown in Fig. 2, the decay of both target and reference compounds correlated well with eq. (7), the intercepts of the linear fits were close to 0 and high correlation coefficients ($R^2$) were observed for most alkanes, exceeding 0.99.

10. Figure 3: In this plot any systematic divergence from linear behavior cannot be assessed by the reader. The same applies in the range of the change in the Y-axis, so as the reader to evaluate the sensitivity of the measurements. Although it is difficult to display it in figures like those (displaced), the authors can add a comparison between dX and dY change in the text, to enable the reader to assess measurements sensitivity, pinpoint the cases that dX and dY differ significantly and refer to figure 3.

Line235: agreement, errors (dY) between 0.12 and 0.41, the units are 10$^{-12}$cm$^3$ molecule$^{-1}$ s$^{-1}$….

11. Line 274 – 279: Difficult to follow, please rephrase and change "within a certain range" in line 281 with "taking into account the experimental uncertainties"

**Reply:** Thanks for your valuable suggestions! Modifications have been made in the revised manuscript.

Line 274 – 279: The obtained $k_{OH}$ values for $C_3$-$C_{11}$ alkanes in this work were compared with literature-reported values (Table 1). For several n-alkanes, such as n-butane, the average rate coefficient obtained is (2.63±0.23), the unit is $10^{-12}$ $cm^3$ molecule$^{-1}$ s$^{-1}$ (all units in this paragraph are $10^{-12}$ $cm^3$ molecule$^{-1}$ s$^{-1}$). The result is highly consistent with the value (2.56±0.25) obtained by Greiner (Greiner, 1970a) and the value (2.72±0.27) obtained by Perry et al. (Perry et al., 1976), with a consistency of 3% or better.

Line 281: …they still exhibit consistency taking into account the experimental uncertainties.

12. Section 3.2: There is no physical interpretation for the observed discrepancies both between this work and other experimental ones (e.g., 355 - 356), as well as between this work and SAR. The extensive experimental study the authors carried out should have led them to some conclusions about the sources of the observed discrepancies in SAR approaches. This would be of a particular value and could lead to SAR improvements. However, a comprehensive explanation is totally missing which is the main weakness of the manuscript.

**Reply:** Thanks for your valuable suggestions! The physical interpretation for the observed discrepancies has been added in the revised manuscript.

Line 389: …It is worth considering that the presence of ring strain can influence the kinetics of H-atom abstraction in cyclic alkanes, leading to an overestimation of reaction rate coefficients when using unadjusted F(-CH2-). And hence, further data and analysis for F(-CH2-) with these cyclic alkanes are needed.

13. Line 403: This is not THE tropospheric temperature range. Please change to "temperatures relevant to the troposphere".

**Reply:** Thanks for your valuable suggestions! Modifications have been made in the revised manuscript.

Line 405: …were carried out in this study in the temperatures relevant to the troposphere (273-323 K),

14. In several cases, e.g. line 459, 466, 513, 516 etc. "*" was used in Arrhenius expressions instead of "x".

**Reply:** I'm sorry for the mistakes! Modifications have been made in the revised manuscript.

15. Figure 5: Please describe what the fits represent and change the insets since the fit are NOT linear regression! Describe in figure captions what expressions were used.

**Reply:** I'm sorry for the mistakes! Modifications have been made in the revised manuscript.

[Figure]

Figure 5. Arrhenius plots for the reaction of n-Octane (a), n-Heptane (b), Isopentane (c) and 2,3-Dimethylbutane (d) with OH radical in wide temperature range along with available literature data. The error bar was taken as 2σ. The expression for the non-linear curve fits takes the form $k(T) = A \times \exp(Ea/RT) \times (T/300)^n$, and the expression for linear regression takes the form $k(T) = A \times \exp(Ea/RT)$

16. Figure 6: See comment for figure 5 (15).

[Figure]

Figure 6. Arrhenius plots for the reaction of Methylcyclopentane (a), 2-Methylhexane (b), 3-Methylheptane (c), 3-Methylhexane (d) and 2-Methylheptane (e) with OH radical along with available literature data. The error bar was taken as 2σ. The expression for the non-linear curve fits takes the form $k(T) = A \times exp(Ea/RT) \times (T/300)^n$, and the expression for linear regression takes the form $k(T) = A \times exp(Ea/RT)$.

17. Figure 7. log-log plot instead of double log plot. However, it would have been more of value and would have made more sense to present the axes scale in log basis and not the actual values, so as the reader to directly compare Cl and OH rate coefficients as well. This way the values would represent the rate coefficients.

**Reply:** Thanks for your valuable suggestions! Modifications have been made in the revised manuscript.

[Figure]

Figure 7. Log-log plot of the rate coefficients for the reaction of Cl-atoms versus the reaction of OH radicals with the saturated alkanes ($C_3$-$C_{11}$ alkanes studied above). The solid line represents the unweighted least-squares fit to the data. The alkanes represented by serial number can be identified as follows: (1) Propane; (2) Isobutane; (3) n-Butane; (4) Isopentane; (5) n-pentane; (6) Cyclopentane; (7) 2,3-Dimethylbutane; (8) 2-Methylpentane; (9) 3-Methylpentane; (10) Methylcyclopentane; (11) 2,4-Dimethylpentane; (12) Cyclohexane; (13) 2-Methylhexane; (14) 2,2,4-

Trimethylpentane; (15) n-Heptane; (16) Methylcyclohexane; (17) n-Octane; (18) n-nonane; (19) n-Decane; (20) n-Undecane.

18. Line 569: Correct "The atmospheric lifetime of alkanes in the troposphere can be estimated" with "The atmospheric lifetime of alkanes in the troposphere, due to their reaction with OH radicals, can be estimated".

**Reply:** Thanks for your valuable suggestions! Modifications have been made in the revised manuscript.

Line 566: The atmospheric lifetime of alkanes in the troposphere, due to their reaction with OH radicals, can be estimated using the following formula…

19. Conclusions: It would be of value to include potent weaknesses of SAR methods so as to be able to evaluate them and incorporate them in the future.

**Reply:** Thanks for your valuable suggestions! The biggest problem in the current estimation of SAR methods is the overestimation of the rate coefficients for the reaction of cyclic alkanes with OH radicals, which is also reflected in the conclusions.

Line595: …This raises reasonable suspicion that these methods may still lack consideration of additional factors. and a more appropriate empirical ring strain factor needs to be derived based on broader range of experimental data from monocyclic hydrocarbons in the future….

20. Supporting Information: Include typical RR plots at different temperatures so as the reader to be able to evaluate the quality of the measured data.

[Figure]

Figure S3. Typical kinetic data as acquired with the multivariate relative rate technique at 273 K for the reaction of the alkanes with the OH radical using n-hexane as reference compound. Numbers in parentheses are intercepts.

[Figure]

Figure S4. Typical kinetic data as acquired with the multivariate relative rate technique at 283 K for the reaction of the alkanes with the OH radical using n-hexane as reference compound. Numbers in parentheses are intercepts.

[Figure]

Figure S5. Typical kinetic data as acquired with the multivariate relative rate technique at 293 K for the reaction of the alkanes with the OH radical using n-hexane as reference compound. Numbers in parentheses are intercepts.

[Figure]

Figure S6. Typical kinetic data as acquired with the multivariate relative rate technique at 303 K for the reaction of the alkanes with the OH radical using n-hexane as reference compound. Numbers in parentheses are intercepts.

[Figure]

Figure S7. Typical kinetic data as acquired with the multivariate relative rate technique at 313 K for the reaction of the alkanes with the OH radical using n-hexane as reference compound. Numbers in parentheses are intercepts.

[Figure]

Figure S8. Typical kinetic data as acquired with the multivariate relative rate technique at 323 K for the reaction of the alkanes with the OH radical using n-hexane as reference compound. Numbers in parentheses are intercepts.

---

## Author Response (AR4)

**To editor:**

1. Line 37: Change 'to' to 'of'.

**Reply:** Thanks for your valuable suggestions! Modifications have been made in the revised manuscript.

Line 36 …constitute a significant portion of VOCs…

2. Line 78: Capital P in Pitts

**Reply:** Thanks for your valuable suggestions! Modifications have been made in the revised manuscript.

Line 75 …Finlayson-Pitts et al.,…

3. Figure 1 Schematic: I think you mean 'humidity sensor'. Also "Controller" is missing the 'r' on the end.

**Reply:** Thanks for your valuable suggestions! Modifications have been made in the revised manuscript.

[Figure]

Figure 1. A schematic of the experimental device (MFC: Mass Flow Controller; UV: ultraviolet lamp)

4. Line 127: Delete the word 'that'

**Reply:** Thanks for your valuable suggestions! Modifications have been made in the revised manuscript.

Line 120: The results indicated  minimal impact…

\

5. Line 217, also Line 460: McGillen should have a capital "G".

**Reply:** Thanks for your valuable suggestions! Modifications have been made in the revised manuscript.

Line 205: …McGillen et al., 2020).

6. Line 237: Replace 'by the' with 'of'.

**Reply:** Thanks for your valuable suggestions! Modifications have been made in the revised manuscript.

Line 225: …values of the database of McGillen et al.

7. Line 244: Change 'branching' to 'branched'.

**Reply:** Thanks for your valuable suggestions! Modifications have been made in the revised manuscript.

Line 232: For branched alkanes, for example…

8. Line 340: Delete 'are'.

**Reply:** Thanks for your valuable suggestions! Modifications have been made in the revised manuscript.

Line 327: …most n-alkanes are fall into the shaded region…

9. Line 410: There is a bracket missing after 340 K.

**Reply:** I'm sorry for the mistake! Modifications have been made in the revised manuscript.

Line 394: …(Arrhenius expression: $k(T)=(2.43\pm0.52)\times10^{-11}$ exp $[-(481.2\pm60)/T]$ at 240-340 K)…

10. Line 440: This would read better as follows: "Fitting our data with evaluated data..."

**Reply:** Thanks for your valuable suggestions! Modifications have been made in the revised manuscript.

Line 424: …Fitting our data with evaluated data (manually…

11. Line 443: I think "for and Arey" can be deleted.

**Reply:** Thanks for your valuable suggestions! "for and Arey" have been deleted in the revised manuscript (Line 427).

12. Line 461-462: I think this should read as follows: "Compared to the database of McGillen et al. at 240-1464 K, the activation energy (Ea/R) obtained in this work is similar, however, the A-factor obtained (4.48) is about 17% higher."

**Reply:** Thanks for your valuable suggestions! Modifications have been made in the revised manuscript.

Line 446-448: Compared to the database of McGillen et al. (McGillen et al., 2020) at 240-1464 K, the activation energy (Ea/R) obtained in this work is similar, however, the A-factor obtained (4.48) is about 17% higher.

13. Line 508: It might be better to start a new paragraph for the discussion of the 2-methylhexane data.

**Reply:** Thanks for your valuable suggestions! Modifications have been made in the revised manuscript (Line 494).

14. Line 515: Change "has certain reference value" to "is reliable".

**Reply:** Thanks for your valuable suggestions! Modifications have been made in the revised manuscript.

Line 502: …expression is reliable.

15. Line 581: The last sentence would read better as follows: "Longer atmospheric residence times apply for short-chain alkanes compared to long-change C8-C11 alkanes; for example, propane has a lifetime of about 11 days.

**Reply:** Thanks for your valuable suggestions! Modifications have been made in the revised manuscript.

Line 567-569: Longer atmospheric residence times apply for short-chain alkanes compared to long-change $C_8$-$C_{11}$ alkanes; for example, propane has a lifetime of about 11 days.

16. Line 600: Change "Whilst" to "Also".

**Reply:** Thanks for your valuable suggestions! Modifications have been made in the revised manuscript.

Line 587: Also, the value of preexponential factor A increases with the increase…